**Arctic spring and summertime aerosol optical depth baseline from**
**long-term observations and model reanalyses - Part 1: climatology**
**and trend**

Peng Xian[1], Jianglong Zhang[2], Travis D. Toth[3], Blake Sorenson[2], Peter R. Colarco[4],
Zak Kipling[5], Norm T. O'Neill[6], Edward J. Hyer[1], James R. Campbell[1], Jeffrey S. Reid[1]
and Keyvan Ranjbar[6]
[1]Naval Research Laboratory, Monterey, CA, USA.
[2]Department of Atmospheric Sciences, University of North Dakota, Grand Forks, ND
[3]NASA Langley Research Center, Hampton, Virginia, USA.
[4]NASA Goddard Space Flight Center, Greenbelt, MD, USA.
[5]European Centre for Medium-Range Weather Forecasts, Reading, UK.
[6]Département de géomatique appliqué, Université de Sherbrooke, Sherbrooke, Québec,
Canada
Correspondence: Peng Xian (peng.xian@nrlmry.navy.mil)

**Abstract**

We present an Arctic aerosol optical depth (AOD) climatology and trend analysis for 2003-2019 spring and summertime periods derived from a combination of multi-agency aerosol reanalyses, remote sensing retrievals, and ground observations. This includes the U.S. Navy Aerosol Analysis and Prediction System ReAnalysis version 1 (NAAPS-RA v1), the NASA Modern-Era Retrospective Analysis for Research and Applications, version 2 (MERRA-2), and the Copernicus Atmosphere Monitoring Service ReAnalysis (CAMSRA). Space-borne remote sensing retrievals of AOD are considered from the Moderate Resolution Imaging Spectroradiometer (MODIS), the Multi-angle Imaging SpectroRadiometer (MISR), and Cloud-Aerosol Lidar with Orthogonal Polarization (CALIOP). Ground-based data include sun photometer data from Aerosol Robotic Network (AERONET) sites and oceanic Maritime Aerosol Network (MAN) measurements. Aerosol reanalysis AODs and space-borne retrievals show consistent climatological spatial patterns and trends for both spring and summer seasons over the lower-Arctic (60-70°N). Consistent AOD trends are also found for the high Arctic (north of 70°N) from reanalyses. The aerosol reanalyses yield more consistent AOD results than climate models, verify well with AERONET, and corroborate complementary climatological and trend analysis. Speciated AODs are more variable than total AOD among the three reanalyses, and a little more so for March-May (MAM) than for June-August (JJA). Black Carbon (BC) AOD in the Arctic comes predominantly from biomass burning (BB) sources in both MAM and JJA, and BB overwhelms anthropogenic sources in JJA for the study period.

AOD exhibits a multi-year negative MAM trend, and a positive JJA trend in the Arctic during 2003-2019, due to an overall decrease in sulfate/anthropogenic pollution, and a significant JJA increase in BB smoke. Interannual Arctic AOD variability is significantly large, driven by fine-mode, and specifically, BB smoke, with both smoke contribution and interannual variation larger in JJA than in MAM. It is recommended that climate models should account for BB emissions and BB interannual variabilities and trends in Arctic climate change studies.

## 1. Introduction

The Arctic is warming faster than the overall global climate, a phenomenon widely known as Arctic amplification (Serreze and Francis 2006; Serreze and Barry 2011). This has led to rapid changes in regional sea ice properties. September sea ice coverage is shrinking at an unprecedented rate (Comiso 2012; Meier et al., 2014). Younger and thinner ice is replacing thick multi-year sea ice (Kwok and Rothrock 2009; Hansen et al, 2013; Rosel et al. 2018). Mechanisms contributing to sea ice changes include increased anthropogenic greenhouse gases (Notz and Stroeve 2016; Dai et al., 2019), sea ice-albedo feedback (Perovich and Polashenski 2012), increased warm and moist air intrusion into the Arctic (Boisvert et al. 2016; Woods et al., 2016; Graham et al. 2017), radiative feedbacks associated with cloudiness and humidity (Kapsch et al. 2013; Morrison et al. 2018), and increased ocean heat transport (Nummelin et al., 2017; Taylor et al. 2018). However, one of the least understood factors of Arctic change is the impact of aerosols on sea ice albedo and concentration (IPCC 2013).

Atmospheric aerosol particles from anthropogenic and natural sources reach the Arctic region through both long-range transport and local emissions, affecting regional energy balance through both direct and indirect radiative processes (Quinn et al., 2008; Engvall et al., 2009; Flanner, 2013; Sand et al., 2013; Markowicz et al., 2021; Yang et al., 2018). Aerosol particles influence cloud microphysical properties as cloud condensation nuclei (CCN) and/or ice nuclei (IN), affecting cloud albedo, lifetime, phase, and probability of precipitation (e.g., Lubin and Vogelmann, 2006; Lance et al., 2011; Zamora et al, 2016; Zhao and Garrett 2015; Bossioli et al., 2021). Additionally, deposition of light-absorbing aerosol species such as dust and black/brown carbon on the surface of snow and ice can trigger albedo feedbacks and facilitate melting and prolong melting seasons (Hansen & Nazarenko, 2004; Jacobson, 2004; Flanner et al., 2007; Skiles et al., 2018; Dang et al., 2017; Kang et al., 2020). However, the impact of aerosol particles on polar climate change is still not well characterized, and their relative importance compared to other warming factors is difficult to isolate and quantify.

Climate modeling studies show that due to stronger feedback processes between the atmosphere-ocean-sea-ice-land the Arctic region is more sensitive to local changes in radiative forcing than tropical and mid-latitude regions (Shindell and Faluvegi 2009; Sand et al., 2013). On the other hand, there seems to be an emerging agreement on a higher sensitivity of Arctic clouds by aerosol particles than lower-latitude regions due to the very low aerosol amounts compared to lower latitudes (Prenni et al., 2007; Mauritsen et al. 2011; Birch et al., 2012; Coopman et al., 2018; Wex et al., 2019). Both underscore the important role aerosol particles may play in the Arctic weather and climate, and the urgency to better quantify the amount of aerosols in the Arctic.

A variety of atmospheric aerosol species exist in the Arctic region. Anthropogenic
pollution contributes significantly to the formation of the Arctic haze, which generally
occurs in later winter and spring due to wintertime build-up in the shallow boundary
layer with effective transport and reduced removal (e.g., Law and Stohl, 2007; Quinn et
al., 2008). Biomass burning (BB) smoke, originating from wildfires in boreal North
America and Eurasia, are often observed and/or modeled being transported into the
Arctic (Eck et al. 2009; Eckhardt et al. 2015; Stohl et al. 2007; Warneke et al. 2009;
Iziomon et al., 2006; Evangeliou et al. 2016; Kondo et al., 2011; Brieder et al., 2014;
Markowicz et al. 2016; Khan et al., 2017; Engelmann et al., 2021). Airborne dust,
emitted from exposed sand or soils due to glacier retreat (Bullard et al., 2016; Groot
Zwaaftink et al., 2016), are likely on the rise as the Arctic warms. Dust can also
originate from lower latitude deserts, e.g., Sahara and Asia, and arrive in the Arctic
through long-range transport (Stone et al, 2007; Breider et al., 2014; AboEl-Fetouh et
al., 2020). As the Arctic sea-ice melts and the ice-free surface increases, emissions of
sea salt and biogenic aerosols (e.g., from dimethylsulfide; Dall et al., 2017; Gabric et al.,
2018) are expected to increase. There are also ultrafine particles nucleated from
gaseous precursors, though in small amounts (Baccarini et al., 2021; Abbatt et al.,
105  2019).

Because of the harsh surface environment endemic to the Arctic, aerosol field
measurements are limited in comparison with the mid-latitude and tropical
environments. Despite an increasing number of field campaigns carried out over the
past two decades (e.g., review by Wendisch et al., 2019; and more recently the
MOSAiC, https://mosaic-expedition.org) and their usefulness in improving process-level
understanding, field measurement periods tend to be short and limited to certain areas
and thus are not necessarily representative spatially and temporally of the whole Arctic.
There are many Arctic-aerosol optical property studies that are based on long-term site
measurements (e.g., Herber et al., 2002; Tomasi et al., 2007; Eck et al., 2009; Glantz et
al., 2014; Ranjbar et al., 2019; AboEl-Fetouh et al., 2020), however, the number of sites
is limited and of irregular spacing (mostly located at the northern edge of the North
American, Eurasian continents, and the Svalbard region).
Climate models that are not well constrained by observations exhibit large variations in
basic aerosol optical properties, with an order of magnitude difference in simulated
regional aerosol optical depth (AOD) and large differences in the simulated seasonal
cycle of AOD over the Arctic (e.g., Glantz et al., 2014; Sand et al., 2017). These results
will not reduce the uncertainty in the radiative impact of aerosols through direct
(including surface albedo effect) and indirect forcings in the Arctic climate. Impacts of
aerosols and clouds, overall, constitute one of the largest sources of uncertainty in
climate models (IPCC 2013). This is apparently exacerbated in a warming Arctic
(Goosse et al., 2018). A modeling study by DeRepentigny et al. (2021) shows that the
inclusion of interannually varying BB emissions, compared with only climatological
emissions, results in simulations of large Arctic climate variability and enhanced sea ice
loss. This finding suggests the sensitivity of climate relevant processes to aerosol
interannual variability in the Arctic.
In this paper, we present an AOD climatology and trend analysis for the 2003-2019
Arctic spring and summertime, based on a combination of multi-national interagency
aerosol reanalyses, satellite remote sensing retrievals, and ground observations. We
define the Arctic and the high-Arctic as regions north of 60°N and 70°N respectively.
The lower-Arctic is defined as regions between 60°N-70°N. To reference lower-latitude
source influences, the area of 50°N-90°N is included for context.
There are clear advantages to using aerosol reanalyses of chemical transport models in
comparison with climate models for Arctic aerosol studies. Smoke emissions are
frequently updated (hourly rather than monthly BB smoke emission sources for
example) while satellite observations of both meteorological and aerosol data are also
incorporated into those aerosol reanalyses through data assimilation. High-latitude fires
are strongly influenced by weather patterns including large-scale transport patterns
(e.g., Flannigan and Harrington 1998; Skinner et al. 1999). Thus, BB smoke in
particular, is more realistically accounted for in aerosol reanalyses.
To our knowledge, this is the first time aerosol reanalysis products are evaluated and
compared over the Arctic. The goal of the study is to provide a baseline of AOD
distribution, magnitude, speciation, and interannual variability over the Arctic during the
sea ice melting season. Statistics of Arctic extreme AOD events is provided in a
companion paper (Part 2). The baseline can be used for evaluating aerosol models,
calculating aerosol radiative forcing, and providing background information for field
campaign data analysis and future field campaign planning in a larger climate context.
This paper is organized as follows: Sect. 2 and 3 introduce the data sets and methods
respectively. Sect. 4 verifies the reanalyses. Results are reported in Sect. 5.
Discussions and conclusions are provided in Sect. 6 and 7.
**2. Data**
A combination of aerosol reanalyses, satellite-based aerosol remote sensing data, and
ground-based aerosol measurements are used to describe source dependent AOD and
its trend over the Arctic during spring (March-May, ie., MAM) and summertime (June-
August, ie., JJA). Note that "MAM" and "JJA" are meant to represent convenient and
informative acronyms for springtime and summertime. In the sections where we discuss
MAM and JJA trends we refer to, respectively, year to year trends of springtime and
summertime AODs (not seasonal trends from March to April to May or June to July to
August averaged over the multi-year sampling period). The aerosol reanalyses include
the Navy Aerosol Analysis and Prediction System reanalysis (NAAPS-RA; Lynch et al.,
2016) developed at the Naval Research Laboratory, the NASA Modern-Era
Retrospective Analysis for Research and Applications, version 2 (MERRA-2; Randles et
al., 2017), and the Copernicus Atmosphere Monitoring Service ReAnalysis (CAMSRA;
Inness et al., 2019) produced at ECMWF. The remote sensing data include AOD
retrievals from the Moderate Resolution Imaging Spectroradiometer (MODIS; Levy et
al., 2013), the Multi-angle Imaging SpectroRadiometer (MISR; Kahn et al., 2010), and
Cloud-Aerosol Lidar with Orthogonal Polarization (CALIOP). Sun photometer data from
Aerosol Robotic Network (AERONET; Holben et al., 1998) sites and oceanic Maritime
Aerosol Network (MAN, Smirnov et al., 2009) measurements. Overviews of remote
sensing techniques for Arctic aerosols can be found in Tomasi et al. (2015) and
Kokhanovsky et al. (2020). The analysis period is focused on 2003-2019, when all three
aerosol reanalyses are available. A summary of the datasets is provided in Appendix A.
2.1 MODIS AOD
AOD data from MODIS on Terra and Aqua was based on Collection 6.1 Dark Target
and Deep Blue retrievals (Levy et al., 2013). Additional quality control and some
corrections were applied as described in Zhang and Reid 2006, Hyer et al. 2011, Shi et
al. 2011, and Shi et al. 2013, and were updated for the Collection 6.1 inputs. The
quality-assured and quality-controlled MODIS C6 AOD data (550 nm) are a level 3
product that is produced at 1°x1° latitude/longitude spatial and 6-hrly temporal
resolution. Those 6-hrly (averaged) MODIS AOD data were then monthly-binned in
order to study long-term aerosol climatology and trends. Seasonal means and trends
were derived only when the total count of 1°x1° degree and 6-hrly data was greater than
10 for a season.
2.2 MISR AOD
The MISR instrument onboard the Terra satellite platform provides observations at nine
different viewing zenith angles across four different spectral bands ranging from 446 to
866 nm. These instrumental configurations facilitate AOD retrievals over bright surfaces,
such as desert regions (Kahn et al., 2010). MISR Version 23 AOD data at 558 nm
(Garay et al., 2020) were analyzed between Jan 2003 and December 2019. No MISR
AOD is available over Greenland due to snow and ice coverage. Monthly gridded MISR
AOD data were created by averaging only MISR data with 100% clear pixels, as defined
by each pixel's 'cloud screening parameter', at a spatial resolution of 1°x1°
latitude/longitude. Only monthly grid cells whose number of MODIS 100%-cloud-clear
AODs was greater than 20 were used to derive the climatology and trend.
2.3 CALIOP AOD
Cloud-Aerosol Lidar with Orthogonal Polarization (CALIOP), the primary instrument on
the Cloud-Aerosol Lidar and Infrared Pathfinder Satellite Observations (CALIPSO)
satellite, is a polarization-sensitive lidar that operates at two wavelengths (532 and 1064
nm; Winker et al. 2003).  It has, since its launch in 2006, collected a continuity of vertical
aerosol and cloud profiles. We primarily utilized daytime and nighttime 532 nm aerosol
extinction coefficient data from the Version 4.2 (V4.2) Level 2 (L2) aerosol profile
product (5 km horizontal/60 m vertical resolution) (Kim et al., 2018), with the V4.2 L2
aerosol layer product used for quality assurance (QA) procedures. The CALIOP aerosol
profiles are, as implemented and described in detail in past studies, rigorously QAed
before analysis (Campbell et al. 2012; Toth et al. 2016; 2018). Only cloud-free CALIOP
profiles are used, as determined through the atmospheric volume description (AVD)
parameter included in the aerosol profile product (i.e., we implemented a strict cloud
screening procedure for which we excluded CALIOP profiles with any range bin
classified as cloud by the AVD parameter). A significant portion of CALIOP aerosol
profile data consists of retrieval fill values (-9999s, or RFVs) that are, in part, due to the
minimum detection limits of the lidar. In fact, for some areas in the Arctic region, over
80% of CALIOP profiles consist entirely of RFVs (Toth et al. 2018). These result in
column AODs being equal to zero: including them in the composites would artificially
lower the mean AOD. They were thus excluded from our analysis. We also tested
retaining AOD=0 values in our analysis and that did not change the AOD trends (see
more discussions in section 6). Lastly, the cloud-free QAed profiles without AOD=0
profiles were used to compute mean CALIOP AODs at 2° x 5° latitude/longitude
resolution. To ensure spatial and temporal representation, seasonal means and trends
were derived only when the total count of gridded data is any season exceeded 20.
2.4 AERONET
The AErosol RObotic NETwork (AERONET) is a ground-based global sun photometer
network. AERONET instruments measure sun and sky radiance at several wavelengths,
ranging from the near-ultraviolet to the near-infrared. This network has been providing
daytime measurements of aerosol properties since the 1990s (Holben et al., 1998;
Holben et al., 2001). Only cloud-screened, quality-assured version 3 Level 2 AERONET
data (Giles et al., 2019) are used in this study.
The 500 nm fine mode (FM) and coarse model (CM) AODs from the Spectral
Deconvolution Method (SDA) of O'Neill et al. (2003), along with the FM spectral
derivative at 500 nm are used to extrapolate FM AOD to 550 nm. It is assumed the CM
AOD at 500 nm and 550 nm are equal. Total AOD is simply the sum of FM and CM
AODs. The SDA product is an AERONET product that has been verified using in situ
measurements (see for example Kaku et al., 2014) and a variety of co-located lidar
experiments (see, for example, Saha et al., 2010 and Baibakov et al., 2015). The FM
and CM separation is effected spectrally: this amounts to a separation of the FM and

CM optical properties associated with their complete FM and CM particle size distributions. This optical separation, characterized by the ratio of FM AOD to total AOD at 550 nm is referred to as the fine mode fraction (FMF). An analogous FM and CM AOD separation in terms of a cutoff radius applied to a retrieved or measured particle size distribution is referred to as the sub-micron fraction (SMF; where the numerator of the SMF is the FM AOD associated with the AOD contribution of particles below a cutoff radius). The SMF is the basis for separating FM and CM components in the AERONET (AOD & sky radiance) inversion. The SDA algorithm and the AERONET inversion generate FM and CM AODs that are moderately different (see Sect. 4 Kleidman et al., 2005). The advantage of the SDA is its significantly higher retrieval resolution (~ a few minutes versus ~ an hour for the AERONET inversion) and thus retrieval numbers, its independence from a variable cut off radius and its greater operational generality (being applicable to other networks such as the MAN sunphotometer network).

AERONET data were binned into 6-hr intervals centered at normal synoptic output times of the reanalyses (0, 6, 12, and 18 UTC) and then averaged within the bins. The monthly-mean temporal representativeness was rendered more likely by only including means with more than 18 6-hr data bins. Ten AERONET sites (Table 1, Fig. 1) were selected based on regional representativeness (coupled with the reality of the sparsity of AERONET sites in the Arctic), the availability of data records between Jan 2003 and Dec 2019 (the main study period), and for easier comparison with other Arctic studies (e.g., Sand et al., 2017). To explore the potential impact of different sampling resolutions on the results (e.g., Balmes et al., 2021), we generated daily AOD statistics (Table S1) that could be compared with Table 1 6hrly statistics. In general, the mean and median of MAM or JJA AODs (including total, FM and CM AODs) at the ten AERONET sites change very slightly (mostly 0.00, or <=0.01). The daily AOD standard deviation was less than its 6hrly analogue.

We found that thin clouds could occasionally be identified and retrieved as CM aerosols in level 2, version 3 AERONET data. These retrievals were manually removed by identifying such thin clouds using Terra and Aqua visible-wavelength imagery from NASA Worldview and comparing 6-hrly NAAPS-RA with AERONET AODs. CM AODs greater than 3-sigma level were then also removed (as per AboEl-Fetouh et al., 2020).

2.5 MAN AOD

The Marine Aerosol Network (MAN) is a hand-held Microtops sun photometer (research vessel) counterpart to AERONET employed for ocean measurements in areas where no-land based AERONET site can exist (Smirnov et al., 2009, 2011). The products share AERONET nomenclature and data processing is similar to that of AERONET.

Level 2 data above 70°N for the period of 2003-2019 were employed in this study. SDA-
based FM and CM AOD at 550 nm were derived and averaged over 6-hr time bins.
2.6 NAAPS AOD reanalysis v1
The Navy Aerosol Analysis and Prediction System (NAAPS) AOD ReAnalysis (NAAPS-
RA) v1 provides 550 nm speciated AOD at a global scale with 1°x1° degree spatial and
6-hrly temporal resolution for the years 2003-2019 (Lynch et al., 2016). This reanalysis
is based on NAAPS with assimilation of quality-controlled retrievals of AOD from
MODIS and MISR (Zhang et al., 2006; Hyer et al., 2011; Shi et al., 2011). AODs from
anthropogenic and biogenic fine aerosol species (ABF, a mixture of sulfate, BC, organic
aerosols and secondary organic aerosols from non-BB sources), dust, biomass-burning
smoke, and sea salt aerosols are available. The aerosol source functions were tuned to
obtain the best match between the model FM and CM AODs and the AERONET AODs
for 16 regions globally. Wet deposition processes were constrained with satellite-
derived precipitation (Xian et al., 2009). The reanalysis reproduces the decadal AOD
trends found using standalone satellite products (e.g., Zhang et al., 2010; 2017 who
excluded polar regions due to lack of verification data).
2.7 MERRA-2 AOD reanalysis
NASA Modern-Era Retrospective Analysis for Research and Applications, version 2
(MERRA-2) includes aerosol reanalysis, which incorporates assimilation of AOD from a
variety of remote sensing sources, including MODIS and MISR after 2000. The aerosol
module used for MERRA-2 is the Goddard Chemistry, Aerosol Radiation and Transport
model (GOCART; Chin et al. 2000; Colarco et al., 2010), which provides simulations of
sulfate, black and organic carbon, dust and sea salt aerosols. A detailed description and
global validation of the AOD reanalysis product can be found in Randles et al. (2017)
and Buchard et al. (2017). For this study, monthly mean speciated AODs and total AOD
at 550 nm with 0.5° latitude and 0.625° longitude spatial resolution were used.
2.8 CAMSRA AOD reanalysis
The Copernicus Atmosphere Monitoring Service (CAMS) Reanalysis (CAMSRA, Inness
et al., 2019) is a new global reanalysis of atmospheric composition produced at the
ECMWF. It followed on the heels of the MACC reanalysis (Inness et al., 2013) and
CAMS interim reanalysis (Flemming et al., 2017). The dataset covers the period of
2003–2020 and is being continued for subsequent years. The model is driven by the
Integrated Forecasting System (IFS) used at ECMWF for weather forecasting and
meteorological reanalysis (but at a coarser resolution). It incorporates additional
modules activated for prognostic aerosol species (dust, sea salt, organic matter, black
carbon and sulfate) and trace gases. The radiative impact of aerosol particles and
ozone on meteorology is included. Satellite retrievals of total AOD at 550 nm are
assimilated from MODIS for the whole period, and from the Advanced Along-Track
Scanning Radiometer for 2003–2012, using a 4D variational data assimilation system
with a 12-hour data assimilation window along with meteorological and trace gas
observations. The speciated AOD products are available at a 3-hourly temporal
resolution and a ~0.7° spatial resolution, and monthly mean AODs at 550 nm were used
in this study. Model development has generally improved the speciation of aerosols
compared with earlier reanalyses, and evaluation against AERONET globally is largely
consistent over the period of the reanalysis.
2.9 Multi-reanalysis-consensus (MRC) AOD
All three of the individual reanalyses are largely independent in their underlying
meteorology and in their aerosol sources, sinks, microphysics, and chemistry. They
were also generated through data assimilation (DA) of satellite and/or ground-based
observations of AOD. The assimilation methods, and the assimilated AOD observations,
including the treatments of the observations prior to assimilation (quality control, bias
correction, aggregation, and sampling, etc.), often differ. There is, on the other hand,
consistent use of MODIS data with its daily global spatial coverage.
Based on the three aerosol reanalysis products described above, we made an MRC
product following the multi-model-ensemble method of the International Cooperative for
Aerosol Prediction (ICAP, Sessions et al., 2015; Xian et al., 2019). The MRC is a
consensus mean of the three individual reanalyses, with a 1°x1° degree spatial and
monthly temporal resolution. Speciated AODs and total AOD at 550 nm for 2003-2019
are available. This new product is validated here, along with the three component
reananlysis members, using ground-based Arctic AERONET observations. Validation
results in terms of bias, RMSE, and coefficient of determination ($r^2$) for monthly-mean
total, FM and CM AODs are presented in Tables 2, 3, 4. The MRC, in accordance with
the ICAP multi-model-consensus evaluation result, is found to generally be the top
performer among all of the reanalyses for the study region.
2.10 Fire Locating and Modeling of Burning Emissions (FLAMBE) v1.0
FLAMBE is a biomass-burning emission inventory derived from a satellite-based active
fire hotspot approach (Reid et al., 2009; Hyer et al., 2013). FLAMBE can take satellite
fire products from either geostationary sensors, which offer faster refresh rates and
observation of the full diurnal cycle, or polar orbiters, which have a greater sensitivity.
There are significant daily sampling biases and additional artifacts from day to day shifts
in the orbital pattern for polar-orbiting satellites (e.g., Heald et al., 2003, Hyer et al.,
2013). However, the polar-only version of FLAMBE, which employed MODIS-based fire
data, is more appropriate for reanalysis and trend analysis. This is because multiple
changes in the geostationary constellation over the stud period posed a challenge in
terms of smoke source-function consistency. The FLAMBE MODIS-only smoke source
was also used in the NAAPS-RA v1 because of the same temporal consistency
requirement. FLAMBE show similar BB emission trends as the time series of yearly BB
emission for the Arctic region based on other inventories for a similar study period
(using BC emission of Fig. 2 in McCarty et al., 2021). These inventories include the
Global Fire Assimilation System (GFAS; Kaiser et al., 2012), and the Global Fire
Emission Dataset (GFED; Randerson et al., 2006; van derWerf et al., 2006).
**3. Method**
The Arctic AOD climatology and trends are analyzed in this study using remote sensing
products derived from MODIS, MISR, CALIOP, and AERONET (each sensor typically
generating aerosol products of different native wavelengths). The 550 nm AOD was
employed as the benchmark parameter for this study since the three aerosol reanalyses
AODs and the MODIS AOD are all available at 550 nm while the 558nm and 532nm
AODs of MISR and CALIOP are appreciably close to 550 nm. AERONET and MAN
modal AODs at 550 nm were derived using the SDA method as described in Sect. 2.4
and 2.5. Arithmetic means were employed for all the data processing in order to be
consistent with the arithmetic statistics that are usually reported in the literature and with
the arithmetic statistics of the monthly data from the aerosol reanalyses. Various studies
have shown that geometric statistics are more representative of AOD histograms (see,
for example, Hesaraki et al., 2017 and Sayer et al., 2019). However, Hesaraki et al.
(2017) showed that arithmetic statistics could be employed to readily estimate
geometric statistics[1]. This option effectively renders the reporting of arithmetic or
geometric statistics less critical.
The species of interest are biomass burning (BB) smoke, anthropogenic and biogenic
fine aerosols (ABF) in NAAPS, and its equivalent of sulfate for MERRA-2 as well as
CAMSRA and dust and sea salt aerosols. Anthropogenic aerosol particles, as an
external climate forcer, have drawn some attention in climate studies (e.g., Wang et al.,
2018; Ren et al., 2020; Yang et al., 2018; Sand et al., 2016; Eckhardt et al., 2015;
Brieder et al., 2017). However, BB smoke, which can be both natural and
anthropogenic in origin, has been shown to be the largest contributor (over the last two
decades) to Arctic summer AOD and concentration (Evangeliou et al. 2016; Sand et al.
2017 for modelling studies and Eck et al. 2009; Eckhardt et al. 2015; Stohl et al. 2007;
Warneke et al. 2009 for observational-based studies). Recent measurements of BC in
Arctic snow also show a strong association with BB based on tracer correlations and
optical properties (Hegg et al., 2009; Doherty et al., 2010; Hegg et al., 2010; Khan et al.,

---

[1] with an erratum: the equation (2) transformation to geometric mean should be $\tau_{g,x} = \frac{<\tau_x>}{\exp\left(\frac{\ln^2 \mu_x}{2}\right)}$

2017). A climate modeling study recently found that much larger Arctic climate variability
and enhanced sea ice melting were introduced using BB emissions with interannual
variability as opposed to climatological monthly-mean BB emissions (DeRepentigny et
al., 2021), a result that underscored the importance of quantifying the magnitude and
interannual variability of BB smoke in Arctic climate forcing estimates. Thus BB smoke
AOD is separated out from the total AOD as a singularly important species in this study.
The separation of species in this analysis is a bit arbitrary since the representation of
different aerosol types and sources in each reanalysis is slightly different. The NAAPS
model is unique compared to other reanalyses and operational models in that it carries
aerosol species by source rather than chemical speciation. For example, biomass
burning and a combined ABF are carried as separate species and permit explicit
hypothesis testing about the sources, sinks, and optical properties. Conversely,
MERRA-2 and CAMSRA carry organic carbon (OC)/organic matter (OM), black carbon
(BC) and various inorganic species combining a multitude of anthropogenic, biogenic
and open biomass burning source pathways. In this study the sum of OC/OM and BC
AOD is used to approximate BB smoke AOD from CAMSRA and MERRA-2. The ratio of
BC to the sum of BC and OC/OM is about 10% for areas north of 60°N on average for
both MERRA-2 and CAMSRA for both MAM and JJA (the single exception to this is that
the MERRA-2 ratio is about 20% in MAM).
It is worth noting that all the three reanalyses use hourly/daily BB smoke emission
inventories that use dynamic smoke sources detected by polar-orbiting satellites.
Examples include FLAMBE (Reid et al., 2009) for NAAPS-RA, Quick Fire Emissions
Dataset (QFED) for MERRA-2 after 2010 (GFED with monthly BB emission before
2010, Randerson et al., 2006; van derWerf et al., 2006), and Global Fire Assimilation
System (GFAS, Kaiser et al., 2012) for CAMSRA. This is expected to yield a better
spatial and temporal representation of BB smoke emissions compared to climate
models which use monthly mean BB inventories (e.g., Sand et al., 2017).
We also assume all dust and sea salt are CM, while other model aerosol species,
including ABF in NAAPS-RA, sulfate in MERRA-2 and CAMSRA, BB smoke in NAAPS-
RA, black carbon and organic carbon in MERRA-2 and CAMSRA are FM aerosol
particles. This approximation (the sequestering of dust and sea salt to the coarse mode
regime) is based on the fact that FM dust and sea salt only contribute a small portion of
the total dust or sea salt AOD at 550 nm. For example, FM mode dust represents about
30% and 39% of total dust AOD globally in MERRA-2 and CAMSRA respectively. The
numbers are 17% and 10% for sea salt. While NAAPS-RA makes the simplifying
microphysical assumption that all dust and sea salt are CM. This usage renders the FM
and CM bulk-aerosol comparisons more tractable (with the rider that we must remain
conscious of any artificial separation that might be created by any FM or CM
oversimplification).
The significance test for trend analysis applies the same calculation method as in Zhang
et al. (2010; 2017), an approach which, in turn, was based on the method of
Weatherhead et al. (1998). This trend analysis method requires a continuous time
series of data.
**4. Comparison of AODs from aerosol reanalyses and AERONET**
The number of AERONET observations are tied to the increase in the number of
daylight hours and are therefore more numerous during the summer than in the spring.
This translates to their generally being more temporally representative of 6 hr or daily
means in JJA. As a consequence, we preferentially used a JJA climatology to illustrate
reanalyses vs AERONET comparisons. Fig. 1 shows the 2003-2019 mean JJA FM and
CM AODs from AERONET and the speciated AODs from NAAPS-RA, MERRA-2, and
CAMSRA (all at 550 nm). All three aerosol reanalyses appear to capture the total AOD
magnitudes to varying extents. The AERONET retrievals show that total AOD during the
Arctic JJA season is dominated by contributions from FM aerosols. Large FM AOD
values (generally indicative of strong BB smoke influence) are found in Yakutsk and
Tiksi in Siberia, and Bonanza Creek in Alaska. CM aerosols also contribute a
substantial fraction, varying from a minimum of 15% in regions close to BB smoke
sources to a maximum of ~25% at the Norwegian Sea and Greenland Sea coastal sites
(Hornsund, Andenes, and Ittoqqortoormitt): these sites are likely impacted by sea salt
aerosols lifted by North Atlantic cyclonic events. NAAPS-RA produces AERONET-
comparable FM and total AODs in general while showing a tendency to overestimate
CM AODs (see Table 2 for explicit biases). The other two reanalyses (MERRA-2 and
CAMSRA) produce higher FM AOD and total AOD and lower CM AOD compared to
AERONET (see also Table 2).
Differences exist between the three reanalyses with respect to the FM and CM
partitioning of aerosol species. For example, sea salt aerosols always dominate in the
CAMSRA (dust + sea salt) CM: this comment even applies to some inland sites (e.g.,
Bonanza-Creek) and implies a modeling issue. Dust is the dominant CM species in
NAAPS-RA and MERRA-2. This latter result was found at all AERONET site positions: it
is likely attributable to elevated dust layers transported from lower latitudes (Stone et al,
2007; Jacob et al., 2010; Breider et al., 2014; Aboele-Fetouh et al., 2020). The
proportional contribution of dust to total AOD is the largest in NAAPS-RA: a result that
could have contributed to its high bias in CM AOD (Table 2). The contribution of organic
matter to FM AOD is generally larger in CAMSRA than in the other two reanalyses. On
the whole, BB smoke is the largest contributing species to total JJA AOD over the
Arctic. This is consistent across all the reanalyses except for some sites in NAAPS-RA
(e.g., Andenes, Hornsund, and Kangerlussuaq where ABF AOD is slightly larger than
BB smoke AOD). This can be partially due to the different types of speciation employed
in NAAPS-RA: ABF includes anthropogenic and biogenic pollution aerosols, including
sulfate, BC and organic aerosols of all origins (except for biomass burning aerosols). It
is also worth noting that mean AERONET AODs are, in general, higher (0.01-0.02, and
can be ~0.1 higher for the sites close to BB sources) than their median counterparts
(Table 1) as well as their geometric means. This is because AOD histograms are
typically more lognormal than normal in form (asymmetric linear-AOD histograms with
positively skewed tails as per, for example, Hesaraki et al., 2017): arithmetic means are,
accordingly, often driven by extreme (>95% percentile for example) AOD events.
Because these extreme events constitute an important part of the Arctic aerosol
environment, the AOD means are presented here.

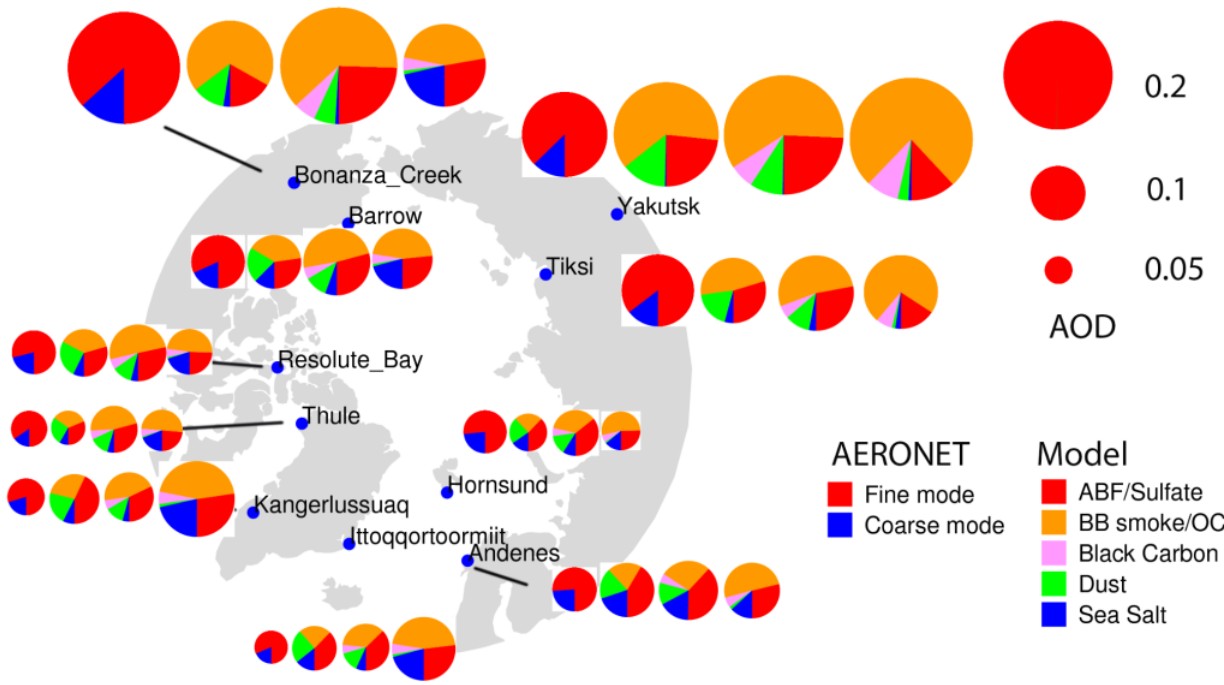

**Figure 1.** Polar projection map showing the locations of the AERONET Arctic sites
(small solid blue circles) used in this study. Long-term (2003-2019) JJA-mean FM and
CM AODs at 550 nm from AERONET (leftmost circle of each group of four circles) and
respectively, the speciated pie-charts of 550 nm AODs from NAAPS-RA, MERRA2, and
CAMSRA for each site. Warm colors (red, orange, and pink) represent fine mode and
cool colors (green and blue) represent coarse mode.
**Table 1.** Geographical coordinates of AERONET sites used in this study, and seasonal
mean total and SDA-derived FM and CM AOD at 550nm for MAM and JJA based on
2003-2019 data when available. "n" represents the number of  6-hrly AERONET data.

| sites | latitude | longitude | elev (m) | region | MAM (mean\|median\|std) | | | | MAM FMF | | JJA (mean\|median\|std) | | | | JJA FMF | |
|---|---|---|---|---|---|---|---|---|---|---|---|---|---|---|---|---|
| | | | | | total AOD | FM AOD | CM AOD | n | mean | median | total AOD | FM AOD | CM AOD | n | mean | median |
| Hornsund | 77.0°N | 15.6°E | 12 | Svalbard | 0.10\|0.09\|0.05 | 0.07\|0.06\|0.04 | 0.03\|0.02\|0.03 | 846 | 0.71 | 0.75 | 0.08\|0.06\|0.07 | 0.06\|0.04\|0.07 | 0.02\|0.01\|0.02 | 971 | 0.77 | 0.83 |
| Thule | 76.5°N | 68.8°W | 225 | Greenland | 0.08\|0.07\|0.05 | 0.06\|0.05\|0.03 | 0.03\|0.01\|0.04 | 1,009 | 0.75 | 0.81 | 0.07\|0.05\|0.07 | 0.06\|0.04\|0.06 | 0.01\|0.01\|0.02 | 1,509 | 0.85 | 0.88 |
| Kangerlussuaq | 67.0°N | 50.6°W | 320 | Greenland | 0.07\|0.06\|0.04 | 0.05\|0.04\|0.02 | 0.02\|0.02\|0.03 | 957 | 0.69 | 0.72 | 0.07\|0.05\|0.05 | 0.05\|0.04\|0.05 | 0.01\|0.01\|0.02 | 1,768 | 0.77 | 0.78 |
| Ittoqqortoormiit | 70.5°N | 21.0°W | 68 | Greenland | 0.06\|0.05\|0.04 | 0.04\|0.04\|0.02 | 0.02\|0.01\|0.03 | 545 | 0.72 | 0.78 | 0.06\|0.04\|0.04 | 0.05\|0.03\|0.05 | 0.01\|0.01\|0.02 | 1,280 | 0.80 | 0.81 |
| Andenes | 69.3°N | 16.0°E | 379 | Norway | 0.08\|0.07\|0.05 | 0.05\|0.04\|0.03 | 0.03\|0.02\|0.03 | 821 | 0.67 | 0.71 | 0.08\|0.07\|0.05 | 0.06\|0.05\|0.05 | 0.02\|0.01\|0.02 | 1,008 | 0.75 | 0.78 |
| Resolute_Bay | 74.7°N | 94.9°W | 35 | Nunavut | 0.10\|0.08\|0.05 | 0.07\|0.06\|0.04 | 0.03\|0.02\|0.03 | 520 | 0.73 | 0.78 | 0.08\|0.05\|0.10 | 0.06\|0.04\|0.10 | 0.02\|0.01\|0.03 | 1,178 | 0.78 | 0.83 |
| Barrow | 71.3°N | 156.7°W | 8 | Alaska | 0.11\|0.09\|0.07 | 0.08\|0.06\|0.05 | 0.03\|0.02\|0.04 | 605 | 0.73 | 0.77 | 0.10\|0.07\|0.15 | 0.08\|0.05\|0.15 | 0.02\|0.01\|0.02 | 1,155 | 0.79 | 0.82 |
| Bonanza_Creek | 64.7°N | 148.3°W | 353 | Alaska | 0.10\|0.08\|0.09 | 0.06\|0.04\|0.08 | 0.04\|0.03\|0.04 | 953 | 0.61 | 0.60 | 0.21\|0.09\|0.36 | 0.18\|0.06\|0.35 | 0.03\|0.02\|0.03 | 1,717 | 0.75 | 0.76 |
| Tiksi | 71.6°N | 129.0°E | 17 | Siberia | 0.10\|0.10\|0.03 | 0.08\|0.08\|0.03 | 0.02\|0.01\|0.02 | 39 | 0.80 | 0.82 | 0.13\|0.08\|0.18 | 0.11\|0.07\|0.17 | 0.02\|0.01\|0.02 | 449 | 0.80 | 0.85 |
| Yakutsk | 61.7°N | 129.4°E | 119 | Siberia | 0.15\|0.11\|0.15 | 0.11\|0.08\|0.13 | 0.04\|0.02\|0.04 | 1,516 | 0.76 | 0.80 | 0.16\|0.09\|0.24 | 0.14\|0.07\|0.24 | 0.02\|0.01\|0.02 | 2,579 | 0.81 | 0.84 |
| MAN | >70°N | - | | Arctic Ocean | 0.11\|0.10\|0.06 | 0.06\|0.06\|0.04 | 0.04\|0.04\|0.03 | 85 | 0.62 | 0.62 | 0.06\|0.05\|0.07 | 0.04\|0.03\|0.07 | 0.02\|0.02\|0.01 | 435 | 0.66 | 0.67 |


Table 1 provides the geographical coordinates of the ten AERONET sites and the
(arithmetic) mean, median and standard deviation of total, FM and CM AODs at 550 nm
for both MAM and JJA based on available 2003-2019 data (the availability of AERONET
data can be appreciated from the monthly time series in Fig. 2). Analogous MAN
statistics are provided in the last row of Table 1 (see also Fig. S1 for geographical
distributions of MAN measurements). The seasonal mean total AOD for Resolute Bay,
the Greenland sites, Hornsund and the MAN measurements are < ~ 0.1 (0.06-0.10)
while the Alaskan and Siberian site values  are  >~ 0.1 (0.10 to 0.15 with Bonanza
Creek displaying a substantially larger JJA value of 0.21). All sites, except Bonanza
Creek, tend to have moderately higher median AOD in MAM: this is consistent with
other Arctic sunphotometer studies (Tomasi et al., 2015; Xie et al., 2018). The JJA
decrease, according to the reanalyses (Fig. 4 and 5), is related to higher FM
ABF/sulfate and/or CM dust and sea salt in MAM. This AOD seasonal difference may
have evolved in the past two decades with a decreasing trend in ABF/sulfate as
discussed in Sect. 5.3. The seasonal mean AOD is greater in JJA than in MAM for
Yakutsk, Tiksi and Bonanza Creek: this is likely due to strong FM AOD variations
associated with BB smoke events (see, for example, the discussions concerning the
seasonal competition between FM AOD smoke and FM AOD Arctic haze, in AboEl-
Fetouh et al., 2020). The standard deviations of the total and FM AODs are also high for
those three sites.
The Table 1 median and mean of the FMF vary, respectively, between 0.60 to 0.88 and
0.61 to 0.85 with higher FMF in JJA than in MAM. The MAM to JJA increase is coherent
with the month-to-month increase of AboEl-Fetouh et al., (2020) although their 550 nm
arithmetic means tend to be larger (monthly-binned extremes of 0.81 to 0.98). Most, or
at least a significant part of this difference is likely attributable to differences between
our FMF (SDA) separation of the product and the SMF (AERONET-inversion)
separation of AboEl-Fetouh et al.'s climatology: the SMF is generally larger than the
FMF because it tends to attribute a fraction of the CM particle size distribution and thus
a fraction of the CM AOD to the FM AOD (see, for example, the 550 nm SMF vs FMF
comparisons Section 4 of Kleidman et al., 2005). More discussions about the
differences in terms of FMF vs. SMF and arithmetic vs. geometric statistics are available
in the supplement material.
**Table 2.** Total, FM and CM AOD bias of CAMSRA, MERRA-2, NAAPS-RA and their
consensus mean MRC compared to AERONET monthly data.

| sites | Bias-total AOD | | | | Bias-FM AOD | | | | Bias-CM AOD | | | |
|---|---|---|---|---|---|---|---|---|---|---|---|---|
| | CAMSRA | MERRA2 | NAAPS-RA | MRC | CAMSRA | MERRA2 | NAAPS-RA | MRC | CAMSRA | MERRA2 | NAAPS-RA | MRC |
| Hornsund | -0.02 | 0.01 | 0.00 | 0.00 | -0.01 | 0.01 | -0.01 | 0.00 | -0.01 | 0.01 | 0.02 | 0.00 |
| Thule | 0.00 | 0.02 | 0.00 | 0.01 | 0.01 | 0.02 | -0.01 | 0.01 | -0.01 | 0.00 | 0.01 | 0.00 |
| Kangerlussuaq | 0.02 | 0.02 | 0.02 | 0.02 | 0.03 | 0.02 | 0.02 | 0.02 | -0.01 | 0.00 | 0.02 | 0.00 |
| Ittoqqortoormiit | 0.04 | 0.03 | 0.02 | 0.03 | 0.04 | 0.02 | 0.00 | 0.02 | 0.00 | 0.01 | 0.02 | 0.01 |
| Andenes | 0.03 | 0.04 | 0.02 | 0.03 | 0.03 | 0.02 | 0.00 | 0.02 | 0.00 | 0.02 | 0.02 | 0.01 |
| Resolute_Bay | 0.01 | 0.02 | 0.01 | 0.01 | 0.03 | 0.02 | 0.00 | 0.02 | -0.02 | 0.00 | 0.01 | 0.00 |
| Barrow | 0.02 | 0.03 | 0.00 | 0.02 | 0.04 | 0.03 | -0.01 | 0.02 | -0.02 | 0.00 | 0.02 | 0.00 |
| Bonanza_Creek | 0.06 | 0.04 | 0.00 | 0.03 | 0.09 | 0.05 | 0.00 | 0.05 | -0.02 | -0.01 | 0.00 | -0.01 |
| Tiksi | 0.02 | 0.02 | -0.01 | 0.01 | 0.04 | 0.02 | -0.01 | 0.02 | -0.02 | 0.00 | 0.01 | 0.00 |
| Yakutsk | 0.03 | 0.04 | 0.01 | 0.03 | 0.05 | 0.05 | 0.00 | 0.03 | -0.02 | 0.00 | 0.01 | -0.01 |
| mean | 0.02 | 0.03 | 0.01 | 0.02 | 0.04 | 0.03 | 0.00 | 0.02 | -0.01 | 0.00 | 0.01 | 0.00 |
| median | 0.02 | 0.03 | 0.01 | 0.02 | 0.04 | 0.02 | 0.00 | 0.02 | -0.02 | 0.00 | 0.02 | 0.00 |


**Table 3.** Same as Table 2, except for RMSE.

| sites | RMSE-total AOD | | | | RMSE-FM AOD | | | | RMSE-CM AOD | | | |
|---|---|---|---|---|---|---|---|---|---|---|---|---|
| | CAMSRA | MERRA2 | NAAPS-RA | MRC | CAMSRA | MERRA2 | NAAPS-RA | MRC | CAMSRA | MERRA2 | NAAPS-RA | MRC |
| Hornsund | 0.04 | 0.02 | 0.02 | 0.02 | 0.03 | 0.02 | 0.02 | 0.02 | 0.02 | 0.01 | 0.02 | 0.01 |
| Thule | 0.02 | 0.03 | 0.02 | 0.02 | 0.03 | 0.03 | 0.02 | 0.02 | 0.02 | 0.01 | 0.02 | 0.01 |
| Kangerlussuaq | 0.03 | 0.03 | 0.03 | 0.03 | 0.04 | 0.02 | 0.02 | 0.02 | 0.01 | 0.01 | 0.02 | 0.01 |
| Ittoqqortoormiit | 0.04 | 0.03 | 0.02 | 0.03 | 0.05 | 0.03 | 0.01 | 0.02 | 0.01 | 0.01 | 0.02 | 0.01 |
| Andenes | 0.03 | 0.04 | 0.03 | 0.03 | 0.03 | 0.03 | 0.02 | 0.02 | 0.01 | 0.02 | 0.03 | 0.02 |
| Resolute_Bay | 0.03 | 0.04 | 0.02 | 0.03 | 0.04 | 0.04 | 0.02 | 0.03 | 0.02 | 0.01 | 0.02 | 0.01 |
| Barrow | 0.05 | 0.05 | 0.03 | 0.04 | 0.06 | 0.04 | 0.03 | 0.03 | 0.02 | 0.01 | 0.02 | 0.01 |
| Bonanza_Creek | 0.11 | 0.10 | 0.07 | 0.08 | 0.12 | 0.10 | 0.06 | 0.08 | 0.03 | 0.02 | 0.01 | 0.02 |
| Tiksi | 0.05 | 0.04 | 0.02 | 0.03 | 0.06 | 0.04 | 0.02 | 0.03 | 0.02 | 0.01 | 0.01 | 0.01 |
| Yakutsk | 0.07 | 0.07 | 0.04 | 0.06 | 0.08 | 0.07 | 0.04 | 0.06 | 0.03 | 0.01 | 0.01 | 0.01 |
| mean | 0.05 | 0.05 | 0.03 | 0.04 | 0.05 | 0.04 | 0.03 | 0.03 | 0.02 | 0.01 | 0.02 | 0.01 |
| median | 0.04 | 0.04 | 0.03 | 0.03 | 0.05 | 0.04 | 0.02 | 0.03 | 0.02 | 0.01 | 0.02 | 0.01 |


**Table 4**. Same as Table 2, except for r$^2$.

| sites | r2-total AOD | | | | r2-FM AOD | | | | r2-CM AOD | | | |
|---|---|---|---|---|---|---|---|---|---|---|---|---|
| | CAMSRA | MERRA2 | NAAPS-RA | MRC | CAMSRA | MERRA2 | NAAPS-RA | MRC | CAMSRA | MERRA2 | NAAPS-RA | MRC |
| Hornsund | 0.23 | 0.78 | 0.75 | 0.73 | 0.35 | 0.73 | 0.71 | 0.67 | 0.27 | 0.45 | 0.55 | 0.56 |
| Thule | 0.50 | 0.47 | 0.73 | 0.64 | 0.52 | 0.45 | 0.70 | 0.62 | 0.01 | 0.26 | 0.44 | 0.41 |
| Kangerlussuaq | 0.48 | 0.54 | 0.42 | 0.53 | 0.52 | 0.52 | 0.35 | 0.52 | 0.00 | 0.57 | 0.16 | 0.35 |
| Ittoqqortoormiit | 0.68 | 0.75 | 0.67 | 0.79 | 0.63 | 0.81 | 0.76 | 0.83 | 0.24 | 0.36 | 0.14 | 0.35 |
| Andenes | 0.67 | 0.63 | 0.68 | 0.71 | 0.68 | 0.66 | 0.64 | 0.71 | 0.10 | 0.23 | 0.21 | 0.21 |
| Resolute_Bay | 0.52 | 0.51 | 0.67 | 0.63 | 0.53 | 0.49 | 0.73 | 0.62 | 0.02 | 0.06 | 0.03 | 0.05 |
| Barrow | 0.33 | 0.68 | 0.70 | 0.62 | 0.45 | 0.76 | 0.69 | 0.68 | 0.05 | 0.27 | 0.41 | 0.41 |
| Bonanza_Creek | 0.81 | 0.78 | 0.80 | 0.83 | 0.83 | 0.79 | 0.82 | 0.85 | 0.06 | 0.43 | 0.45 | 0.46 |
| Tiksi | 0.77 | 0.80 | 0.87 | 0.84 | 0.82 | 0.82 | 0.90 | 0.86 | 0.02 | 0.20 | 0.10 | 0.15 |
| Yakutsk | 0.70 | 0.70 | 0.80 | 0.77 | 0.78 | 0.71 | 0.80 | 0.80 | 0.01 | 0.41 | 0.42 | 0.42 |
| mean | 0.57 | 0.66 | 0.71 | 0.71 | 0.61 | 0.67 | 0.71 | 0.72 | 0.08 | 0.32 | 0.29 | 0.34 |
| median | 0.60 | 0.69 | 0.72 | 0.72 | 0.58 | 0.72 | 0.72 | 0.70 | 0.04 | 0.32 | 0.31 | 0.38 |


a)

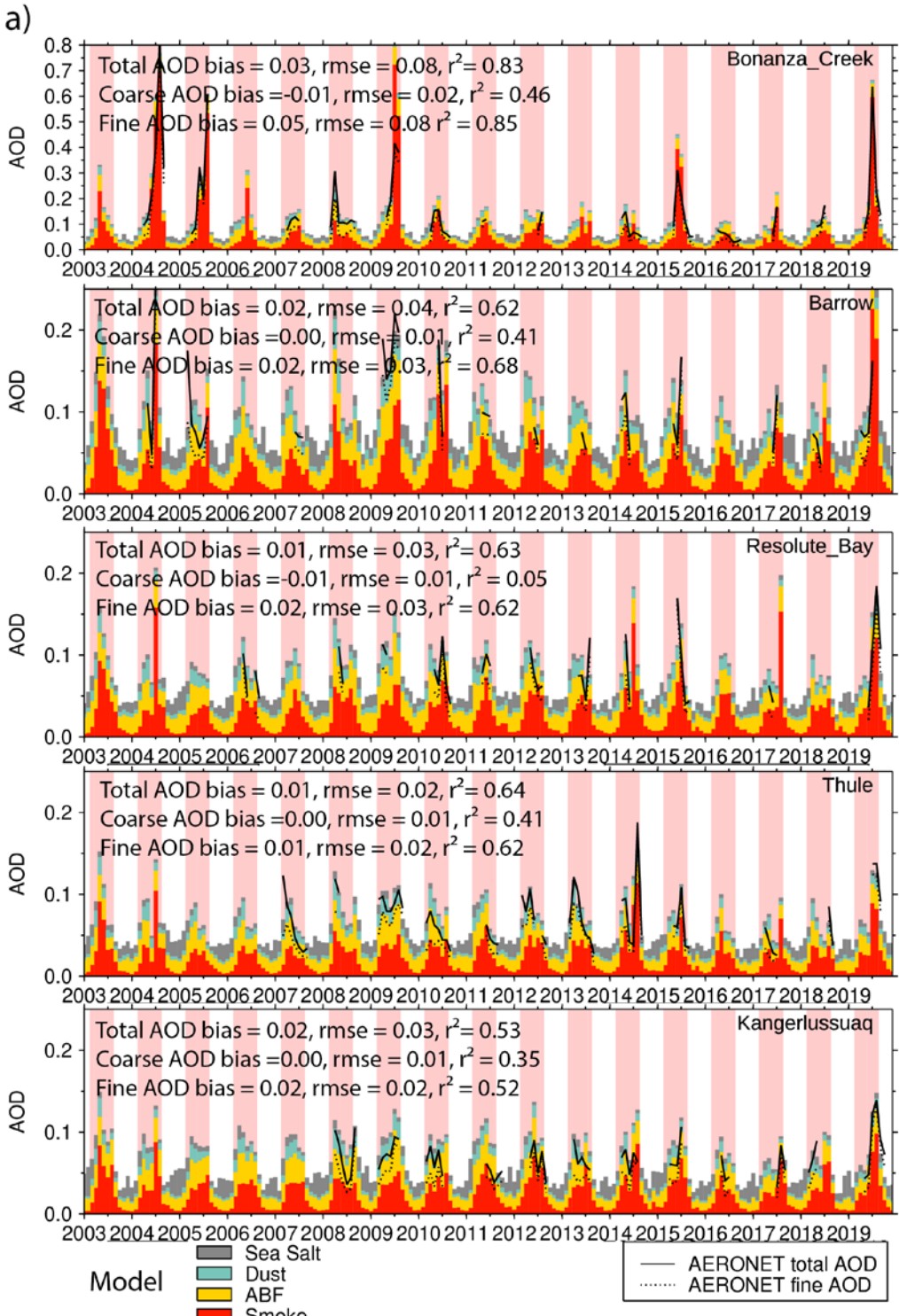


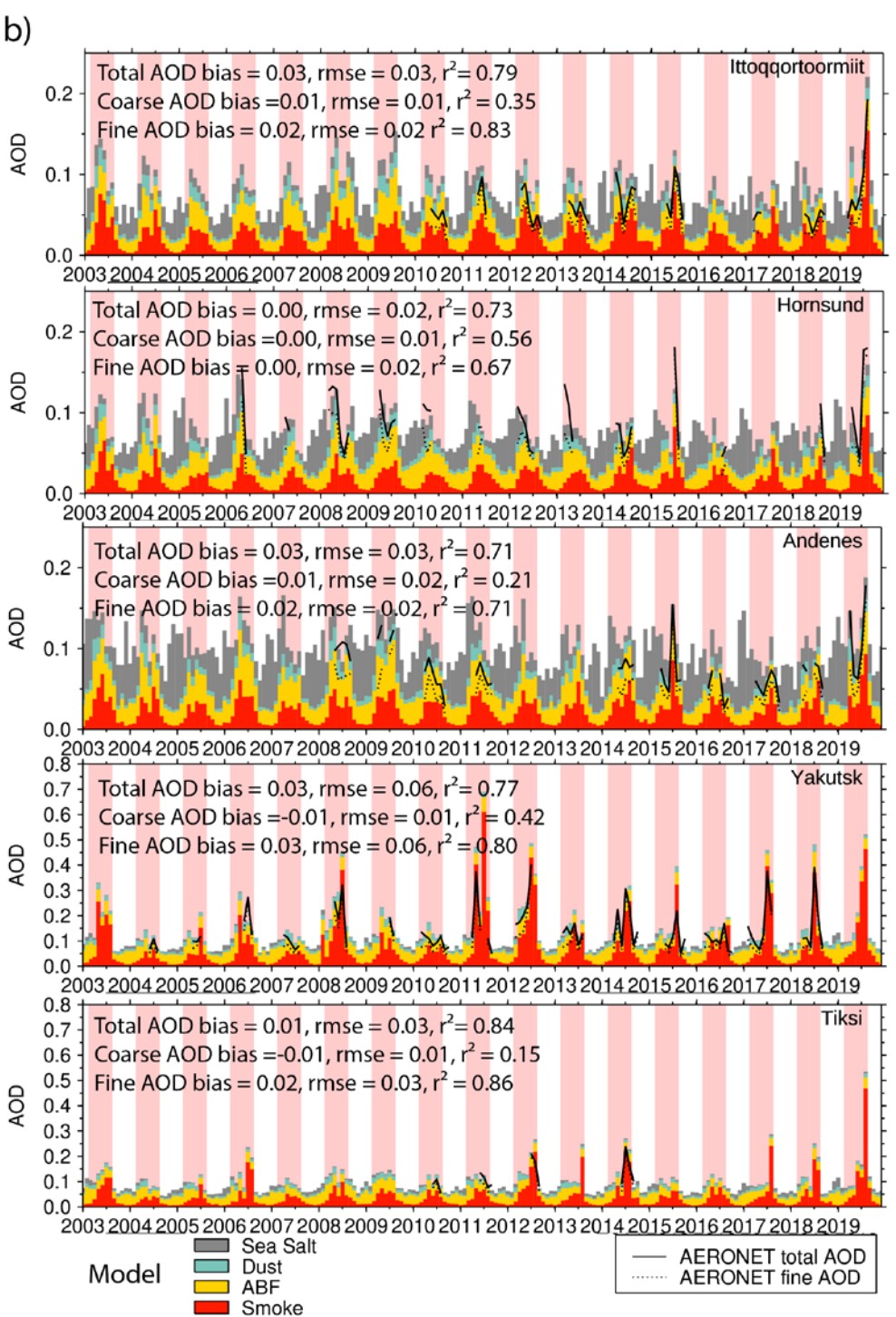

**Figure 2.** Monthly time series of FM, CM, and total AERONET AODs and MRC speciated AOD at a) Bonanza Creek, Barrow, Resolute_Bay, Thule, Kangerlussuq, and b) Ittoqqortoormitt, Hornsund, Andenes, Yakutsk, and Tiksi sites. The JJA periods are highlighted with pink shading for easy reading. The legends of each time series show MRC bias, RMSE and $r^2$. Monthly mean AERONET AODs is obtained only when the total number of 6-hr data exceeds 18 to ensure temporal representativeness.

Fig. 2 shows the time series of monthly mean FM, CM and total AODs from the ten
AERONET stations (CM AOD can be inferred from the difference between total AOD
and FM AOD) and the speciated AODs from MRC (recall the approximation of assigning
dust and sea salt to the CM, and ABF/sulfate and smoke to the FM). The MRC monthly-
binned verification statistics at the ten AERONET sites are given in the Fig. 2 legends.
Verification statistics of individual aerosol reanalysis members and the MRC based on
monthly data are presented in Tables 2, 3, and 4 for bias, RMSE, and $r^2$ respectively.
The MRC is consistently biased slightly high for FM AOD across all sites and about
neutral for CM AOD for most. As a result, total AOD tends to bias slightly high, with
biases ranging from 0.00 to 0.03. RMSE values range from 0.02 to 0.03 for most sites,
except for Bonanza Creek, Yakutsk and Barrow with RMSE values of 0.06, 0.05 and
0.04 (driven mainly by FM variations). The $r^2$ values range from 0.53 to 0.84, with FM
AOD $r^2$ values ranging from much higher to marginally higher than the CM AOD values.
This is understandable as FM AOD displays large variabilities (which models are more
capable of capturing) while CM AOD displays relatively low values and smaller absolute
variabilities on seasonal and interannual time scales. Also, emissions of CM aerosols
like dust and sea salt, are driven dynamically by model or reanalysis surface winds
where the surface wind dependency increases exponentially in amplitude: the
simulation of this dependency has been a challenge to all global aerosol models
(Sessions et al., 2015; Xian et al., 2019).
Our previous experience with multi-reanalysis and multi-model ensembles indicates, in
general, that the consensus of multi-reanalyses or multi-models show better verification
scores than individual component members (Sessions et al., 2015; Xian et al., 2019;
Xian et al., 2020). However, these studies are based on more global analyses for which
the Arctic impact is relatively weak because of the sparsity of observational Arctic data.
Tables 2, 3 and 4 indicate that the Arctic is rather unique inasmuch as the MRC is not
necessarily the top AOD-estimation performer. NAAPS-RA generally has moderately
better bias, RMSE and $r^2$ verification scores for the total and FM AODs compared to
MERRA-2 and CAMSRA while CM AOD does not perform as well. In previous MRC and
multi-model consensus evaluations, all component members either performed
comparably in terms of AOD RMSE, bias and $r^2$ or the number of multi models was
relatively larger (e.g., 5 to 6 for the International Cooperative for Aerosol Prediction
multi-model consensus). This study is the first time that all three developing centers
have systematically evaluated their AOD reanalysis performance on an Arctic-wide
climate scale.
**5. Seasonal Analysis**
In this section we present spring and summertime Arctic AOD climatologies derived
from space-borne remote sensing retrievals and aerosol reanalyses. We then present
the seasonal cycle, interannual variability and trends of total and speciated AODs.
5.1 Spring and Summertime AOD Climatology for the Arctic
5.1.1 Space-based remote sensing AOD climatology

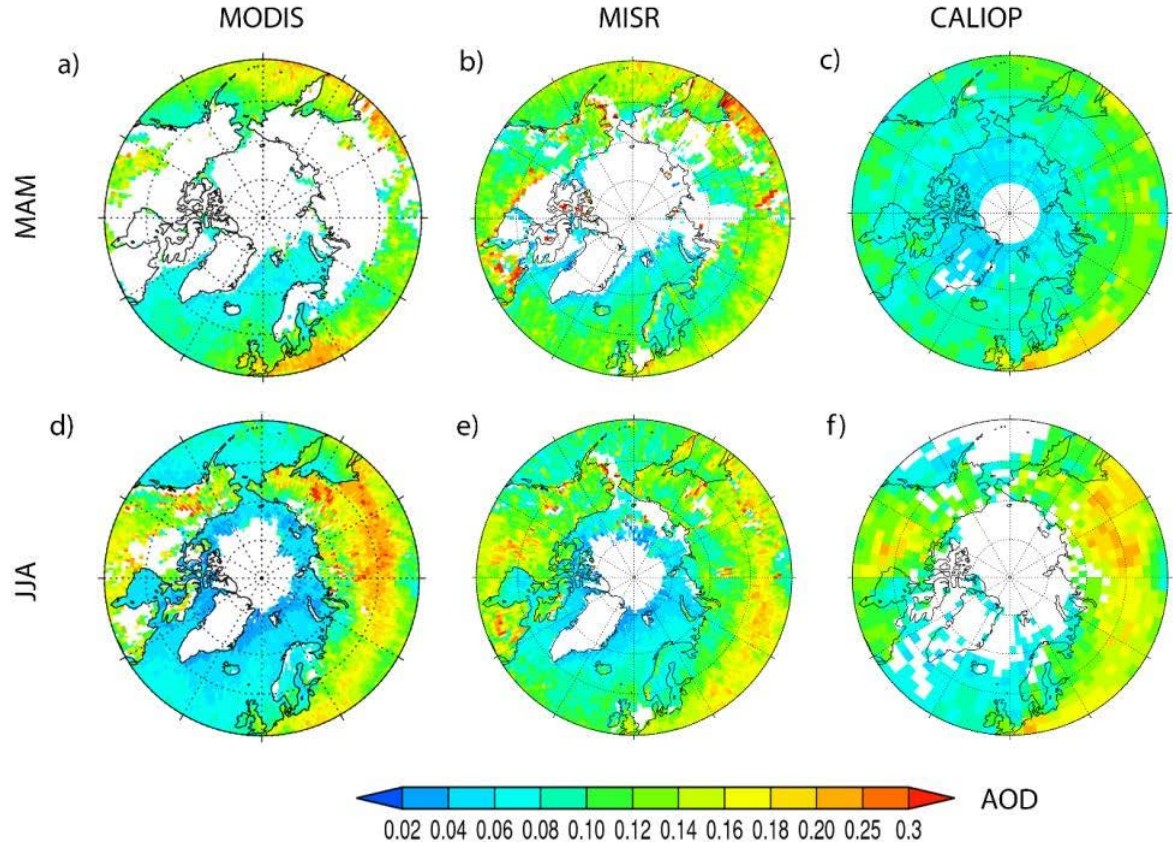

**Figure 3.** Satellite-derived, mean climatological MAM (upper) and JJA (lower) MODIS
AOD at 550 nm (left), MISR AOD at 558 nm (middle), and CALIOP AOD at 532 nm
(2006-2019, right). These are based on MODIS C6 DT+DB and MISR AOD v23 over
2003-2019, and CALIOP AOD over 2006-2019. White area means lack of data.
Bright, snow- and ice-covered surfaces, large solar zenith angles (SZA) (to the extreme
of sub-horizon SZAs during the polar night), and extensive cloud coverage result in limited
(quality assured) Arctic aerosol retrievals by passive-based sensors like MODIS and
MISR. The latitude limit of an active, downward-looking, polar-orbiting sensor like CALIOP
on CALIPSO results in a polar region profile gap above 82°N. Known issues of CALIOP
with retrieval filled values (RFVs) (Toth et al., 2018) and high noise to signal ratio over
the Arctic also limit its aerosol retrievals near the Arctic. These challenges are reflected
as no data coverage (Fig. 3) in the high Arctic and Greenland, and over large regions of
North America and Siberia in both March-April-May (MAM) and June-July-August (JJA)
in the AOD climatology maps based on MODIS, MISR, and CALIOP. Compared to MAM,
JJA has larger data coverage from MODIS and MISR over higher latitudes as aerosol
retrievals from MODIS and MISR are based on reflected sunlight. Also, when snow and
sea ice melt in summer, darker ocean and land surfaces that are suitable for applying
passive-based aerosol retrieval methods are exposed. MAM data coverage for CALIOP
is more than that of JJA due to less solar contamination during the night than during
daytime for lidars. Nevertheless, the long operation time of these sensors (about two
decades) provides sufficient data to construct a climatology for the near Arctic and the
midlatitude where most sources of Arctic aerosols reside.
In general, the AOD patterns from the three sensors are similar. High AODs of 0.15-0.25
appear in the 50°N-65°N latitude belt over land, i.e., large areas of boreal and subarctic
Siberia, east and central Europe and North America sector in both spring and summer,
with AOD mostly higher than 0.2 over Siberia in JJA, associated with biomass burning
events (Fig. 3). The average AOD over water is considerably lower, ranging from 0.02 to
0.12, with relatively high AOD in the northeast Pacific influenced by outflows from the
Eurasian Continent, and lower AOD over the north Atlantic, and the lowest (0.02-0.06)
over the Arctic Ocean. It is also visible that AOD over water is slightly higher in MAM than
in JJA, which is consistent with other observation-based studies within the Arctic circle
(e.g., Tomasi et al., 2015), possibly related to higher pollution levels from the upstream
continents in MAM. CALIOP AOD exhibits a similar spatial pattern as MODIS and MISR.
Additionally, AOD over Greenland is on the order of 0.02-0.06, and is a minimum
compared to other regions due to its high elevations (nearly 2km on average). AOD over
Siberia and North America is distinctively higher in JJA than in MAM based on CALIOP.
This seasonal difference can also be seen with MISR and can be explained by seasonal
boreal fire activities, i.e., boreal fire is generally more active in JJA than in MAM (Giglio
et al., 2013). The seemingly larger seasonal difference in CALIOP than in MODIS and
MISR over Siberia and North American could also be associated with different averaging
times (2006-2019 vs. 2003-2019, and Fig. 2) as well as data sampling rate, as the swath
for MODIS and MISR is on the order of a few hundred to a few thousand kilometers, while
the swath for CALIPSO is on the order of 70m (see e.g., Colarco et al., 2014).
5.1.2 Arctic AOD climatology derived from aerosol reanalyses
Fig. 4 and 5 show spatial distributions of 2003-2019 mean total and speciated AOD
from the three aerosol reanalyses and their consensus mean for spring and summer
respectively. Although there is limited AOD data available for DA in the Arctic, lower
latitude aerosols, whose AOD is constrained with DA, can affect Arctic AOD through
transport and thus exert an indirect AOD constraint there. Additionally, all the
reanalyses use satellite-fire-hotspot-based BB emissions with fine temporal resolution
(hourly to daily), which exert a source constraint, especially temporally (emission

magnitude differs more than timing among the different models). As a result, there are good similarities in spatial distributions of total AODs among the three reanalyses. For example, AOD values are high in the 50°N-65°N belt over the Eurasia continent and its downwind Pacific region (0.16-0.30), low and on the order of 0.1 or less for regions north of 70°N, and at a minimum over Greenland for MAM. The high AODs over boreal North America and Siberia BB regions are more prominent in JJA compared to MAM. In general, the distribution patterns and magnitude of total AOD are comparable to those derived from MODIS, MISR, and CALIOP where available to a large extent.

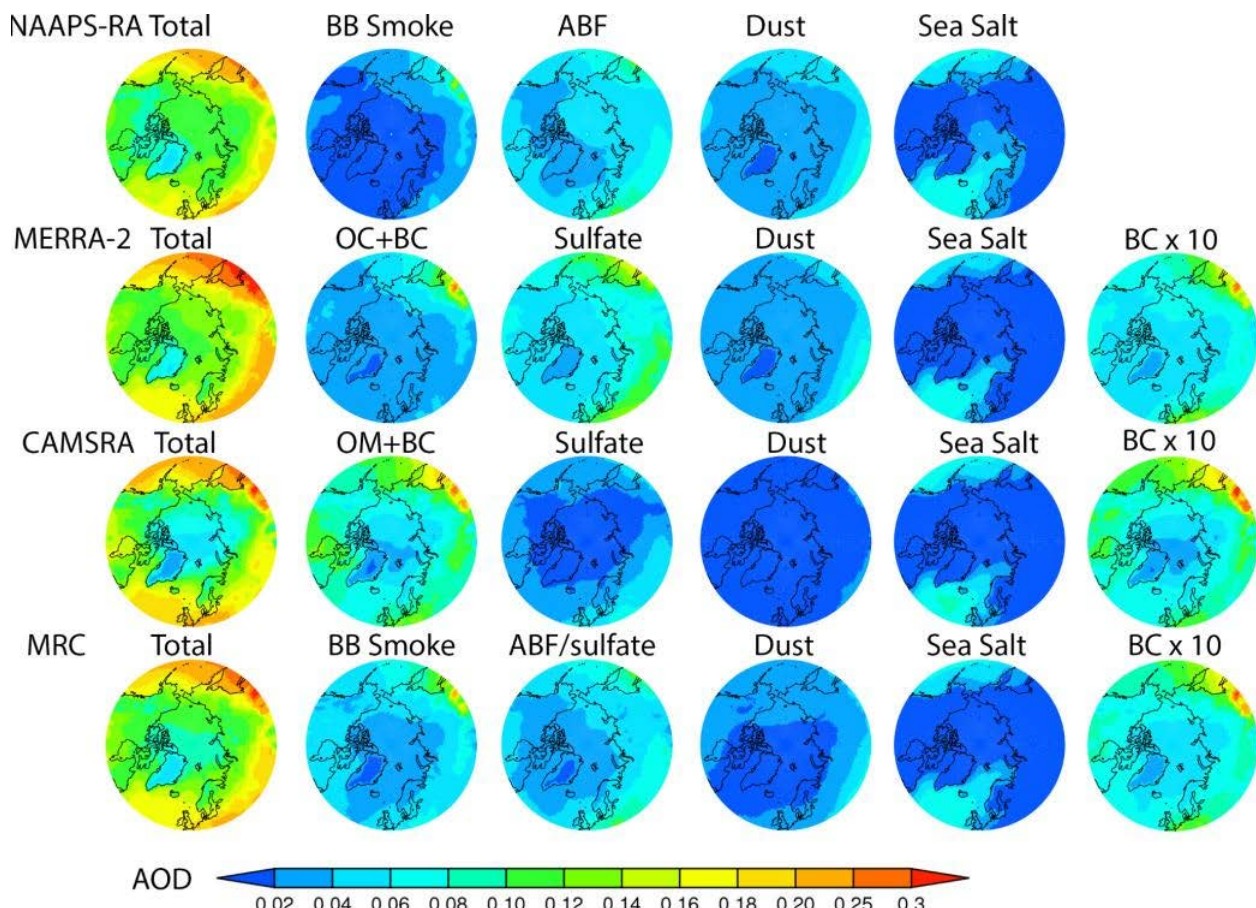

**Figure 4**. 2003-2019 Climatological MAM-mean total and speciated AOD at 550 nm from NAAPS-RA, MERRA-2 and CAMSRA over the Arctic. As MERRA2 and CAMSRA do not have a biomass-burning-induced single aerosol species, the sum of the organic carbon (OC)/organic matter (OM) and black carbon (BC) AODs is used to approximate biomass-burning smoke AOD. The ratio of BC to the sum of BC and OC/OM in MAM for area >60°N is about 18% for MERRA-2 and 10% for CAMSRA. The ratios change little for area >70°N and area >80°N.

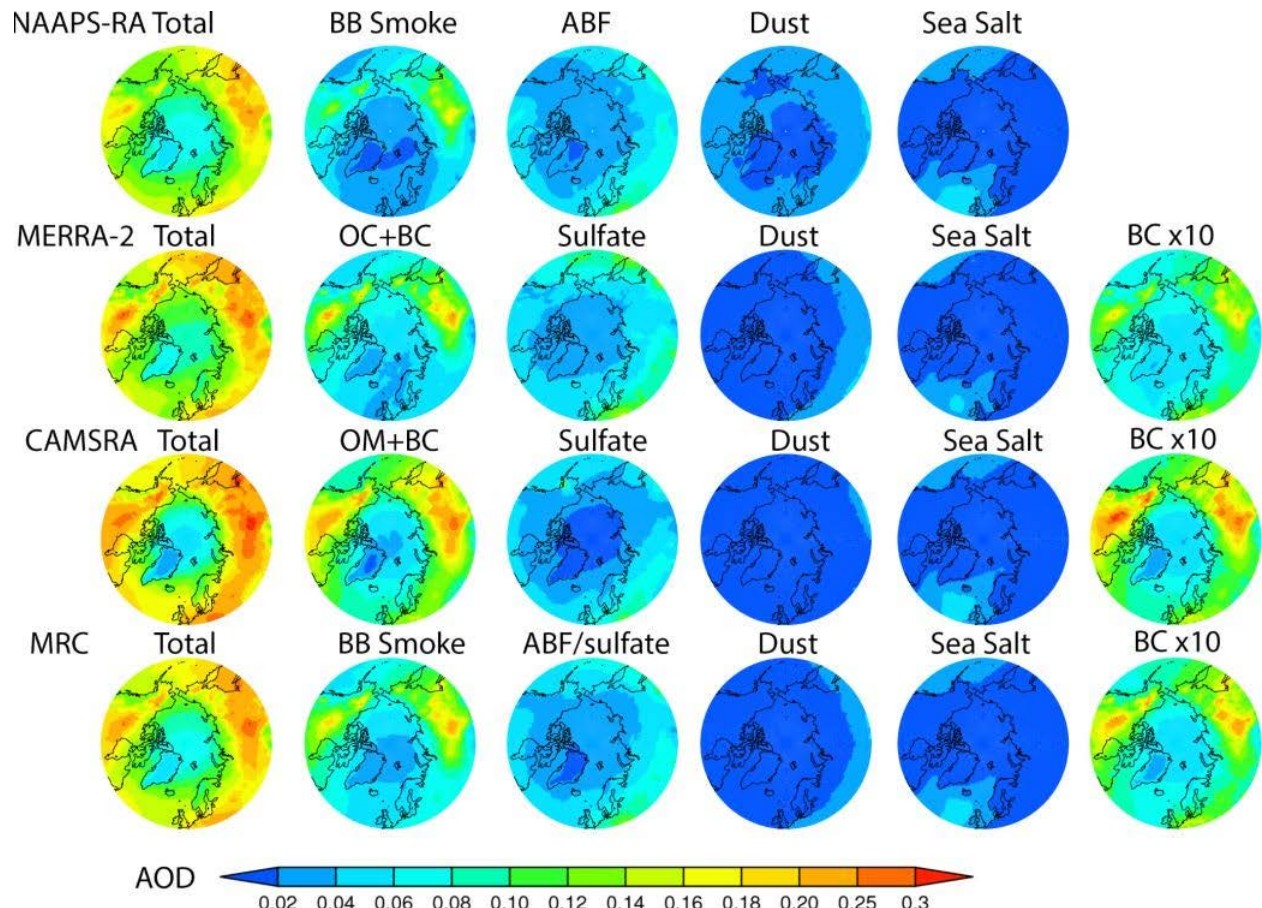


**Figure 5**. Same as Figure 4, except for JJA. The ratio of BC to the sum of BC and
OC/OM in JJA is between 10%-11% for area >60°N for both MERRA2 and CAMSRA.
This ratio changes little for area >70°N and area >80°N.
Speciated AODs have more variability than total AOD among the three reanalyses, and
a little more so for MAM than for JJA (Fig. 4, 5, 6). This is understandable because
passive retrievals of AOD are more available in summer than in spring near the Arctic,
and therefore reanalyses have more observational constraints in summer. While total
AOD is constrained through data assimilation, however, speciated AOD is not and
models must rely on their physics and boundary conditions. The MRC shows that BB
smoke and ABF/sulfate are similar in magnitude for the Arctic in MAM. However, by
model, NAAPS-RA and MERRA-2 suggest the dominance of ABF/sulfate over BB
smoke, and the reverse for CAMSRA. Based on the high bias of FM AOD verified with
AERONET (Sect. 4, Table 2), CAMSRA possibly overestimates OC and BC, and hence
BB smoke. BB smoke becomes the dominant species in JJA as boreal BB activity
increases in summer on average and ABF/sulfate turns to the 2nd place overall. The
strengthening of smoke AOD from spring to summer is a consistent feature across all
the reanalyses despite that CAMSRA tends to have higher BB smoke AOD and lower
sulfate AOD compared to the other two reanalyses in both seasons. ABF/sulfate AOD
level is slightly higher in MAM than in JJA for MRC (from slightly less than 0.04 to about
0.03 for 60-90°N regional average). A June minimum in total AOD is apparent from all
reanalyses, associated with a general decrease in ABF/sulfate, dust and sea salt AODs
after springtime and before severe BB activities in July and August. The spatial
distributions of seasonal mean BC AOD from MERRA-2 and CAMSRA greatly resemble
those of smoke AOD, and more so for JJA than MAM, except over Europe. This
suggests a dominant role of the BB source over the anthropogenic sources of BC AOD
over the Arctic for spring and summer seasons. This also supports McCarty et al.
(2021)'s BC emission estimate that wildfire emissions account for more than half of all
BC emissions north of 60°N yearly (noting much lower BB emissions during wintertime
when anthropogenic BC emission is at its maximum).
For both seasons, dust and sea salt are secondary contributors to the total AOD in the
Arctic, except for the noticeable influences of Saharan and Asian dust in spring (Stone
et al., 2007; Brieder et al., 2014) and of sea salt in the North Atlantic, Greenland Sea,
Norwegian Sea, and North Pacific associated with cyclonic activities, especially in
spring. It is also noteworthy that dust AOD in CAMSRA is much lower than the other two
models (<0.02) in the spring.
From the 10-degree zonal average, it is also seen that monthly and regional mean AOD
gradually decreases from lower latitudinal belts to higher latitudinal belts (Fig. 7). Total
AOD for the 60°-70°N belt, on average, increases from MAM to JJA due to the
seasonality of BB activities. However, the total AOD for the 80°-90°N belt decreases
slightly from MAM to JJA. This means the latitudinal gradient of total AOD is larger in
JJA than in MAM, which is most likely due to more wet removal of aerosols during
transport from source regions to the high Arctic in summer (Garrett et al., 2010, 2011). It
is also noted that the latitudinal gradient of AOD from CAMSRA is larger than those
from the two other reanalyses, suggesting stronger aerosol removal in the Arctic in
CAMSRA compared to MERRA-2 and NAAPS-RA.

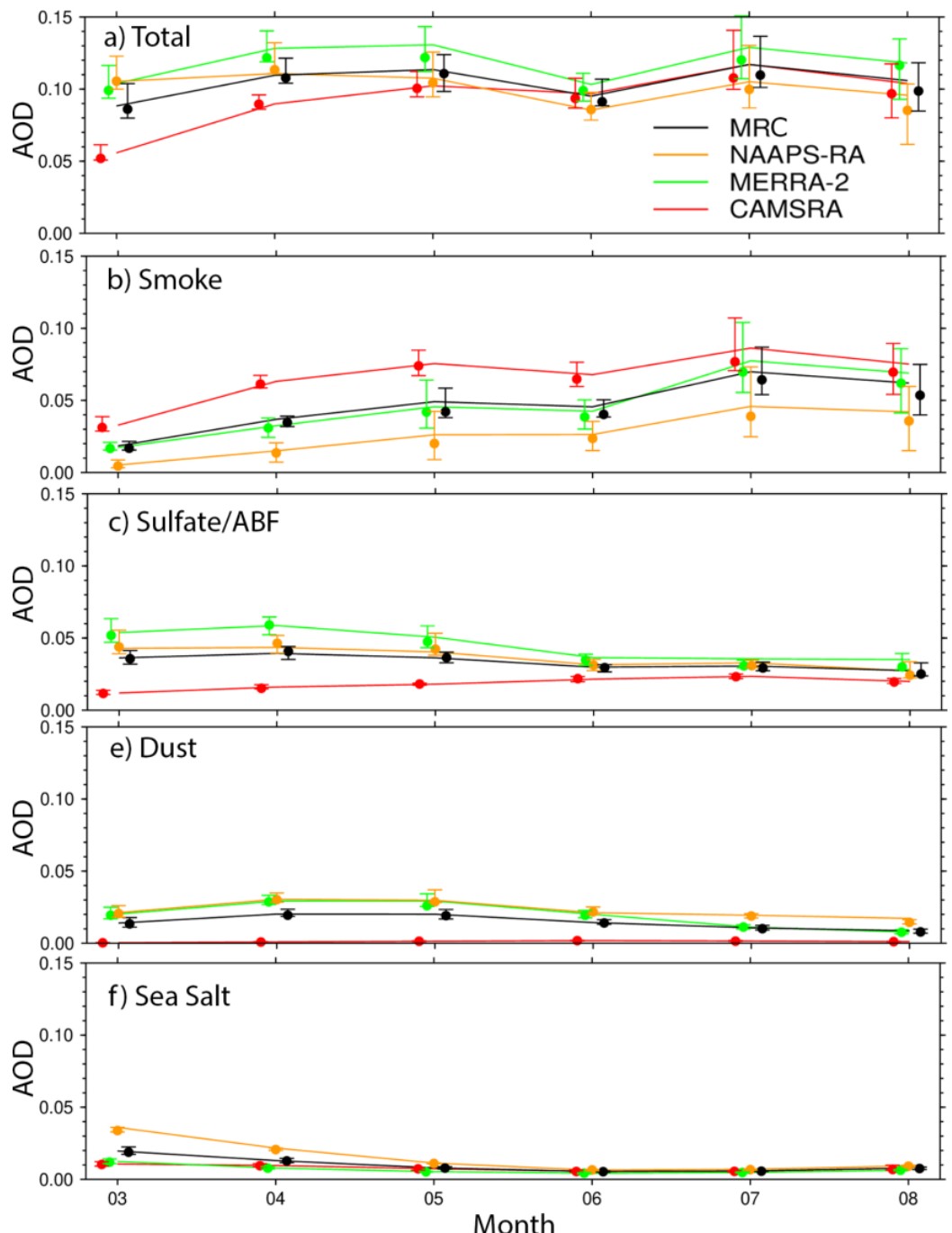

**Figure 6.** Climatological (2003-2019) seasonal cycle of Arctic (60°-90°N) average total
and speciated AODs at 550 nm from the three aerosol reanalyses and the MRC. The
top and bottom whiskers represent the 25% and 75% percentiles of monthly AODs, and
dots represent the median of monthly AODs.

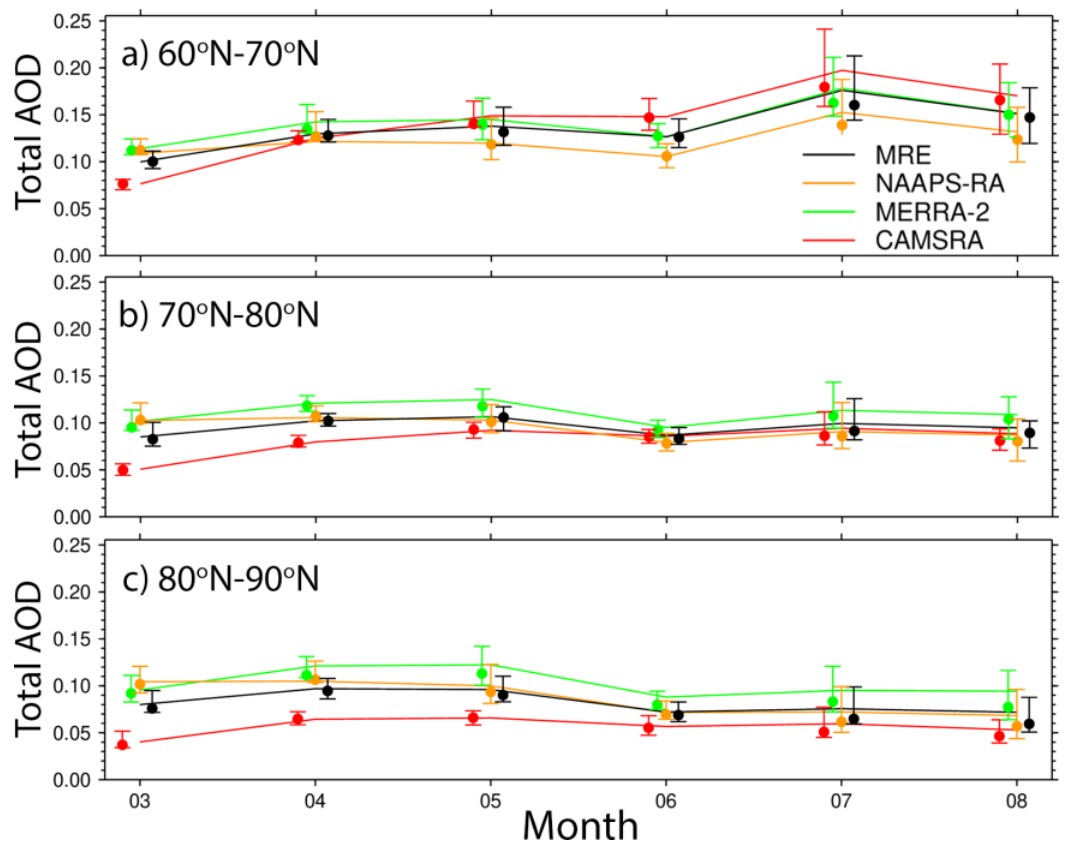

**Figure 7**. Similar to Figure 6, but for different latitudinal belts and total AOD.

## 5.2 Interannual variability of AOD in the Arctic

### 5.2.1 Interannual variability of AOD

There are, as can be seen in Fig. 2 (and supported by the MAM/JJA discussion in Sect. 4), significant interannual AOD variabilities, especially for sites close to boreal fire sources. For example, the summertime peak of the total AERONET AOD at Bonanza Creek, Alaska, is around 0.6 - 0.8 in 2004, 2005, and 2019, while it is <~ 0.1-0.2 for other years between 2003-2019. The year to year difference between high- and low-amplitude summertime peak AOD values at Yakutsk, Siberia, can be 6 fold. The MRC shows that these large interannual variabilities consistent with AERONET FM AOD variabilities, are very likely attributable to interannual variabilities in BB smoke.

For sites far from smoke sources, like Ittoqqortoormiit on the east coast of Greenland, Hornsund in Svalbard, and Thule on the northwest coast of Greenland, the high-amplitude peak AODs are about 2-3 times the low-amplitude peak AODs. This interannual spring to summer variability is also largely associated with BB smoke as suggested by the MRC and the coherent variation of the AERONET FM AOD. Some of the strongest AOD events reported in previous studies have been shown to be associated with the long-range transport of BB smoke. For instance, the strong AOD peak in the summer of 2015 over Hornsund and Andenes was shown to be associated

with a series of intense fires that originated in North America (Markowicz et al., 2016). The strong peak AODs in August 2017 over Resolute Bay, Eureka and Thule were most probably related to intense, fire-induced pyroCB events in North America and the long-range transport of high-altitude smoke (Ranjbar et al., 2019; Das et al., 2021). The high amplitude AOD peak in the spring of 2006 over Hornsund was traced to agricultural fires in Eastern Europe (Stohl et al., 2007). The boreal fires in North America in the summer 2004 led to the maximum-amplitude AOD peaks (over the 2003-2019 period of Fig. 2) for the two Alaskan sites and enhanced AOD on the pan-Arctic scale (Stohl et al., 2004). Some of the high-amplitude AOD peak events were recorded during intensive field campaigns. These included the ARCTAS/ARCPAC multi-platform campaign in the summer of 2008 (Matsui et al., 2011; Saha et al, 2010; McNaughton et al., 2011) and the NETCARE research vessel (Canadian Arctic) campaign in the spring of 2015 (Abbatt et al., 2019).

The MAN measurements and AERONET sites adjacent to the North Atlantic, the Greenland Sea, and the Norwegian Sea, notably Ittoqqortoormiit, Hornsund, and Andenes have higher CM AODs and higher CM to total AOD ratio compared to other sites: this is due to contributions from sea salt aerosols. Sea salt AOD, indicated by the MRC, is normally higher in MAM than in JJA.

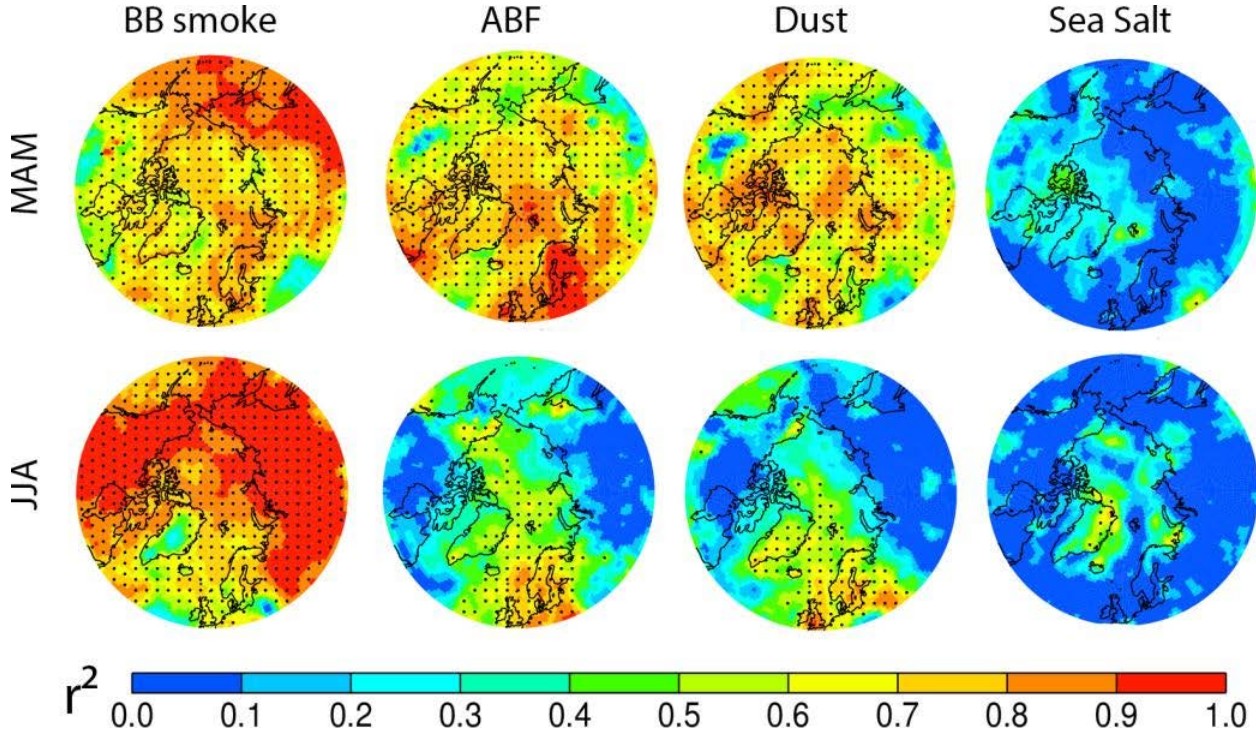

**Figure 8.** Interannual variability of MRC MAM (upper panel) and JJA-mean (lower panel) total AOD at 550 nm explained by biomass-burning smoke AOD, ABF, dust, and sea salt aerosols (i.e., the square of the correlation coefficient between speciated AOD

and total AOD) respectively. $r^2$ in dotted area is statistically significant at the 95% level
using a two-tailed Student $t$ test.
5.2.2 Attribution of AOD interannual variability
It can be observed in Fig. 6 that the simulated interannual (60-90°N) AOD variability
(represented by the Fig. 6 whisker bars) is mostly attributable to the large interannual
variability of smoke AOD (especially from May to August). This is consistent across all
the reanalysis products. For March and April, the contribution from sulfate/ABF is as
important as BB smoke, if not larger. The interannual variation of dust AODs, as
indicated with MERRA-2 and NAAPS-RA data, is non-negligible in MAM.
Regarding spatial distribution, Fig. 8 shows the interannual variabilities of spring and
summer Arctic AOD explained by different aerosol species (i.e., the square of the
correlation coefficient between speciated AOD and total AOD) suggested by MRC for
2003-2019. Consistent with the variability of monthly AOD time series shown in Fig. 2
and 6, both MAM and JJA interannual variabilities are explained mostly by BB smoke,
with a higher degree of explanation for JJA than for MAM, and a lower degree of
explanation for over the North Atlantic, Norwegian Sea and Greenland than over North
American and Eurasian sectors overall. For north of 70°N, smoke explains 60%-80% of
MAM and about 80% (except Greenland) of JJA AOD interannual variabilities. Over
North American and Eurasian sectors (>60°N), the number is about 100% for JJA. The
second-largest contributor is ABF/sulfate and dust for MAM and to a lesser extent for
JJA. Contribution from sea salt is the least and is only statistically significant east of
Greenland in JJA.
The contribution from ABF/sulfate is above 80% over the industry- and -population-
concentrated European and northeast North American sectors and their outflow regions
of the North Atlantic, Greenland Sea, Norwegian Sea, and the Arctic Ocean in MAM,
while this number decreases to above 60% over Europe and the European Arctic
(including water) and is insignificant over North America in JJA. Dust, possibly from
Asian and high-latitude sources, could explain some of the interannual AOD variabilities
over some regions, e.g., Greenland and Greenland Sea in JJA and additionally North
Pacific and the Arctic ocean in MAM, however there exist large uncertainties in this
evaluation based on the worse verification score of CM compared to FM AOD (Tables
2,3,4). And only CAMSRA among the three reanalyses considers high-latitude dust. Co-
variability of species, e.g., BB smoke, ABF/sulfate, and dust, is discernible due to the
same transport pathways from the mid-latitudes to the Arctic. It is also possible that
these species covary because of artifacts introduced by intrinsic treatment in AOD data
assimilation for low AOD situations (Zhang et al., 2008).

793 5.3 Total and speciated AOD trends over 2003-2019
795 The total AOD trends for spring and summer over 2003-2019 derived from MODIS,
796 MISR, and for 2006-2019 from CALIOP are presented in Fig. 9. Because of the scarcity
797 of valid retrievals over the Arctic, the valid trend analysis is mostly limited to south of
798 70°N, and the north Atlantic region, and with less coverage in MAM than in JJA from
799 MODIS and MISR and less coverage in JJA than MAM from CALIOP for reasons
800 mentioned in Sect. 5.1.1.
801
802 5.3.1 AOD trends for springtime
803 For MAM, there is a general negative trend in total AOD over the 50-60°N belt and the
804 North Atlantic, with the largest negative trend of -0.06 to -0.10 AOD/decade being over
805 Europe, most probably due to a decrease in ABF/sulfate from decreased anthropogenic
806 emissions as indicated by the reanalyses (Fig. 10). The negative trend from CALIOP is
807 slightly smaller than those from MODIS and MISR, again possibly attributed to a shorter
808 length of the data record, where earlier and more polluted years for Europe and North
809 America (2003-2006) is not included. All the reanalyses also show a negative trend in
810 total AOD pan-Arctic (-0.01 to -0.02 AOD/decade), except for a close-to-neutral trend
811 over the Arctic ocean and a very slight positive trend over boreal North America from
812 CAMSRA. All the reanalyses suggest that the negative trend over the southeast Siberia
813 and East Asian outflow region is associated with a decrease in BB smoke, and a
814 decrease in ABF/sulfate from NAAPS-RA and MERRA-2 in tandem. Other consistent
815 features found across the reanalyses include the negative trend over Europe associated
816 with decreasing ABF/sulfate, which is possibly related to anthropogenic emission
817 decrease over the past two decades (Breider et al., 2017), as well as a weak positive
818 trend of sea salt over the North Atlantic, which is possibly due to the observed increase
819 in cyclonic activities there (Rinke et al., 2017; Waseda et al., 2021; Valkonen et al.,
820 2021). It is worth noting that NAAPS-RA does not include emission trend for ABF, and
821 MERRA-2 doesn't either after 2008, which means the ABF/sulfate trends seen from
822 these two reanalyses are mostly driven by a negative AOD correction applied by the
823 data assimilation systems. This corroborates the negative trend in ABF/sulfate.

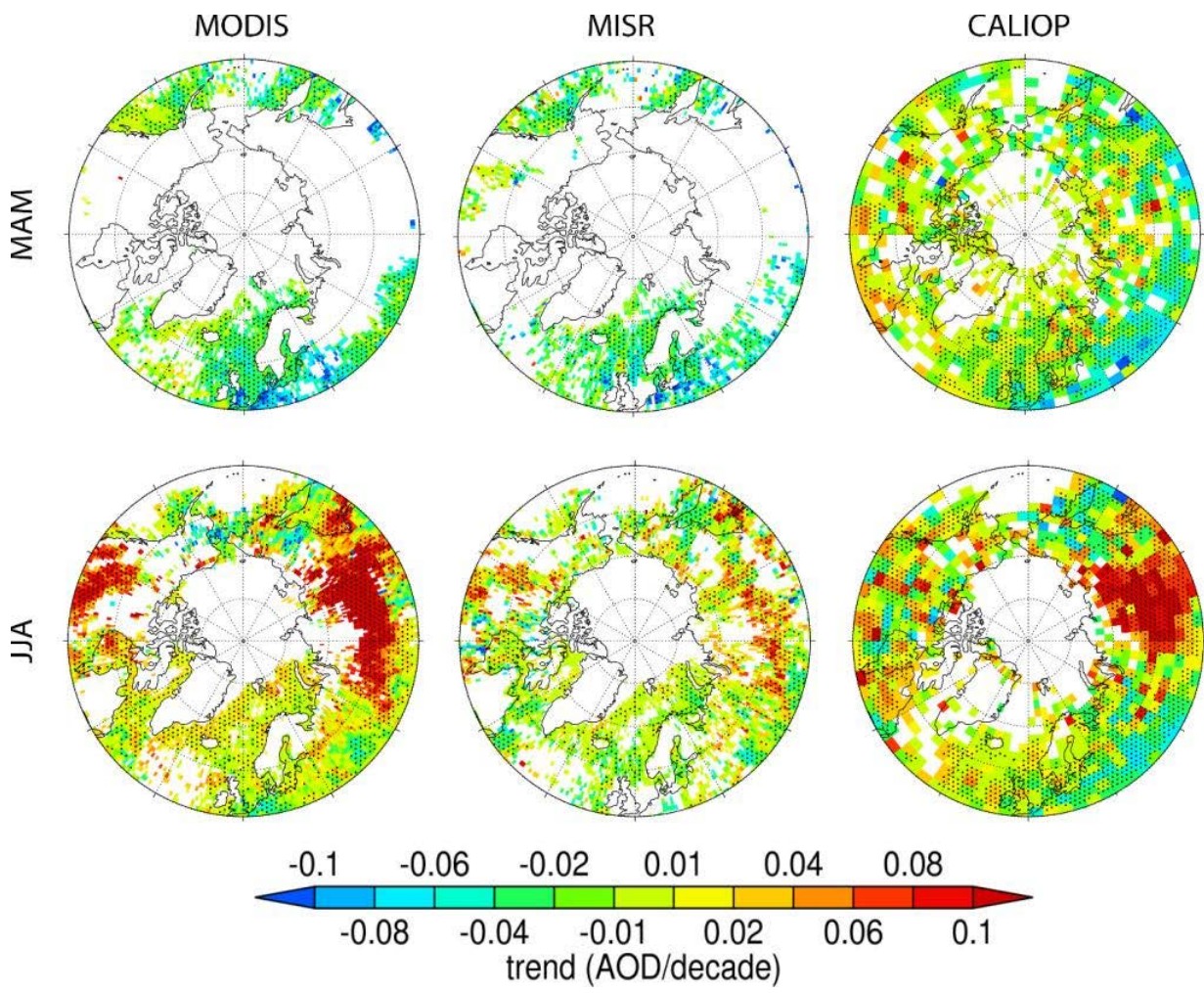

824
**Figure 9.** MAM and JJA AOD trends from MODIS, MISR, and CALIOP for the
corresponding time periods and AOD wavelengths shown in Figure 3. The trend in the
dotted area is statistically significant.

5.3.2 AOD trends for summertime
For JJA, the most prominent feature across all three space-borne sensors is the strong
positive trend of total AOD over vast regions of Siberia and North America with a
magnitude of around or greater than 0.10 AOD/decade. All the reanalyses capture this
positive trend and indicate it is attributed to a significant increase in BB smoke AOD in
these regions over 2003-2019 (Fig. 11). This is in accordance with strong positive
regional trends in BB emissions north of 50°N and north of 60°N derived from FLAMBE,
a MODIS-fire hotspot-based emission inventory (Fig. 12), and from other BB emission
inventories, e.g., GFED and GFAS (Fig. 2 in McMarty et al., 2021). At the same time,
there are negative trends in total AOD over Alaska, northeast of Russia, and North
Pacific from the reanalyses, which is seemingly consistent with the trend in remote
sensing AODs (though for some satellite datasets the coverage is spotty in these
regions). These trends are driven by BB smoke and smoke emission trends as
suggested by all the reanalyses and FLAMBE. In addition, there is a continued negative
trend from MAM to JJA in ABF/sulfate over Europe, which is also reflected in total AOD
trend, as shown in the reanalyses. This is consistent with the discernible negative
though weak trend from the three sensors. JJA AOD trends in dust and sea salt are
neutral from the reanalyses.
Besides rising surface temperature, climate phenomena such as the El Niño–Southern
Oscillation (ENSO), Arctic Oscillation (AO), and Pacific Decadal Oscillation (PDO) have
been reported as affecting fire activity in several key boreal fire source regions (Balzter
et al., 2007; Macias Fauria and Johnson, 2007; Kim et al., 2020). However rising
surface temperature, probably contributes more to the observed trend in BB emission in
the high latitudes. With the rising surface temperature, lightning activity and lightning-
caused wildfires in summertime high latitude regions were observed to increase in the
past two decades (Zhang et al., 2021; Bieniek et al, 2020; Coogan et al., 2020). In
addition, agricultural fire activity in Eastern Europe and European Russia (peaking at
April to May) and central Asia and Asiatic Russian (peaking in August) (Korontzi et al,
2006; Hall et al., 2016) also affects the seasonality of total BB emissions. The MAM
negative trend in BB smoke may be relevant to a strengthening of agriculture burning
regulations in the later part of the 2003-2019 time period. For example, the MAM BB
emission maxima in 2003, 2006 and 2008 are all associated with wide-spread
springtime agriculture burnings in high latitudes (Korontzi et al, 2006; Stohl et al., 2007;
Saha et al., 2010). The aforementioned climate oscillations also modulate interannual
variations of the transport of pollutants from the mid latitudes to the Arctic (e.g.,
Eckhardt et al., 2003; Fisher et al., 2010).
5.3.3 High Arctic AOD trends
For the high Arctic (>70°N), AOD trends are hardly seen with the same color scale as
those for the lower latitudes because of lower AOD. Thus, they are shown separately in
Fig. 13, where time series of MAM and JJA area-mean total, smoke, and ABF/sulfate
AODs are shown individually and for all the reanalyses and the MRC over the 2003-
2019 time period. There is a negative trend across models in MAM total AOD with -
0.017 AOD/decade (-18%/decade), and a positive trend in JJA total AOD with 0.007
AOD/decade (8%decade) based on the MRC. The largest contributor to the MAM
negative trend is ABF/sulfate, and the smoke AOD trend is also negative. In the
summertime, ABF/sulfate trend continues to be negative; however, the smoke AOD
trend turns positive, with a high positive trend of 0.010 AOD/decade (22%/decade). BC
AOD trends from MERRA-2 and CAMSRA are dominantly driven by smoke AOD, and
have similar trends with smoke AOD in percentage per decade. The negative trend in
ABF/sulfate AOD is in line with the decreasing trend in surface sulfate mass
concentrations measured over Arctic observational sites (e.g., Breider et al., 2017). The
negative trend in MAM and positive trend in JJA for smoke AOD are consistent with the
seasonal-and-area-mean BB emission trends shown in Fig. 12 (e,f). The magnitudes of
the trends among the three aerosol reanalyses are different, but the signs are the same,
corroborating the trend analysis results based on the MRC. These results are consistent
with the trend analysis for lower latitude source regions as shown in Fig. 9-11. All these
results also demonstrate that the Arctic aerosol baseline is changing quickly (Schmale
et al., 2021), and the estimation here could contribute to the understanding and
quantification of this new baseline.

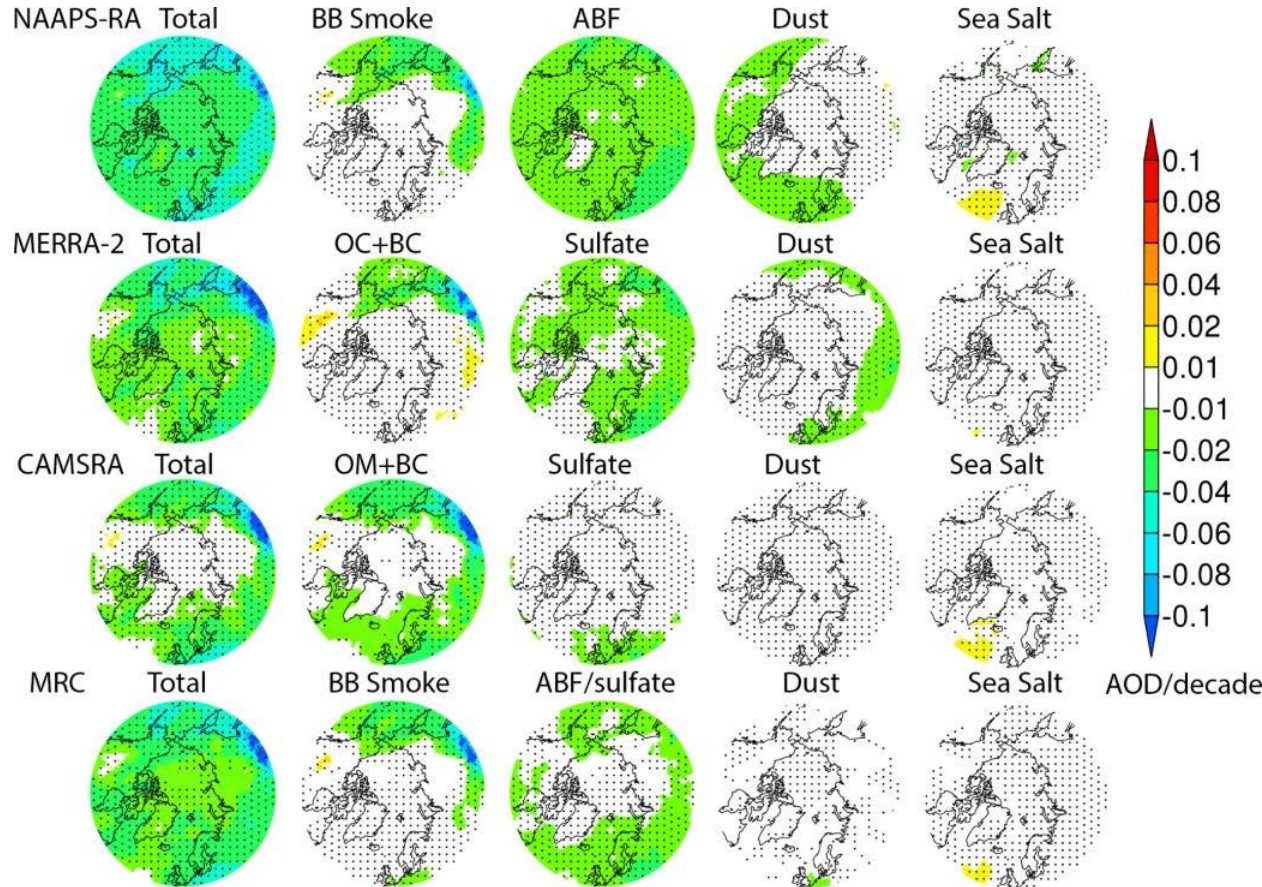

**Figure 10**. Trends of MAM 550 nm total AOD and contributions from biomass-burning
smoke /(BC+OC)/(BC+OM), ABF/Sulfate, dust and sea salt from NAAPS-RA, MERRA-2
and CAMSRA and the MRC.

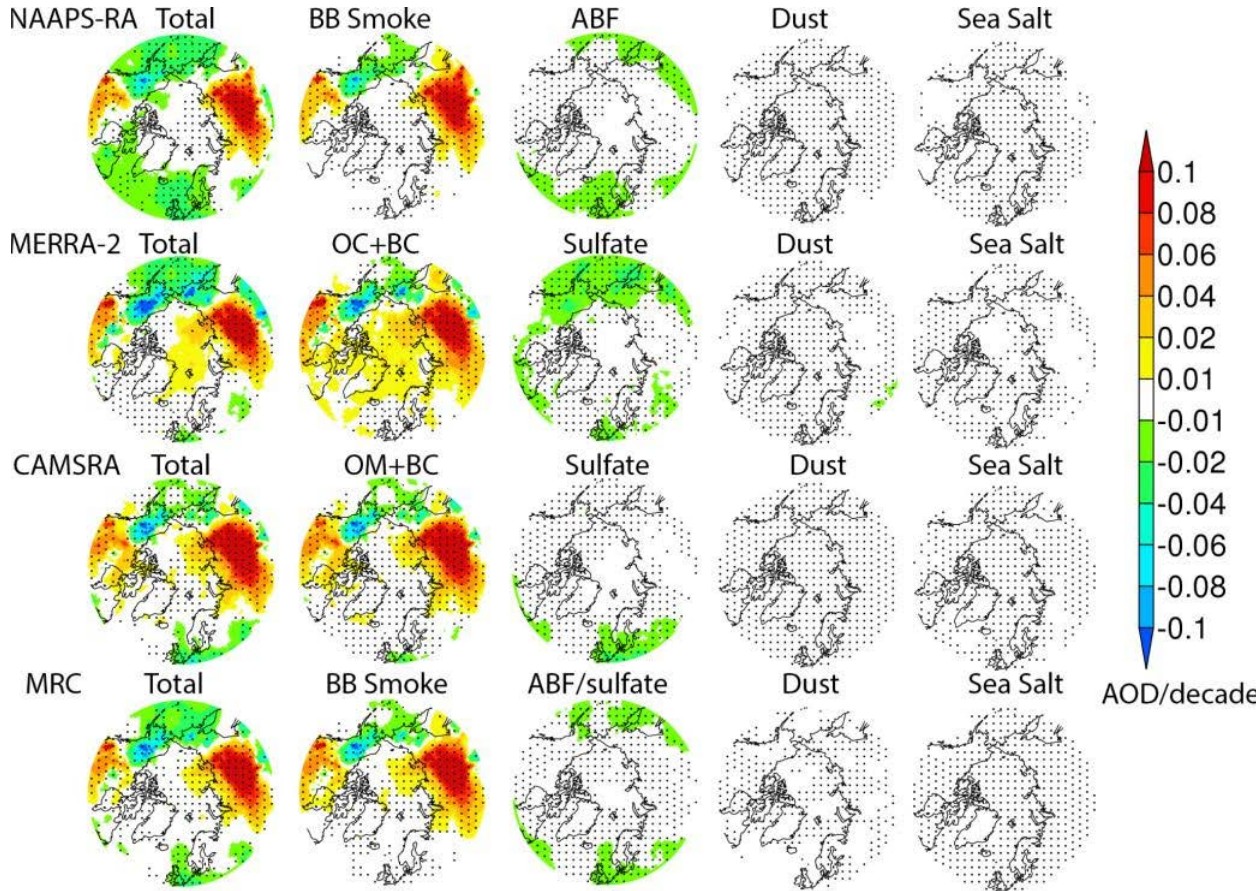

**Figure 11**. Same as Fig. 10, except for JJA.

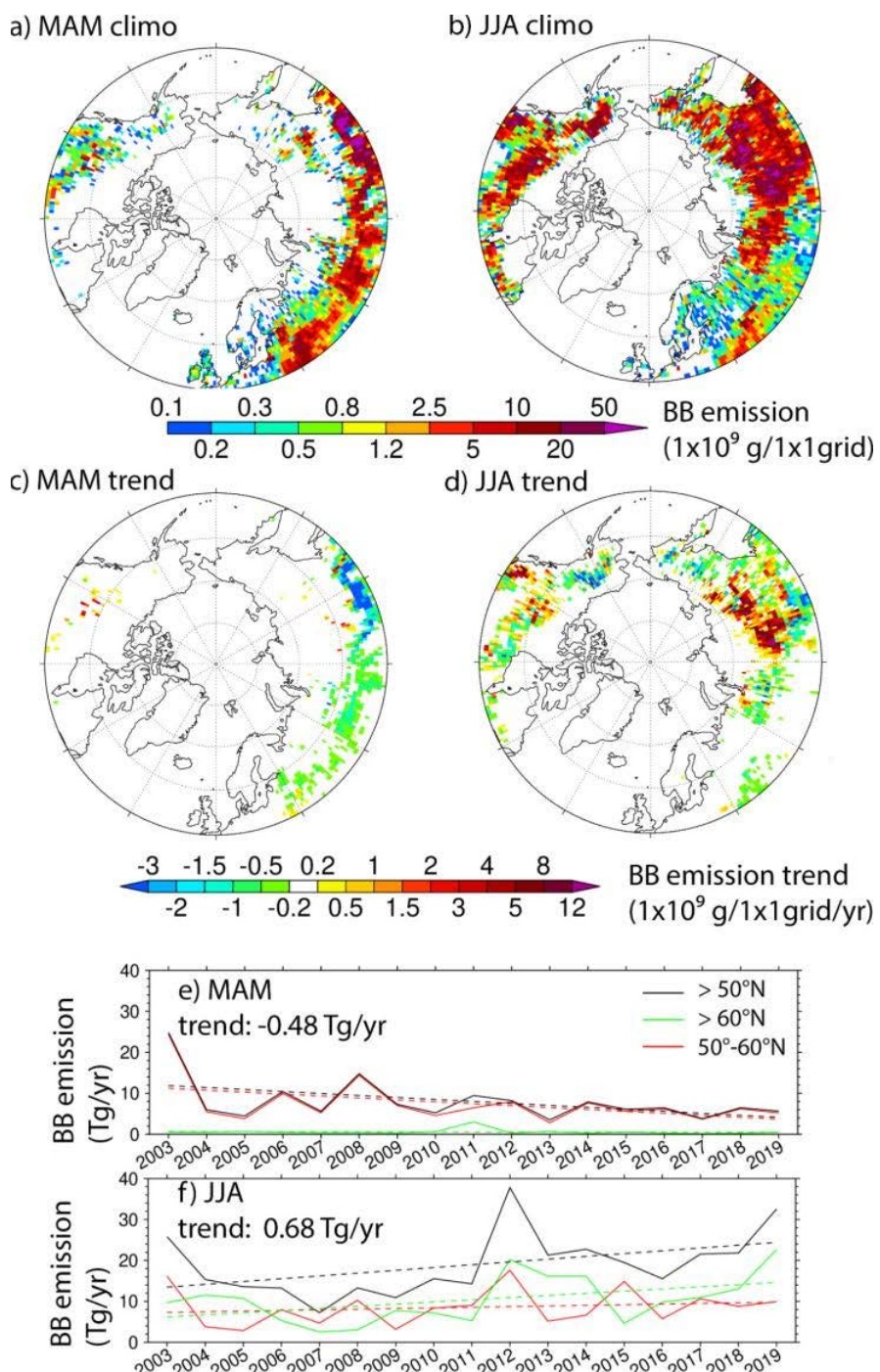

**Figure 12.** MAM/JJA seasonal total BB smoke particle emission climatology and trend
for 2003-2019 derived from FLAMBE (a-d). e) and f) Time series of seasonal-total and
area-mean (>50°N, >60°N and 50-60°N) BB smoke (PM2.5 particle) emissions for MAM
and JJA respectively. Dashed lines represent linear trends, which are statistically
significant with a confidence level of 95%. The trend for north of 50°N is also displayed
in texts.

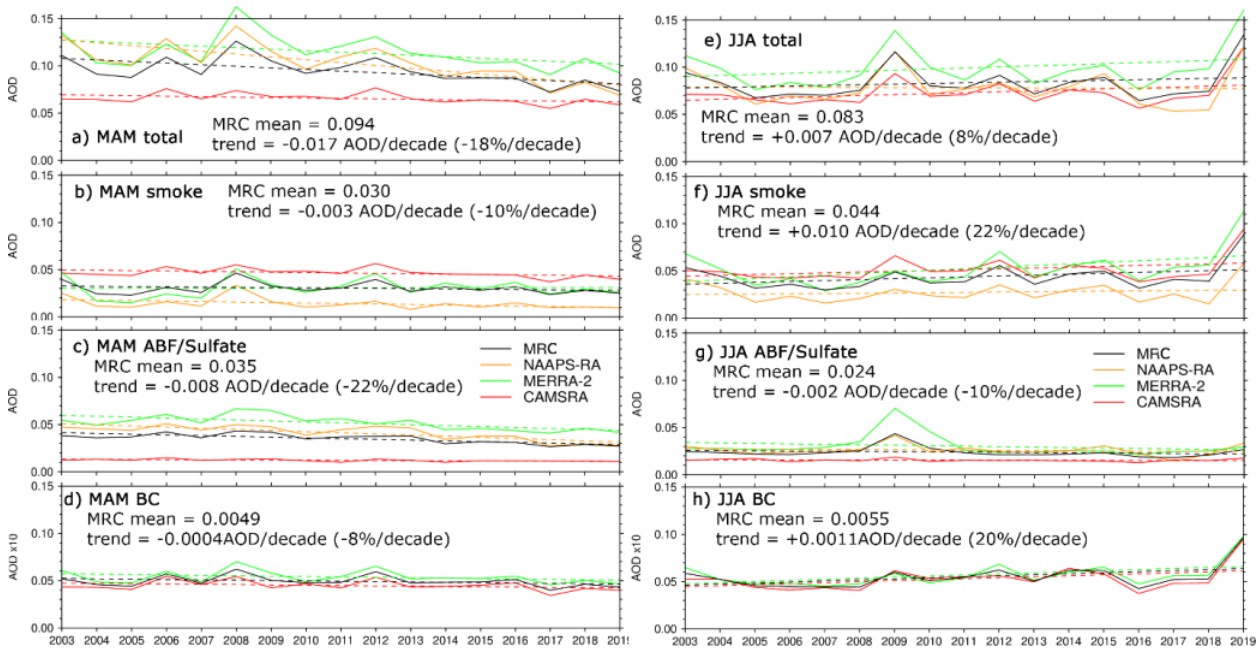


**Figure 13.** Time series of MAM and JJA 70°-90°N area mean total, BB smoke, ABF/sulfate and BC AODs from the reanalyses and the MRC for 2003-2019 time period. Solid lines are AODs, and dashed lines are linear regressions indicating trends. For easier visualization, BC AOD is multiplied by 10.

## 6. Discussion

The quality control processes applied on the AOD retrievals from MODIS, MISR, and CALIOP help to generate a consistent AOD climatology and trend near the Arctic. The cloud-clearing process on the MISR data and QA processes on the MODIS data removed a good volume of data (about 40% for MISR and MODIS). However, these QA processes help to retain only the best-quality data, which yield a closer magnitude of AOD for MODIS and MISR to AERONET AODs near the 70°N latitude circle (around or less than 0.1), compared to ~0.2 using regular level 3 MODIS and MISR data in Figs 20 and 23 of Tomasi et al., 2015, especially for springtime. The manual QA process on the AERONET AOD data also reveals more frequent cloud contamination in springtime than in summertime. Often artificial AOD value of zero are observed over the Arctic in CALIOP V4.2 L2 and L3 data, resulted partially from algorithmically setting altitude bins with retrieval filled values in the aerosol profile to zero, as these represent undetectable levels of faint aerosol (i.e.,Toth et al., 2016; 2018). With AOD=0 values retained in the CALIOP V4.2 L2 data analysis (same processing in CALIOP V4.2 L3), the climatological seasonal mean AOD magnitude is much smaller (about half) than that shown in Fig. 3 and the AOD trends are slightly smaller than those in Fig. 9, although the spatial patterns of the seasonal AOD and trends are similar to those obtained with AOD data after removing the AOD=0 values (Fig. S2). After removing the pixels with filled and

zero values, CALIOP AOD seasonal spatial AOD distributions are similar to those from
MODIS and MISR.
The total AOD at 550 nm from the three aerosol reanalyses are much more convergent
in spatial distribution, magnitude, and seasonality in the Arctic compared to the climate
models, and are similar to those from the remote sensors near the Arctic.  For example,
for AEROCOM models in Sand et al., 2017, MAM AODs averaged over nine Arctic
AERONET sites (all included in this study) are an order of magnitude different for the
highest and lowest AOD models, and peak AOD season varies among winter, spring
and summer; In the CMIP5 models in Glantz et al., 2014, spring and summertime AODs
over the Svalbard area also show an order of magnitude difference and there are
different seasonality for some of the models. The possible reasons for the convergence
of AOD in the reanalyses include 1) the hourly/daily resolved satellite-hotspot-based BB
emissions used by these reanalyses apply fine-temporal and interannual-variability-
resolved emission constraints; 2) despite that the commonly assimilated satellite AOD
(e.g., MODIS AOD in all three reanalyses) has limited coverage in the Arctic due to
retrieval challenges of dealing with bright surfaces and high cloud coverage, the
observational constraint of model fields through assimilation of AOD in the lower
latitudes is effective in constraining Arctic AOD to a good extent through transport; 3)
more accurate meteorology representations. It is reasonable that the AOD spread
among the three reanalyses increases with latitude, and into the early months (e.g.,
March) when retrieval coverage for lower latitudes is less than summer months.
Except for the chemical processes relevant to conversion of $SO_2$ to sulfate, the aerosol
reanalysis products (or their underlying aerosol models) don't include other new particle
formation processes that may be important over the Arctic open water/leads in
Springtime or over packed ice during transitional summer to Autumn season (Abbatt et
al., 2019; Baccarini et al., 2021). High latitude dust sources, e.g., glacier dust, which are
present for some areas in the Arctic (Bullard et al., 2016), are only included in
CAMSRA, despite that Arctic dust AOD in CAMSRA is much lower than those in the
other two models (Fig. 6e).
To show the contribution of biomass burning on total AOD in the Arctic, we
approximated BB smoke with the sum of BC and OC/OA from MERRA-2 and CAMSRA.
This approximation is rather arguable. It is better suited for JJA than MAM, as the
climatological seasonal mean of Arctic AOD is dominated by BB smoke in JJA, which
means that BC and OC/OC are mostly from BB sources, while the contribution of BC
and OC/OA from anthropogenic sources is relatively higher in early spring (Figs. 4, 5).
So smoke AOD is overestimated from MERRA-2 and CAMSRA and more so for MAM.
This explains the larger difference in smoke AOD (ratio to total AOD) in MAM than in
JJA between the two reanalyses and NAAPS-RA, which explicitly tracks aerosol mass
from BB sources (Figs. 4, 5, 6). While NAAPS-RA includes BC and OA from
anthropogenic sources and sulfate into ABF, which is an arguably reasonable
configuration for pollution species, as observational studies show a strong correlation
between sulfate and elemental BC surface concentrations at pan-Arctic sites away from
BB sources, indicating the sources contributing to sulfate and BC are similar and that
the aerosols are internally mixed and undergo similar removal (Eckhardt et al., 2015).
BB smoke is expected to have different vertical distributions from anthropogenic
pollution if smoke is emitted above the boundary layer. Some estimates based on
satellite observations near local noon have suggested that the fraction of smoke
escaping the boundary layer is only ~10% (Val Martin et al., 2010), but taking account of
the diurnal cycle of fire activity and potential for pyroconvection, the actual fraction of
elevated smoke could be much larger (Fromm et al., 2010; Peterson et al., 2015;
Peterson et al., 2017).
Stratospheric aerosols from volcanic eruptions can contribute to the total AOD in the
Arctic, especially for the four years after the Mount Pinatubo eruption in 1991 (Herber
2002). For our study period (2003-2019), the eruptions of Kasatochi, Redoubt,
Sarychev, and Eyjafjallajökull in August 2008, March 2009, July 2009, and March 2010,
respectively, would have affected the stratospheric AOD and thus total column AOD.
However, these eruptions are at least one order of magnitude smaller than that of
Pinatubo. The stratospheric AOD contribution to the Arctic background AOD is
estimated to be relatively small at ~0.01 (from Fig. 16 of Thomason et al., 2018; non-
Pinatubo affected years in Fig. 5 of Herber 2002), despite that locally and over a short
period the AOD contribution can be large (e.g., O'Neill et al., 2012). All the reanalyses
have some sort of $SO_2$ and sulfate representation from volcanic degassing emissions,
but a full representation for explosive volcanic sources is lacking (except that MERRA-2
has time-varying explosive and degassing volcanic $SO_2$ before December 31, 2010).
The volcanic influence on Arctic AOD, if detectable, would be reflected in the
ABF/sulfate AOD in the reanalyses, but its contribution would be much smaller than the
anthropogenic counterpart for our study period. It is also worth noting that volcanic
activities are not the only influence on the stratospheric aerosol budget: pyroCB-injected
BB smoke can also contribute to stratospheric AOD, as discussed earlier. Stratospheric
BB smoke was also detected over the Arctic with lidar measurements during the
MOSAiC campaign (Engelmann et al., 2021). Stratospheric injection of BB smoke
associated with pyroCB events are not represented in the reanalyses, despite that BB
emission associated with these pyroCB events are included in the emission inventories
with possible large bias in emission amount and height.

Arctic shipping is often brought up as a potentially important source of BC for the Arctic
in the future. All of the reanalyses include shipping emissions, although little interannual
trend is considered especially for the late period in 2003-2019. However "Arctic shipping
is currently only a minor source of black carbon emissions overall" according to the
recent Arctic Monitoring and Assessment Programme (AMAP) report (2021).
**7. Conclusions**
Using remote sensing aerosol optical depth (AOD) retrievals from the Moderate
Resolution Imaging Spectroradiometer (MODIS), the Multi-angle Imaging
SpectroRadiometer (MISR), and Cloud-Aerosol Lidar with Orthogonal Polarization
(CALIOP), and AODs from three aerosol reanalyses, including the U.S. Naval Aerosol
Analysis and Prediction System-ReAnalysis (NAAPS-RA),the NASA Modern-Era
Retrospective Analysis for Research and Applications, version 2 (MERRA-2), and the
Copernicus Atmosphere Monitoring Service ReAnalysis (CAMSRA), and ground-based
Aerosol Robotic Network (AERONET) data, we have reported the Arctic/High-Arctic
(defined as 60°-90°N/70°-90°N) AOD climatology, and trend for spring (March-April-
May, MAM) and summer (June-July-August, JJA) seasons during 2003-2019.
1) **Arctic AOD climatology**: The total AODs from space-borne remote sensing and
the aerosol reanalyses show quite consistent climatological spatial patterns and
interannual trends for both spring and summer seasons for the lower-Arctic (60-
70°N), where remote sensing data is available. AOD trends for the high Arctic
from the reanalyses have consistent signs too. Climatologically, fine-mode (FM)
AOD dominates coarse-mode (CM) AOD in the Arctic. Based on the reanalyses,
biomass burning (BB) smoke AOD increases from March to August associated
with seasonality of BB activities in the boreal region (>50°N);
Sulfate/Anthropogenic and biogenic fine (ABF) AOD is slightly higher in MAM
than in JJA; sea salt AOD is highest in March and decreases with time into later
spring and summer; contribution of dust AOD to total AOD is non-negligible in
April and May. The latitudinal gradient of AOD is larger in JJA than in MAM,
consistent with observed more efficient removal in summertime (Garrett et al.,
2011). Among aerosol species, black carbon (BC) is a very efficient light
absorber, and climate forcing agent (e.g., Bond et al., 2013). We show that over
the Arctic, the contribution of BC AOD from BB source overwhelms
anthropogenic sources in both MAM and JJA, and more so in JJA during 2003-
1050      2019.
2) **Interannual AOD trend**: Total AOD exhibits a general negative trend in the
Arctic in MAM, and strong positive trends in North Americas, Eurasia boreal
regions (except Alaska and northeast Siberia) in JJA. For the high Arctic, the
total AOD trend is -0.017/decade (-18%/decade) for MAM and 0.007/decade
(8%/decade) for JJA based on the multi-reanalysis-consensus (MRC). The total
AOD trends are driven by an overall decrease in sulfate/ABF AOD in both
seasons (-0.008/decade, or -22%/decade for MAM and -0.002/decade or -
10%/decade for JJA), and a negative trend in MAM (-0.003/decade or -
10%/decade) and a strong positive trend in JJA (0.01/decade or 22%/decade)
from biomass burning smoke AOD. The decreasing trend in sulfate in the Arctic
in recent decades is in line with other studies using surface concentration
measurement (e.g., Eckhardt et al., 2015).  The smoke AOD trends are
consistent with MODIS fire-hotspot-based BB emission trends over the boreal
continents.
3) ***Impact of BB smoke on AOD interannual variability***: The interannual
variability of total AOD in the Arctic is substantial and predominantly driven by
fine-mode, and specifically BB smoke AOD in both seasons and more so in JJA
than in MAM. For AERONET sites close to BB emission sources, the difference
in monthly total AOD can be 6-fold for high versus low AOD years. For remote
regions away from BB sources, the interannual variability of total AOD can also
be explained mostly by smoke AOD.
4) ***Overall performance of the aerosol reanalyses***: The aerosol reanalyses yield
much more convergent AOD results than the climate models (e.g., AeroCOM
models in Sand et al., 2017; CMIP5 models in Glantz et al., 2014) and verify with
AERONET to some good extent, which corroborates the climatology and trend
analysis. Speciated AODs appear more diverse than the total AOD among the
three reanalyses, and a little more so for MAM than for JJA. NAAPS-RA and
MERRA-2 total and FM AODs verify better in the Arctic than CAMSRA, which
tends to have a high bias in FM overall. The reanalyses generally perform better
in FM than CM. The three reanalyses exhibit different latitudinal AOD gradients,
especially in summertime, indicating different removal efficiencies. The emerging
capability of assimilating OMI Aerosol Index (AI) to constrain absorptive aerosol
amount, could potentially fill in the observational gaps for aerosol data
assimilation in reanalyses over the Arctic (Zhang et al., 2021). With more
advanced retrieval algorithms on the current space-borne sensors for over
snow/ice, new sensors on future satellites, improvements on the underlying
meteorology and aerosol representations in models, improvements in aerosol
reanalysis are expected.
The results presented here provide a baseline of AOD spatiotemporal distribution,
magnitude, and speciation over the Arctic during spring and summer seasons for the
recent two decades. This will help improve aerosol model evaluations and better
constrain aerosol radiative and potentially indirect forcing calculation to evaluate aerosol
impact in the Arctic amplification. For example, the contribution of reduction in sulfate to
Arctic surface warming in recent decades (e.g., Shindell and Faluvegi, 2009; Breider et
al., 2017) could potentially be better quantified, with the caveat that speciated AOD
have larger uncertainties than total AOD in the reanalyses. The AOD statistics could
also provide background information for field campaign data analysis and future field
campaign planning in a larger climate context. It is also recommended that climate
models should take into account BB emissions besides anthropogenic climate forcers
and BB interannual variabilities and trends in Arctic climate change studies.
Appendix A. Summary of data used in the study

| Products | Data | resolution | time |
|---|---|---|---|
| MODIS (Moderate Resolution Imaging Spectroradiometer) C6.1L3 | 550nm AOD | 1°x1° monthly | 2003-2019 |
| MISR (Multi-angle Imaging SpectroRadiometer) V23 | 558nm AOD | 1°x1°, monthly | 2003-2019 |
| CALIOP (Cloud-Aerosol Lidar with Orthogonal Polarisation) V4.2L2 | 532nm AOD | 2°x5°, monthly | 2006-2019 |
| AERONET (AErosol RObotic NETwork) V2L3 | SDA total, FM, CM AOD at 550nm | 6hrly, monthly | 2003-2019 |
| MAN (Marine Aerosol Network) Level2 | SDA total, FM, CM AOD at 550nm | 6hrly | 2003-2019 |
| MERRA-2 (Modern-Era Retrospective Analysis for Research and Applications, v2) | Total and speciated AOD at 550nm | 0.5°lat x0.63°lon, monthly | 2003-2019 |
| CAMSRA (Copernicus Atmosphere Monitoring Service Reanalysis) | Total and speciated AOD at 550nm | 0.7°x0.7°, monthly | 2003-2019 |
| NAAPS-RA v1 (Navy Aerosol Analysis and Prediction System reanalysis v1) | Total and speciated AOD at 550nm | 1°x1°, 6hrly, monthly | 2003-2019 |
| MRC (Multi-Reanalysis-Consensus) | Total and speciated AOD at 550nm | 1°x1°, monthly | 2003-2019 |
| FLAMBE (Fire Locating and Modeling of Burning Emissions) v1.0 | BB smoke emission flux | 1°x1°, monthly | 2003-2019 |

Note: These are final form of data used in the result section. Some pre-processing and
quality-control were applied to remote sensing data as described in the data section.
**Code and Data Availability:** All data supporting the conclusions of this manuscript are
available either through the links provided below or upon request.
AERONET Version 3 Level 2 data: http://aeronet.gsfc.nasa.gov
MAN data: https://aeronet.gsfc.nasa.gov/new_web/maritime_aerosol_network.html
MODIS DA-quality AOD: https://nrlgodae1.nrlmry.navy.mil/cgi-
bin/datalist.pl?dset=nrl_modis_l3&summary=Go
Or https://modaps.modaps.eosdis.nasa.gov/services/about/products/c61-nrt/MCDAODHD.html
MISR AOD: ftp://l5ftl01.larc.nasa.gov/misrl2l3/MISR/MIL2ASAE.003/
CALIOP from NASA Langley Research Center Atmospheric Science Data Center:
https://doi.org/10.5067/CALIOP/CALIPSO/LID_L2_05kmAPro-Standard-V4-20 for the Version
4.2 CALIPSO Level 2 5 km aerosol profile and
https://doi.org/10.5067/CALIOP/CALIPSO/LID_L2_05kmALay-Standard-V4-20 for aerosol layer
products. Further QAed data are available upon request.
NAAPS RA AOD: https://usgodae.org//cgi-
bin/datalist.pl?dset=nrl_naaps_reanalysis&summary=Go
MERRA-2 AOD:
https://disc.gsfc.nasa.gov/datasets/M2TMNXAER_V5.12.4/summary?keywords=%22M
ERRA-2%22
CAMSRA AOD: https://www.ecmwf.int/en/research/climate-reanalysis/cams-reanalysis
FLAMBE BB smoke inventory is available upon request from U.S. NRL.

**Author contributions:** P.X. and J.Z designed this study. P.X. performed most of the
data analysis and wrote the initial manuscript. T.T., B.S. and E.H. helped with
processing of CALIOP, MISR and MODIS AOD data respectively. All authors
contributed to scientific discussion, writing and revision of the manuscript.

**Competing interests:** The authors declare that they have no conflict of interest.

**Acknowledgments**
We thank the NASA AERONET and MAN, and Environment and Climate change
Canada AEROCAN groups for the sun-photometer data, and NASA MODIS, MISR and
CALIOP teams for the AOD data used in the study. We acknowledge NASA GMAO,
ECMWF and U.S. ONR and NRL for making the aerosol reanalysis products available.
We acknowledge the use of imagery from the NASA Worldview application
(https://worldview.earthdata.nasa.gov, last access: Sept 26 2021), part of the NASA
Earth Observing System Data and Information System (EOSDIS).
**Financial support**
The authors acknowledge supports from NASA's Interdisciplinary Science (IDS)
program (grant no. 80NSSC20K1260), NASA's Modeling, Analysis and Prediction
(MAP) program (NNX17AG52G) and the Office of Naval Research Code 322. N.O. and
K.R's work is supported by Canadian Space Agency, SACIA-2 project, Ref. No.
21SUASACOA, ESS-DA program.

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

reanalyses, remote sensing retrievals, and ground observations in a companion paper
(Part 1). In this paper, we report the statistics and trends of Arctic AOD extreme events
using the U.S. Navy Aerosol Analysis and Prediction System ReAnalysis version 1
(NAAPS-RA v1), the sun photometer data from the Aerosol Robotic Network
(AERONET) sites, and the oceanic Maritime Aerosol Network (MAN) measurements.
The AERONET and MAN median for 6 hr total AOD at 550 nm in the Arctic is ~0.06-
0.07 while the 95th percentile value is ~0.23, with a dominant contribution from fine-
mode aerosols. Extreme AOD events are defined as events with AOD exceeding the
95th percentile (denoted "$AOD_{95}$") 6 hr or daily AOD data and histogram distributions
representing specific locations or across a given region (the region north of 70° N for
example). The occurrence and magnitude of extreme AOD events in the Arctic are
largely attributable to biomass burning (BB) smoke events for the North American
Arctic, the Asian Arctic, and most areas of the Arctic Ocean. Regionally, the occurrence
of extreme AOD events is more attributable to anthropogenic and biogenic fine aerosols
in the lower European Arctic. The extreme-event occurrence dominance of sea salt
aerosols is largely limited to the North Atlantic and Norwegian Seas. The extreme AOD
amplitudes of ABF and sea-salt AOD are, however, significantly lower than those
regions where extreme smoke AOD is dominant. Even for sites distant from BB source
regions, BB smoke is the principle driver of AOD variation above the $AOD_{95}$ threshold.
Extreme total AOD Arctic events also show large seasonal and interannual variabilities,
with the interannual AOD variability largely modulated by BB smoke.
There is an overall increase in the maximum AOD values in the high Arctic in 2010-
2019 compared to 2003-2009, indicating stronger extreme BB smoke influence in more
recent years. The occurrence of extreme smoke events tended to be more equally
distributed over all months (April-August) during the 2003-2009 period while being more
concentrated in the late season (July-August) during the 2010-2019 period. The
extreme smoke and total AOD trends resembled the extreme-smoke occurrence trends:
more seasonally balanced during the 2003-2009 period and summertime dominance
during the 2010-2019 period. The temporal shift of the occurrence of AOD extreme
events is associated with the shift in extreme smoke AOD events which is, in turn, likely
due to improved control of early-season agriculture burning, increased summertime
lightning frequencies with climate change in the northern hemisphere high latitudes, and
a reduction in anthropogenic pollution aerosols over the 2010-2019 period. The shift in
extreme smoke events is consistent with a general multi-year decreasing springtime
trend and an increasing summertime trend of BB emissions north of 50° N (Part 1).
1.    Introduction
Warming faster than the rest of the world, the Arctic is a focal point for global warming
(Serreze and Francis 2006; Serreze and Barry 2011). Interactions between the
atmosphere, ocean, land surface, and sea ice, compounded by numerous human
factors make the Arctic climate system challenging to predict, with large diversity
between current numerical model outcomes (IPCC 2013). Arctic aerosol particles from
anthropogenic and natural sources affect regional energy balance through direct
radiative processes and indirect cloud processes (Quinn et al., 2008; Engvall et al.,
2009; Flanner, 2013; Sand et al., 2013; Markowicz et al., 2021; Yang et al., 2018).
When deposited on the surface of snow and ice, light-absorbing aerosol particles,
including dust and black/brown carbon from biomass burning and anthropogenic
emissions, can trigger albedo feedbacks and accelerate melting (Hansen & Nazarenko,
2004; Jacobson, 2004; Flanner et al., 2007; Skiles et al., 2018; Dang et al., 2017; Kang
et al., 2020).
Arctic aerosol concentrations are in general relatively weak, with spring and
summertime median/mean 550 nm aerosol optical depths (AOD) of 0.06 - 0.07 (e.g.,
Tomasi et al., 2007; Saha et al., 2010; AboEl-Fetouh et al., 2020) as compared to a
global mean of roughly 0.20 over land and 0.12 over water (e.g., Levy et al., 2010; Xian
et al., 2016; Shutgers et al., 2020; Sogacheva et al., 2020). Extreme AOD events do
occur within the Arctic, mostly associated with large-scale transport from lower latitudes.
Biomass burning (BB) smoke from boreal wildfires, for example, can episodically result
in record-high Arctic AOD (Myhre et al. 2007; Stohl et al., 2007; Markowicz et al., 2016;
Ranjbar et al., 2019).
Extreme AOD events cause large perturbations in regional energy balance (e.g., Myhre
et al., 2007; Stone et al., 2008; Lisok et al., 2018). For example, a BB smoke transport
event from North America to the High Arctic region of Svalbard in early July 2015 led to
500 nm AOD exceeding 1.2 at Spitsbergen (Markowicz et al., 2016). The two-day mean
aerosol direct radiative forcing was estimated to cause overall cooling (-79 W/m$^2$ at the
surface and -47 W/m$^2$ at the top of the atmosphere TOA). However, a corresponding
atmospheric heating rate profile was solved of up to 1.8 K/day within the BB plume
(Lisok et al., 2018). Over bright snow and ice surfaces, or above clouds, top of the
atmosphere (TOA) BB smoke forcing can turn from negative to positive (i.e., warming)
by reducing columnar albedo (Yoon et al., 2019; Markowicz et al., 2021).
Although the microphysical impacts of aerosol particles on Arctic clouds and
precipitation processes are generally more difficult to measure and quantify, Arctic
clouds are generally believed more sensitive to changes in the relatively low
concentration of aerosols compared with the lower latitudes (Prenni et al., 2007;
Mauritsen et al. 2011; Birch et al., 2012; Coopman et al., 2018; Wex et al., 2019).
Extreme aerosol events correspond with an influx of relatively large concentrations of
potential cloud condensation nuclei (CCN) and/or ice nuclei (IN), in what is otherwise a
comparatively pristine background environment (Mauritsen et al. 2011; Leck et al.,
2015). Such extreme events will accordingly have observable impacts on cloud albedo,
lifetime, phase, and probability of precipitation (e.g., Lance et al., 2011;  Zhao and
Garrett 2015; Zamora et al, 2016; Zamora et al., 2016; Bossioli et al., 2021) and further
influence the regional energy budget. Dry deposition (and blowing snow processes), as
well as wet deposition of BB smoke particles, can also trigger sustained surface
radiative forcing by inducing surface snow discoloration and attendant surface albedo
reduction (Warren and Wiscombe, 1980; Stohl et al., 2007; Hadley and Kirchstetter,
102  2012).

Extreme aerosol events, especially BB smoke events, often modulate the interannual
variability of Arctic AOD (Part 1), as well as to the total annual aerosol budget in the
Arctic. The modeling study by DeRepentigny et al. (2021) shows, in comparison with BB
emissions characterized by a fixed annual cycle, that the inclusion of interannually
varying BB emissions leads to larger Arctic climate variability and enhanced sea-ice
loss. Their finding illustrates the unique sensitivity of climate-relevant processes to
regional aerosol interannual variability, and further suggests that extreme aerosol
events play an important Arctic climate role. It is accordingly important to understand
how extreme aerosol-event statistics change with the changing Arctic climate to better
inform climate simulations and our baseline understanding of how the region is poised
to evolve.
This is the second of two papers examining spring and summertime Arctic AOD
climatologies and their trends. In Part 1 (Xian et al., 2022), we report a baseline Arctic
AOD climatology from AERONET, MAN, and satellite AOD data for those two seasons
and the skill of three reanalysis AOD products in simulating those climatologies. This
paper focuses on reporting statistics and trends of extreme Arctic-AOD events. We
define such events as those corresponding with AOD exceeding the 95th percentile
mark in 6 hr or daily AOD data relative to climatological means at a specific location or
across a given region (the region north of 70°N for example). The data we employ are
described in Sec. 2, while results are provided in Sec. 3. Conclusions are presented in
Sec. 4.
2. Data
We employ 6 hr AERONET AODs as well as speciated daily and 6 hr NAAPS-RA AOD
to depict the frequency and magnitude of the large fine-mode (FM) AOD events. The
companion paper made use of three independent aerosol reanalysis products. For this
study, the NAAPS-RA reanalysis was chosen given its slightly better (AERONET-
referenced; Part 1) performance in terms of FM and total AOD bias, RMSE, and $r^2$
scores, as well as its capability of separating BB smoke from other aerosol species. To
simplify some of the discussion below, we frequently employed the symbol "$AOD_n$" to
represent the AOD associated with the n% percentile of its cumulative (histogram)
distribution.  One important application of this $AOD_n$ formulation was to employ a
particular value ($AOD_{95}$) as a threshold for the definition of extreme events (see Section
3.1 below).
2.1 AERONET
The AErosol RObotic NETwork (AERONET) is a federated ground-based sun
photometer network with over 600 active sites across the globe. AERONET's Cimel
photometers measure sun and sky radiance at several wavelengths, ranging from the
near-ultraviolet to the near-infrared. While the exact set of bands depend on the model,
all Cimel configurations include 440, 670, 870 and 1020 nm bands. All the sites used
here also included 380 and 500 nm bands. The network has been providing high-
accuracy daytime measurements of aerosol optical properties since the 1990s (Holben
et al., 1998; Holben et al., 2001). Cloud-screened and quality-assured Version 3 Level 2
AERONET data (Giles et al., 2019), are used in this study.
FM and CM AOD at 550 nm are derived based on the Spectral Deconvolution Method
(SDA) of O'Neill et al. (2003) and averaged over 6 hr time bins. The same ten
AERONET sites employed in Part 1, were selected (Fig. 1) for this study. Those sites
had been chosen based on their regional representativeness as well as the availability
of data records between Jan 2003 and Dec 2019 (our primary study period).
Optically thin clouds, mostly cirrus, occasionally contaminate CM aerosol retrievals in
Level 2, Version 3 AERONET data (Chew et al., 2012; Ranjbar et al., 2022). Data were
manually inspected, and retrievals screened, using Terra and Aqua imagery at visible
wavelengths from NASA Worldview and by comparing 6-hrly NAAPS-RA with
AERONET AODs.  This step is likely an incomplete one, given the likely lesser
sensitivity of MODIS imagers to thin clouds (Marquis et al., 2017).  As such, CM AODs
that deviate by more than the 3-sigma level from the background climatological mean
were also removed (as per AboEl-Fetouh et al., 2020).
2.2 AERONET Marine Aerosol Network AOD Datasets
The Marine Aerosol Network (MAN) is part of the broader AERONET global network: in
this case however, it is limited to AODs collected over open water.  Hand-held
Microtops sun photometers are deployed during research cruises of opportunity
(Smirnov et al., 2009, 2011). Data processing is similar to that of AERONET with
product nomenclature similar to AERONET. Level 2 data acquired above 70°N in the
2003-2019 period are used in this study. FM and CM AOD at 550 nm are derived using
the SDA and averaged over 6 hr time bins.
2.3 NAAPS AOD reanalysis v1
The Navy Aerosol Analysis and Prediction System (NAAPS) AOD reanalysis (NAAPS-
RA) v1 provides 550 nm speciated AOD at a global scale with 1°x1° degree
latitude/longitude and 6 hr resolution for the years 2003-2019 (Lynch et al., 2016). This
reanalysis features assimilation of quality-controlled and quality-assured AOD retrievals
from MODIS and MISR (Zhang et al., 2006; Hyer et al., 2011; Shi et al., 2011). A first-
order approximation of secondary organic aerosol (SOA) processes is adopted.
Production of SOA from its precursors is assumed to be instantaneous and is included
with the original anthropogenic species to form a combined anthropogenic and biogenic
fine (ABF) species (a mixture of sulfate, BC, organic aerosols and secondary organic
aerosols from non-BB sources). Monthly anthropogenic emissions come from a 2000-
2010 average of the ECMWF MACC inventory (e.g., Granier et al., 2011). BB smoke is
derived from Fire Locating and Modeling of Burning Emissions inventory (FLAMBE,
Reid et al., 2009). This version of FLAMBE uses MODIS, near-real-time satellite-based
thermal anomaly data to initialize the smoke source where corrections that minimize the
impact of inter-orbit variations are applied to the MODIS data (Lynch et al., 2016).
FLAMBE processing is applied consistently through the reanalysis time period while a
smoke-particle emission climatology and its spring and summertime trends (both north
of 50°N and 60°N) are provided in Fig. 12 of Part 1. Dust is emitted dynamically and is a
function of modeled friction velocity to the fourth power, surface wetness, and surface
erodibility. In this model run, erodibility is adopted from Ginoux, et al., (2001) with
regional tuning). Sea-salt modeling is the same as Witek et al. (2007) and sea-salt
emission is driven dynamically by sea surface wind. Verification of monthly-binned
NAAPS-RA modal and total AODs at 550 nm using monthly-binned AERONET data
from 10 Arctic sites (Table 1 of Part 1)  shows (coupled with the bias and rmse results of
Figure 2 in Part 1) that NAAPS-RA is able to capture the AOD interannual variability.
The spatial distributions and magnitudes of climatological and seasonal AOD averages
and their trends for 2003-2019 are also consistent with those derived from MODIS,
MISR, and CALIOP (Part 1).
3. Results
Seasonal AOD averages and trends derived from remote sensing measurements for the
2003-2019 sampling period are provided in Part 1. The interannual Arctic AOD
variability is, as discussed in Part I, considerable and driven mostly by FM aerosol
events (notably BB transport events). Regional statistics and trends of extreme AOD
events are presented in this section: 6-hr AERONET AOD as well as speciated daily
and 6-hr NAAPS-RA AOD are employed to characterize the frequency and magnitude
of strong FM AOD events.
3.1 Verification of NAAPS-RA AOD over the Arctic
The reanalysis performance for 6-hr time bins was evaluated in order to study extreme
events. Our choice of $AOD_{95}$ as an extreme event threshold was influenced by the fact
that it was an upper-limit cumulative probability indicator that was robust. We reasoned,
at the same time, that it should be comparable with the analog parameter derived from
NAAPS-RA. Figure 1 displays NAAPS-RA $AOD_{95}$ overplotted with those from the ten
selected AERONET sites for spring and summertime 2003-2019. NAAPS-RA appears
to successfully capture the $AOD_{95}$ amplitude and spatial pattern, as well as those of FM
$AOD_{95}$ and CM $AOD_{95}$. It also shows that FM is the main contributor to $AOD_{95}$ in the
Arctic.

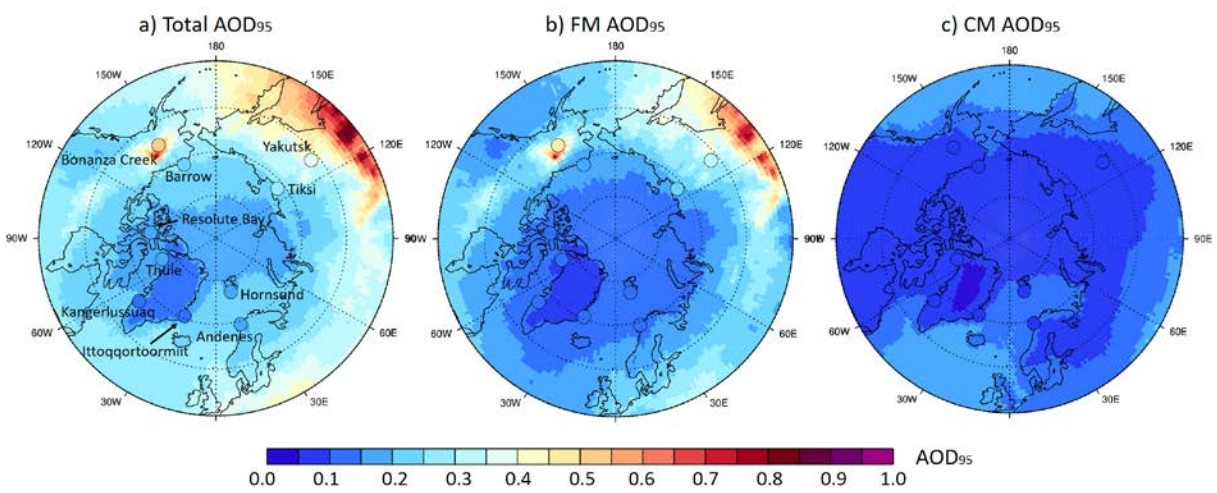


**Figure 1.** Total, FM and CM AOD at the 95th percentile ($AOD_{95}$) for the March-August
time frame from the NAAPS-RA and the ten AERONET sites (based on 6hrly data
between 2003-2019).
Table 1 provides detailed geographical coordinates of the ten AERONET sites
employed in our study, as well as the simulation performance indicators of NAAPS-RA
550 nm total, FM and CM AOD. These AERONET parameters are an analogue to Part
1 and its Table 1 statistics, except that the averaging period extends across both the
spring and summer seasons (meaning in practice that the averaging period is mostly
confined to the April-August time frame). Scatter plots of NAAPS-RA vs AERONET
AODs at all sites north of 60°N are shown in Fig. 2. NAAPS-RA performance indicators
relative to MAN data are shown in Fig. S1 and S2.
NAAPS-RA performance for this large averaging period is reasonable for FM and total
AOD, though it is less skillful at predicting CM AOD. The FM AOD exhibits an average
(Table 1) bias over all stations of -0.01, a root mean square error (RMSE) of 0.08 and a
coefficient of determination ($r^2$) of 0.66. RMSE values for total and FM AOD are
generally large for sites vulnerable to strong smoke influence (Bonanza Creek, Barrow,
Tiksi and Yakutsk). Total AOD $r^2$ values are mostly between 0.5-0.7, except for
Hornsund, Kangerlussuaq and Ittoqqortoormiit. FM AOD $r^2$ values exceed those of the
total AOD for all sites except Kangerlussuaq. The monthly-binned Table 1 total AOD
bias (where the Table 1 averaging over the spring and summer is the simple average of
the spring and summer averages) is similar to the monthly-binned NAAPS-RA bias
results of Table 2, Part 1). This is due to the numerous (6-hr) samples included in the
(signed) AERONET bias averaging.  In contrast, the Table 1 RMSE values are roughly
doubled, and the $r^2$ values drop by about 30% relative to those of Tables 3 and 4 of Part
1. This suggests Table 1 model shortcomings in capturing finer temporal-scale (higher
frequency) AERONET-AOD variations. This is also consistent with model performance
for regions other than the Arctic, and is generally a common result for numerical aerosol
models (Lynch et al., 2016; Yumimoto et al., 2017)
The lesser CM vs FM skill of the NAAPS-RA might be a reflection of AERONET
limitations as one approaches typical instrumental errors ~ 0.01 in total AOD or they
could be a reflection of simulation and / or reanalysis limitations as one approaches very
small values of CM AOD. The lack of model representation of CM smoke and possible
soil particles associated with severe burning events may also contribute. At the same
time, it must be recognized that residual cloud contamination in AERONET (and MAN)
data cannot be ruled out as a "false" indicator of poor simulation skill. Cloud screening
issues aside, a lesser CM vs FM correlation skill is a common feature of both the Table
1 and Table 4 (Part 1) reanalyses. However, modeled monthly CM AOD correlation is
slightly more skillful than the averages derived from 6 hr data (Table 4 in Part 1 vs Table
1) inasmuch as the seasonal CM signal associated with dust and sea salt aerosols are
apparently better resolved in the former case (likely due to the relative insensitivity
of  the model to the higher frequency components of the reference data in the latter
case). It is also noted that the NAAPS-RA is generally less skillful in the Arctic region
relative to global reanalyses (c.f  Fig. 7 in Lynch et al., 2016). This is understandable
given that (compared with lower latitudes) there is little satellite-based Arctic-AOD data
available to constrain the model through assimilation. We note however that Zhang et
al. (2021) attempted to address this problem with assimilation of Ozone Monitoring
Instrument (OMI) Aerosol Index. To date, no remedy has yet been implemented in a
larger RA-quality study.
**Table 1.** Geographical coordinates along with the total, FM and CM AOD statistics (2003-2019
depending on availability) for AERONET and 6-hrly NAAPS-RA 550 nm performance indicators
versus AERONET (mean bias, root mean square error (rmse) and coefficient of determination
(r²))  The last row shows the same statistics for MAN AODs acquired north of 70°N as the bias
reference. These numbers are given as information: as indicated above the table statistics in Part
1 were explicitly computed using monthly binned data (which were, in turn, derived from the 6 hr
data).

| sites | latitude | longitude | elevation (m) | region | AERONET mean total \| FM \| CM | total \| FM \| CM AOD Bias | rmse | r² | n |
|---|---|---|---|---|---|---|---|---|---|
| Hornsund | 77.0°N | 15.6°E | 12 | Svalbard | 0.09\|0.06\|0.03 | -0.01\|-0.02\|0.01 | 0.04\|0.04\|0.03 | 0.55\|0.62\|0.06 | 1,817 |
| Thule | 76.5°N | 68.8°W | 225 | Greenland | 0.07\|0.06\|0.02 | 0.00\|-0.01\|0.01 | 0.04\|0.03\|0.03 | 0.52\|0.60\|0.07 | 2,518 |
| Kangerlussuaq | 67.0°N | 50.6°W | 320 | Greenland | 0.07\|0.05\|0.02 | 0.02\|0.00\|0.01 | 0.05\|0.04\|0.03 | 0.32\|0.30\|0.03 | 2,725 |
| Ittoqqortoormiit | 70.5°N | 21.0°W | 68 | Greenland | 0.06\|0.05\|0.02 | 0.01\|-0.00\|0.01 | 0.04\|0.03\|0.03 | 0.41\|0.49\|0.04 | 1,825 |
| Andenes | 69.3°N | 16.0°E | 379 | Norway | 0.08\|0.05\|0.02 | 0.01\|-0.01\|0.01 | 0.04\|0.03\|0.03 | 0.54\|0.56\|0.16 | 1,829 |
| Resolute_Bay | 74.7°N | 94.9°W | 35 | Nunavut | 0.08\|0.05\|0.02 | 0.01\|-0.01\|0.01 | 0.06\|0.05\|0.03 | 0.55\|0.62\|0.02 | 1,698 |
| Barrow | 71.3°N | 156.7°W | 8 | Alaska | 0.10\|0.08\|0.02 | -0.00\|-0.02\|0.01 | 0.09\|0.08\|0.04 | 0.53\|0.61\|0.07 | 1,760 |
| Bonanza_Creek | 64.7°N | 148.3°W | 353 | Alaska | 0.16\|0.12\|0.03 | -0.02\|-0.02\|-0.00 | 0.16\|0.15\|0.04 | 0.69\|0.70\|0.07 | 2,670 |
| Tiksi | 71.6°N | 129.0°E | 17 | Siberia | 0.12\|0.10\|0.02 | -0.01\|-0.02\|0.01 | 0.09\|0.08\|0.03 | 0.69\|0.73\|0.01 | 488 |
| Yakutsk | 61.7°N | 129.4°E | 119 | Siberia | 0.16\|0.12\|0.03 | -0.01\|-0.02\|0.01 | 0.13\|0.12\|0.04 | 0.61\|0.62\|0.15 | 4,095 |
| MAN | >70°N | - | - | Arctic Ocean | 0.07\|0.05\|0.02 | -0.00\|-0.01\|0.00 | 0.04\|0.03\|0.02 | 0.51\|0.32\|0.07 | 520 |


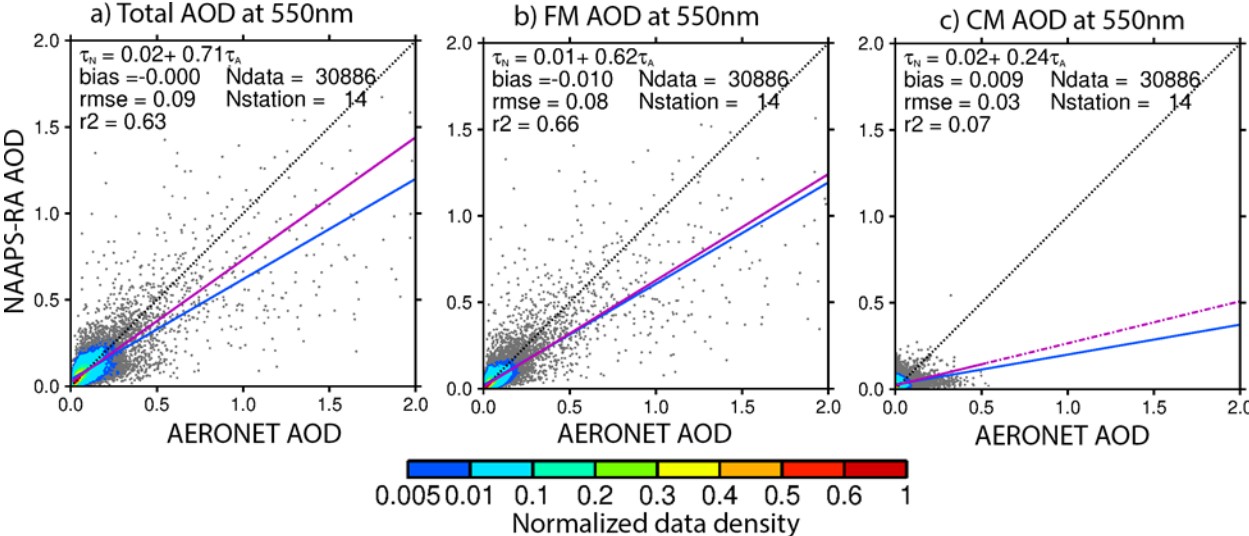


**Figure 2.** Pairwise comparison of the NAAPS-RA 6-hr AOD and AERONET AOD with
respect to total (left), fine (middle) and coarse (right) modes at 550 nm for all sites north
of 60° N for 2003–2019. The normalized data density is shown in color. The solid
magenta line represents a Theil–Sen linear regression: the corresponding linear
equation and bias statistics are shown in the top left hand corner of each graph (where
$\tau_N$ and $\tau_A$ are the NAAPS-RA and AERONET AODs respectively. The solid blue line is a
least-squares linear regression (corresponding equation is not shown). Also shown are
the bias, root mean square error (rmse), coefficient of determination (r²), total number of
stations (Nstation) and total number of 6-hr AERONET data (Ndata). AODs greater than
2.0 are not shown but were incorporated in the statistics calculations.
3.2 General statistics of extreme events
Fig. 1 shows NAAPS-RA and AERONET $AOD_{95}$ values for the March-August time
frame and the 2003-2019 period (see also Table 2). The values of $AOD_{95}$ are high
(0.4~0.55) over Siberia and Alaska (and over the Yakutsk and Bonanza Creek
AERONET stations) due to strong BB smoke influence. North of 70°N, the values are
mostly between 0.15 to 0.25, with the exception of Greenland where they are largely
below 0.15 (weak values that are attributable to the high terrain). It is also shown that
$(FM\ AOD)_{95}$ has similar spatial distribution and magnitude as $AOD_{95}$, suggesting the
dominant contribution of FM to $AOD_{95}$. Contribution of CM is relatively larger over the
North Atlantic and European Arctic, though $(CM\ AOD)_{95}$ and $(FM\ AOD)_{95}$ are
comparable in these regions.
Fig. 3 shows the site-by-site, total, and FM AOD ranges from the 6-hr AERONET data
for all 550 nm retrievals (in black) acquired between 2003-2019. The 6-hr pairwise
NAAPS-RA AOD ranges (in red) facilitate model skill evaluation (see the caption of Fig.
3 for "pairwise" details). In general, the NAAPS-RA largely captures the AERONET FM
and total AOD range. This includes, for example, the AERONET $AOD_5$ to $AOD_{95}$ values
(~ 0.02 to >~ 0.10 for most sites), and the larger 0.02 to ~ 0.4-0.6 range of sites with
known strong BB influence (notably Bonanza Creek, Tiksi, and Yakutsk). Mean and
median AODs are also comparable to AERONET values. Maximum AERONET FM
AODs vary between ~ 0.5 (Ittoqqortoormiit) to < 2.0 for most sites and around 3.0 for
sites with strong BB smoke influence (see also Table 2). Maximum NAAPS-RA AOD
values are often biased low, which is a common challenge for global aerosol models
(e.g. Sessions et al., 2015; Xian et al., 2019).

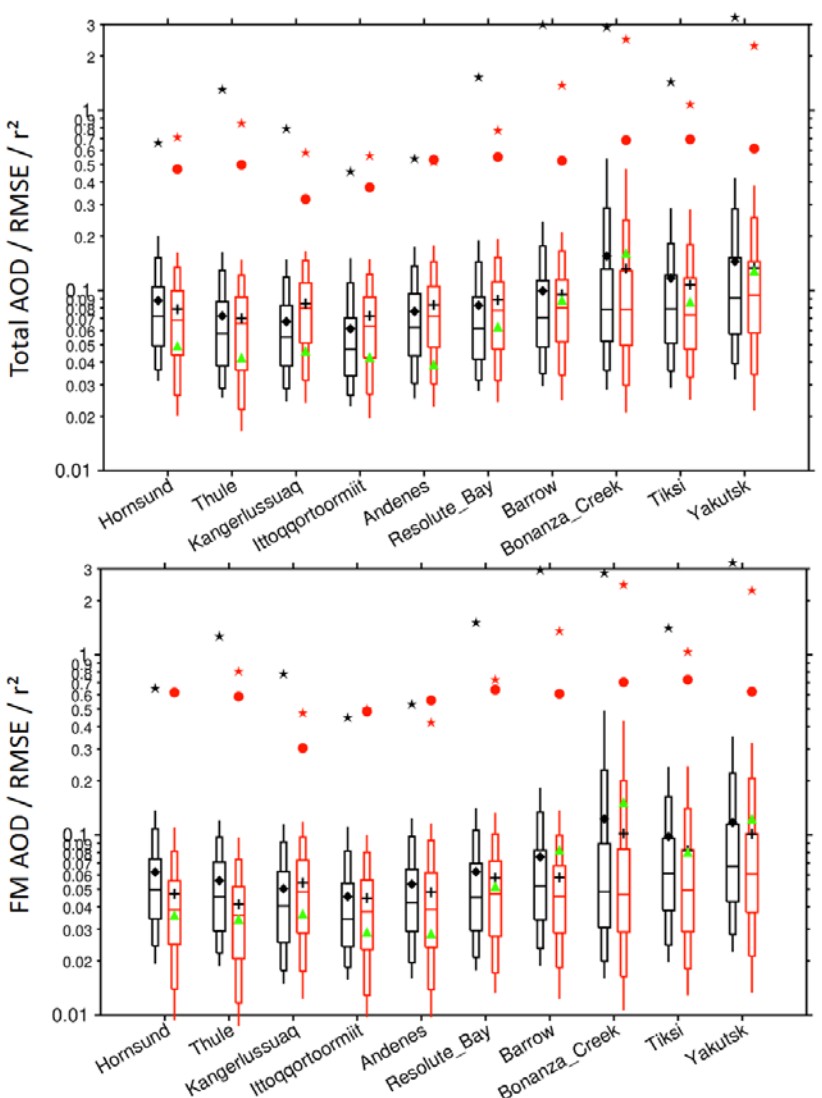


**Figure 3.** Comparison of the 6-hrly (550 nm) total (top) and FM AOD (bottom) of the NAAPS-RA (red) at 95, 90, 75, 50, 25, 10, and 5% percentiles (respective, sequential features of the doubled spear-like symbols from the top tip to the bottom tip) with pairwise AERONET V3L2 data (black) for the ten AERONET sites of Table 1 and Figure 1 for the 2003-2019 time period ("pairwise" refers to those NAAPS-RA AODs that correspond to a resampled AERONET AOD whose $\pm$ 3hr bin contains at least one AERONET retrieval). Also shown are the site means of the NAAPS-RA and AERONET AODs ("+" and "♦" symbols respectively) and the NAAPS-RA RMSE ("▲"), the coefficient of determination ($r^2$) between the NAAPS-RA and AERONET ("●") and the maximum AERONET and NAAPS-RA AODs ("★" and "★" respectively). Note that values greater than 3.0 are not shown.



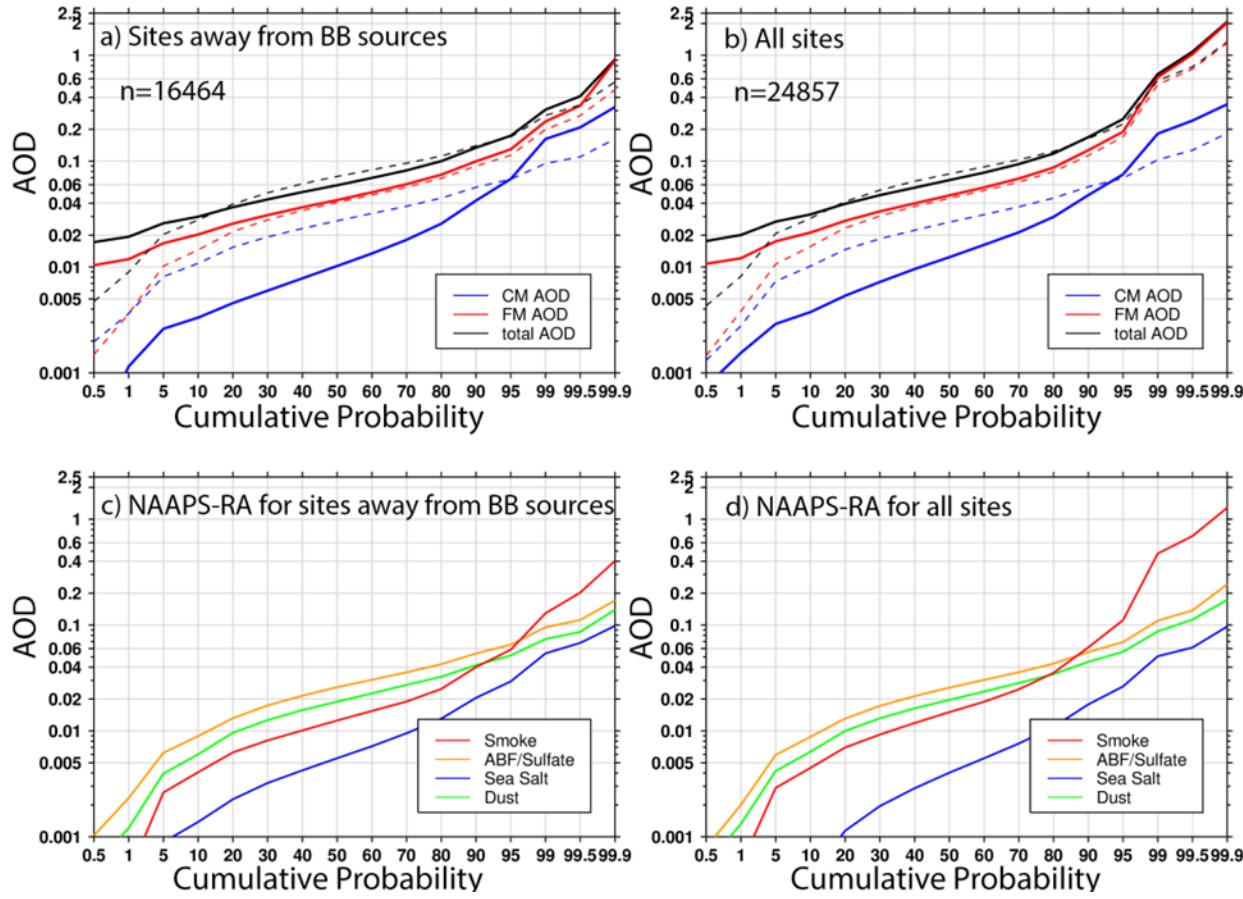

**Figure 4.** Upper panes (a, b): cumulative probability distributions of 2003-2019, 6-hr
total, FM and CM AOD at 550 nm for AERONET V3 L2 data (solid curves) and pair-wise
NAAPS-RA (dashed curves). Lower panes (c,d): cumulative probability distributions for
the corresponding speciated AOD from the NAAPS-RA. Left hand panes (a,c): AOD for
sites that are distant from BB source regions, including Barrow, Resolute Bay,
Kangerlussuaq, Thule, Andenes, Hornsund and Ittoqqootoormiit (see the discussion of
Table 2 for emission considerations with respect to the particular site of Barrow). Right-
hand panels (b,d) all sites north of 60°N. "n" represents the total number of 6-hrly data
points over the 2003-2019 period.
Figure 4 shows the cumulative probability distribution of 6-hr total, FM and CM AODs for
AERONET and pair-wise NAAPS-RA total and modal AODs (Figures 3a and b) and
speciated AODs (Figures 3c and d). The median (50%) AOD for all AERONET sites in
the Arctic (all sites north of 60° N) for 2003-2019 is ~0.06, while the $AOD_{95}$ extreme-
event threshold is ~0.23 with a dominant FM contribution. The CM AOD median for all
measurements is ~0.01, with a $(CM\ AOD)_{95}$ threshold of only ~0.07. NAAPS-RA total
AOD bias is, due to a relatively large positive bias in CM AOD of ~ 0.01 below the 95%
threshold, slightly positive (<0.01) for all sites north of 60° N, and for the 20%-80%
cumulative probability range (a positive bias that is generally evident in Table 1).
It is common for models to bias low for extreme events (e.g. Sessions et al. 2015; Xian
et al., 2019). The negative bias found at the largest CM AOD values could conceivably
be associated with an underestimation of the CM AOD generated by sea-salt aerosols
in the presence of strong winds or CM smoke and soil particles associated with severe
burnings. We should, however, reemphasize this caveat: despite the quality-control
measures taken to filter out cloud-contaminated AERONET data, the impact of CM
residual clouds may still influence estimates of CM AOD.
**Table 2.** AERONET V2L3 FM, CM, and total AOD at 550nm at different percentiles for the listed
Arctic sites along with maximum AOD values in the third last column. "N" represents the total
number of 6-hr AODs for 2003-2019. The percentage of extreme FM events relative to the number
of extreme total AOD events (using our $AOD_{95}$ extreme-event threshold) is also shown in the last
column. The last row shows MAN statistics for data acquired north of 70˚ N.

| | Total \| FM \| CM AOD at 550nm | | | | | | | | FM |
|---|---|---|---|---|---|---|---|---|---|
| | Median | 75% | 90% | 95% | 99% | 99.9% | maximum | N | event |
| Hornsund | 0.072\|0.049\|0.014 | 0.103\|0.074\|0.028 | 0.145\|0.108\|0.048 | 0.184\|0.135\|0.077 | 0.320\|0.300\|0.155 | 0.663\|0.654\|0.222 | 0.663\|0.654\|0.222 | 1975 | 67% |
| Thule | 0.055\|0.043\|0.006 | 0.083\|0.067\|0.014 | 0.121\|0.092\|0.034 | 0.156\|0.116\|0.057 | 0.294\|0.198\|0.164 | 0.914\|0.913\|0.315 | 1.310\|1.272\|0.315 | 2934 | 59% |
| Kangerlussuaq | 0.055\|0.040\|0.009 | 0.082\|0.063\|0.020 | 0.118\|0.091\|0.037 | 0.149\|0.115\|0.059 | 0.234\|0.198\|0.109 | 0.510\|0.461\|0.203 | 0.794\|0.786\|0.222 | 3066 | 75% |
| Ittoqqortoormiit | 0.046\|0.033\|0.006 | 0.069\|0.053\|0.014 | 0.108\|0.083\|0.031 | 0.144\|0.112\|0.054 | 0.238\|0.215\|0.121 | 0.456\|0.446\|0.232 | 0.459\|0.450\|0.233 | 2041 | 73% |
| Andenes | 0.062\|0.042\|0.014 | 0.096\|0.064\|0.027 | 0.136\|0.098\|0.049 | 0.172\|0.123\|0.072 | 0.274\|0.210\|0.148 | 0.451\|0.432\|0.249 | 0.541\|0.534\|0.258 | 2222 | 69% |
| Resolute_Bay | 0.061\|0.045\|0.011 | 0.092\|0.069\|0.021 | 0.143\|0.106\|0.039 | 0.187\|0.140\|0.059 | 0.409\|0.389\|0.152 | 1.530\|1.516\|0.379 | 1.530\|1.516\|0.379 | 1876 | 72% |
| Barrow | 0.071\|0.053\|0.013 | 0.114\|0.082\|0.024 | 0.175\|0.134\|0.047 | 0.232\|0.183\|0.076 | 0.455\|0.415\|0.174 | 2.999\|2.962\|0.328 | 2.999\|2.962\|0.328 | 1920 | 81% |
| Bonanza_Creek | 0.078\|0.048\|0.022 | 0.130\|0.089\|0.036 | 0.280\|0.230\|0.057 | 0.532\|0.497\|0.083 | 1.713\|1.643\|0.186 | 2.619\|2.591\|0.341 | 2.908\|2.857\|0.345 | 3177 | 99% |
| Tiksi | 0.079\|0.061\|0.011 | 0.121\|0.096\|0.021 | 0.182\|0.163\|0.040 | 0.286\|0.239\|0.060 | 0.936\|0.915\|0.123 | 1.442\|1.413\|0.238 | 1.442\|1.413\|0.238 | 631 | 97% |
| Yakutsk | 0.094\|0.069\|0.014 | 0.153\|0.119\|0.027 | 0.272\|0.221\|0.053 | 0.400\|0.345\|0.089 | 0.980\|0.963\|0.201 | 3.018\|2.972\|0.317 | 3.296\|3.259\|0.340 | 4797 | 96% |
| MAN | 0.052\|0.029\|0.021 | 0.090\|0.062\|0.031 | 0.126\|0.097\|0.042 | 0.164\|0.118\|0.052 | 0.281\|0.253\|0.085 | 0.777\|0.761\|0.234 | 0.777\|0.761\|0.234 | 520 | 92% |


BB smoke plays a dominant role compared with other aerosol species above our $AOD_{95}$
extreme-event threshold (see Fig. 4b in particular and note that Fig. 4d shows the
expected dominance of FM AOD). Even for sites distant from BB source regions
(including Resolute Bay, Kangerlussuaq, Thule, Andenes, Hornsund, Ittoqqortoormiit)
BB smoke is the principal driver of AOD variations above the $AOD_{95}$ threshold (see Fig.
4c in particular, supported by FM AOD domination in Fig. 4a). To some extent, Barrow
can be categorized as being a site that is distant from BB emissions. However, it is also
relatively close to the region of Alaska fires, depending on dominant upstream winds
and trajectories (see Eck et al., 2009 for details).
The modal and total AOD values at different percentile levels for the AERONET sites
and MAN data collected north of 70° N are provided in Table 2. For sites closer to BB
sources, including Bonanza Creek, Yakutsk, and Tiksi, the $AOD_{99}$ and (FM AOD)$_{99}$
values are >~ 1.0 while the maximum values are between 1.4-3.3. For the more distant
sites, the $AOD_{99}$ and (FM AOD)$_{99}$ values vary between 0.23-0.46 while the maximum
values are between 0.45-3.0 (1.5 for Resolute Bay and 3.0 for Barrow). FM event
occurrences for the extreme total AOD events, range from 60-99% and accordingly
dominate CM events statistically. Sites closer to the BB source regions show relative
occurrences over 95%.
Large particles like ash and soil components emitted from vigorous burning during
extreme BB smoke events (Reid et al., 2005; Schlosser et al., 2017) can likely be
detected as AERONET CM AOD (see, for example, the correlation between the FM and
"weak" CM particle size distributions for Bonanza Creek in Fig. 9a of Eck et al. [2009]).
The extreme AOD events described above ((FM AOD)$_{99}$ of 1.643 at Bonanza Creek and
0.936 at Tiksi in Table 2, for example) are likely dominated by smoke. The associated
CM AOD means for those two cases showed significantly larger values of 0.049 and
0.033, respectively (significantly larger relative to, for example, the CM AOD means in
Table 1). The coherency of the associated CM AOD mean increase with the FM AOD
mean increase suggests the presence of detectable CM smoke and/or soil particles
induced by severe burning. The inability of the model to simulate potential CM smoke or
soil components associated with severe burning could be a contributing reason as to
why it performs less well in predicting CM AOD near BB sites.
3.3. Extreme biomass burning smoke AOD cases
A distinct class of extreme smoke cases comes from pyrocumulonimbus (pyroCb)
events induced by intense biomass burning sources: these events inject smoke high
into the troposphere or even well into the stratosphere (Fromm et al., 2010; Peterson et
al., 2017). A significant pyroCb smoke event that occurred over British Columbia (BC) in
August 2017 led to significant increases in various optical measures of aerosol
concentration in the lower Canadian and European Arctic (Peterson et al., 2018; Torres
et al., 2020; Das et al., 2021). Ranjbar et al. (2019) showed that a specific Aug. 19,
2017 smoke event over the high Arctic PEARL observatory at Eureka, Nunavut was
induced by the BC pyroCb fires and that it was a statistically significant extreme FM
AOD event. More recent eastern Siberian fires in June - August 2021, induced more
than a dozen cases of elevated smoke intrusion into the high Arctic with some smoke
plumes reaching the North Pole and/or its vicinity.  For example, on the 5$^{th}$ of August,
2021, operational NAAPS ( common chemistry, physics, and BB emission sources with
the NAAPS-RA) resolved a smoke plume north of 80°N (Fig. 5) with AOD values of 2-3.
Smoke AOD over the source region was also 2 to >3 with a similar amplitude to AODs
measured at Yakutsk. CALIOP data suggested a 1-6 km high smoke layer in the source
region.
Other extreme or near-extreme smoke events in the Arctic have been reported. A series
of intense fires originating in North America led to strong AOD peaks in the summer of
2015 over Svalbard (Markowicz et al., 2016; Lisok et al., 2018). Agricultural fires in
Eastern Europe in the spring of 2006 caused record-high AODs and pollution levels in
the European Arctic (Stohl et al., 2007). The North American boreal fires in the summer
of 2004 led to large-amplitude AOD peaks in Alaska and enhanced AODs on a pan-
Arctic scale (Stohl et al., 2004). Strong smoke events were also recorded during
intensive field campaigns, including the ARCTAS/ARCPAC campaign in the summer of
2008 (Matsui et al., 2011; Saha et al, 2010; McNaughton et al., 2011) and the
NETCARE research vessel (Canadian Arctic) campaign in the spring of 2015 (Abbatt et
al., 2019).

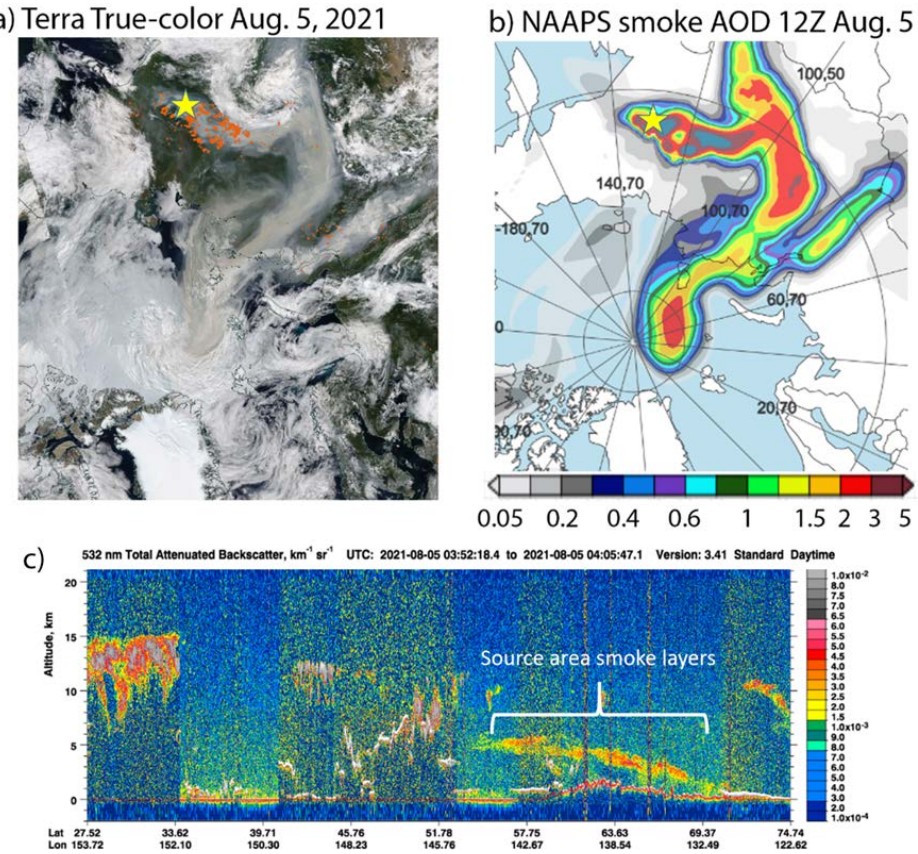


**Figure 5.** An August 5, 2021 example of BB smoke intrusion into the high Arctic from
fires originating in eastern Siberia. a) Composite true-color Terra satellite imagery. The
red dots represent satellite-detected fire hotspots. b) Operational NAAPS smoke AOD
analysis at 12Z. c) CALIOP 532 nm attenuated backscatter coefficient showing the
smoke layers around the source area. The yellow stars on a) and b) represent the
location of Yakutsk, which experienced a daily mean total AOD (500 nm) of 2.0 (FM
AOD ~1.9) and an intra-day peak around 2.5 (based on AERONET V3L1.5 data).
Sources: MODIS-Terra true-color satellite imagery and CALIOP-CALIPSO 532 nm
attenuated backscatter coefficient profile (respectively
https://worldview.earthdata.nasa.gov/ and https://www-calipso.larc.nasa.gov/).
3.4 Geographic distribution of extreme AODs
Having demonstrated that the NAAPS-RA simulations approximately reproduce the
statistics of the Arctic AERONET and MAN data over the 2003-2019 period we allow
ourselves the opportunity to exploit the spatial, temporal, and species-dependent model
capabilities to investigate Arctic wide variations and trends in terms of AOD and AOD
extremes.
The NAAPS-RA total-AOD map at different percentile levels for March-August 2003-
2019 is shown in Fig. 6 (projection north of 50°N). We separated the entire 2003-2019
period into early (2003-2009) and late (2010-2019) subperiods. The end-year of the first
period was chosen as 2009 given the drop in ABF/sulfate emissions due to the civil
Clean Air Acts enacted across the U.S. (e.g., Tosca et al., 2017; Kaku et al., 2018) as
well as Europe and China, and the attendant decrease in ABF/sulfate AOD in these
regions (Lynch et al., 2016; Zhang et al., 2017). This ABF/sulfate AOD decrease was
also observed in the Arctic, as shown in Fig. 13 of Part 1. The median Arctic AOD (<~
0.1 as compared with ~0.06 for the AERONET sites from Fig. 4 and Table 2) are an
order of magnitude smaller than the maximum AODs. Clear BB smoke features in the
North American and Asian boreal burning regions start to emerge in the $AOD_{95}$ maps
(see also Fig. 1). The maximum AOD is high (greater than 2.0) while being relatively low
over the Arctic Ocean (~ 0.3 - 1.0) and the North Atlantic (with the lowest values over
the generally high-elevation Greenland landmass). The maximum AOD is associated
with peak burning activities and generally occurs in July and August (except for the
Norwegian Sea area where the maximum AODs occurs in March-May; this, as can be
seen in Fig. 7, is possibly associated with a combined high AOD level from
anthropogenic pollutions, marine aerosols and springtime agriculture fires).

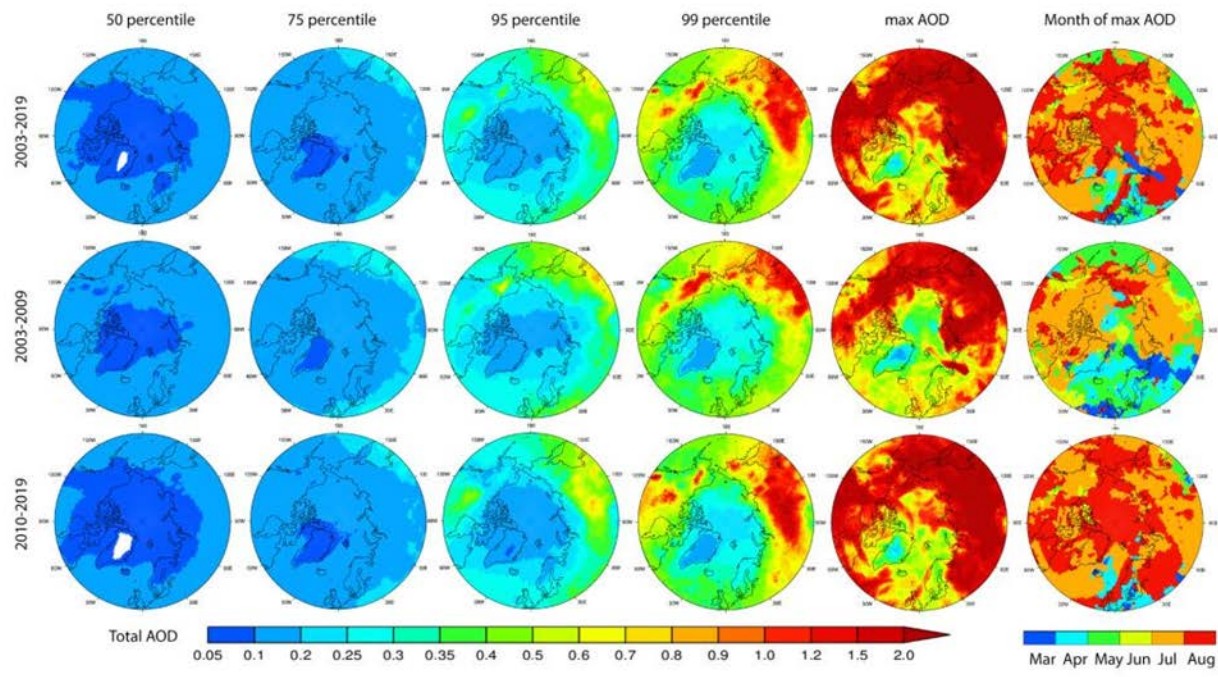


**Figure 6.** NAAPS-RA daily (550 nm) total-AOD maps (north of 50° latitude) at different percentile levels for the March-August time frame, the maximum AOD and (rightmost column) the month that the maximum AOD occurred. The three rows represent respectively, the sampling periods of 2003-2019, 2003-2009, and 2010-2019. The $AOD_{95}$ value for 2003-2019 is the same as that of Fig. 1 (aside from a different color scales).

The occurrence of different aerosol species (%) relative to the occurrence of total AOD for total AOD extreme events (March-August time frame) are shown in Fig. 7. Recall that an extreme total AOD event means total AOD > $AOD_{95}$ locally (the $AOD_{95}$ values can be inferred from the 95th percentile column of Fig. 6 and "locally" refers to the NAAPS-RA grid cell of 1° x 1°). The occurrence maps accordingly indicate which aerosol species are numerically dominant for extreme AOD events. As expected, BB smoke is the prevailing extreme event contributor over the North American and Asian Arctic (especially near the boreal source regions and associated transport pathways) as well as most of the Arctic ocean (except the Barents Sea and the Norwegian Sea). ABF occurrence dominates the low European Arctic. Sea-salt particles and, to a lesser extent, ABF are the most significant occurrence contributors, in the North Atlantic and the Norwegian Sea. Dust occurrences to extreme AOD events are very small (0-10%) except over the predominantly high-elevation region of Greenland where the relative occurrence of high-altitude African dust dominates the relative occurrence of the other species.

In terms of AOD amplitudes for total AOD extreme events (Fig. 8), BB smoke AOD shows dominant contributions, especially in the areas near the boreal source regions and transport pathways (including most areas of the high Arctic). ABF and sea salt show slightly higher extreme-event AODs than BB smoke over the North Atlantic and European Arctic. The regional extreme AODs are not, however, as large as the extreme AODs in the BB smoke-dominant regions.

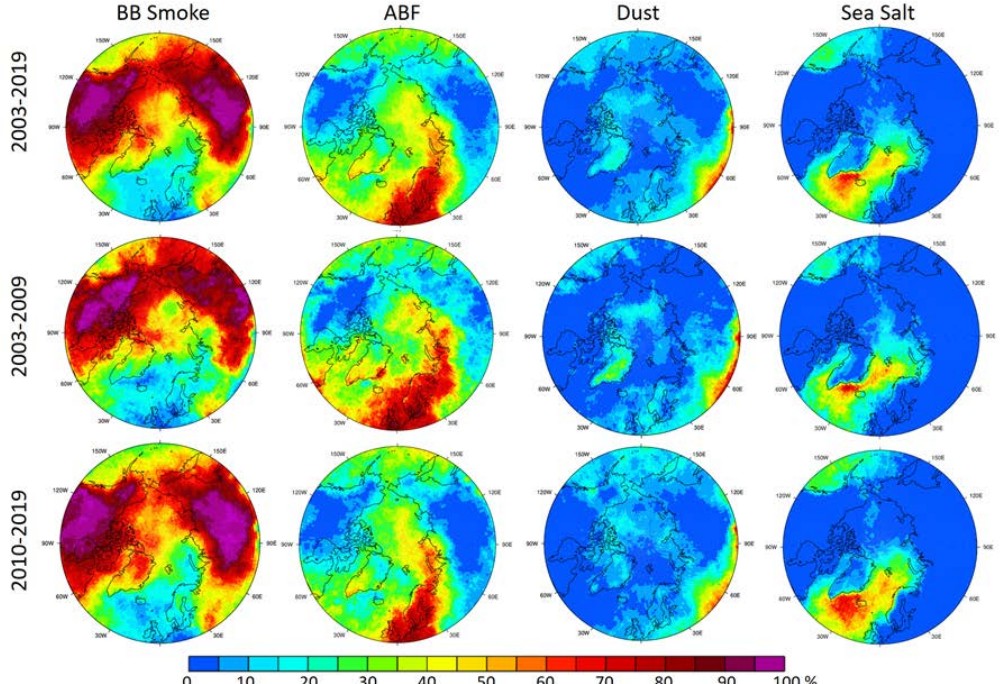

**Figure 7.** Occurrence of different aerosol species (expressed as a percent) relative to the occurrence of total AOD extreme events (daily total AOD > $AOD_{95}$ locally) for the March-August time frame (sampling periods, from top to bottom of 2003-2019, 2003-2009, and 2010-2019). The qualifier "locally" refers to a NAAPS-RA grid cell of 1° x 1°.

482

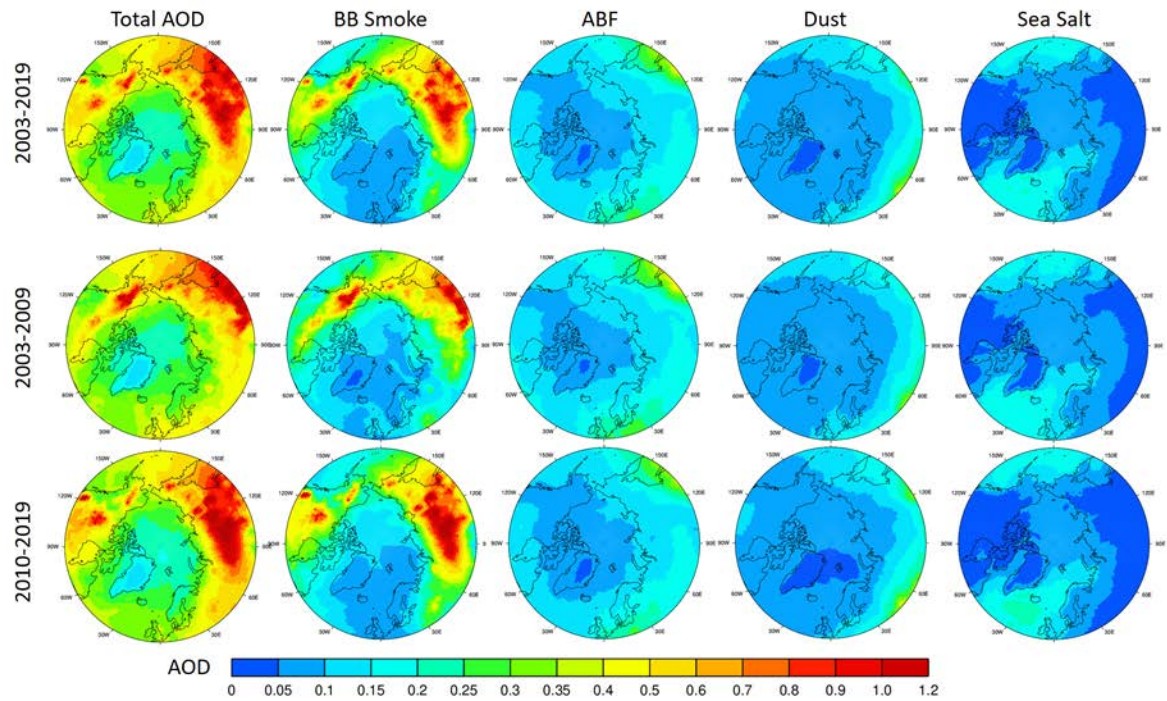

483

**Figure 8.** Mean speciated and total AODs averaged for days with speciated AOD or total AOD > $AOD_{95}$ for the March-August time frame (sampling periods, from top to bottom, of 2003-2019, 2003-2009, and 2010-2019).

3.5 Seasonality of extreme AOD events

Figure 9 depicts the NAAPS-RA seasonal cycle of total and speciated AOD for daily averages across the area north of 70° N ( a latitude limit which largely excludes BB source regions). The seasonal cycle of monthly mean total AOD (black solid circles in Fig. 9) shows relatively higher values in Mar-Apr-May (MAM) compared with the lower AODs in Jun-Jul-Aug (JJA), and a minimum in June. The spread of the ABF AOD seasonal values is moderately stable, with a relatively higher mean/median in MAM than JJA (see the Figure 9 caption for a definition of spread). Sea-salt AOD and its spread are relatively higher in the earlier months (March and April). Dust AOD and spread are generally stable through the season, with a visibly higher mean/median in April and May. Smoke AOD amplitude and spread exhibit the greatest inter-species seasonal variations with the lowest mean and spread in March, increased means and spreads in April, and significantly higher mean and spread in later months. July and August appear to have the largest mean, spread and maximum smoke AODs (a smoke importance statement that is generally consistent with the results of Fig.7). These smoke features significantly contribute to the seasonality of total AOD extremes. It is also noted that the MAM total and smoke AOD means approximately equal their

medians, but that the JJA means are greater than their medians (and that this is
especially true for August). The greater number of smoke AOD extremes in the later
season and the attendant consequence of greater positive histogram skewness would
explain those relative increases in the mean.

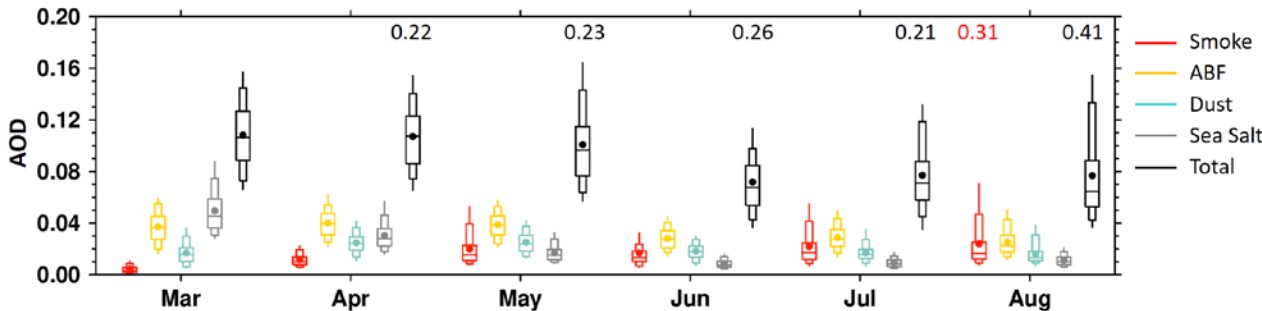


**Figure 9.** Box and whisker plot of daily and area-averaged (70°N-90°N) speciated AOD
at 550 nm from NAAPS-RA (2003-2019) for different months. The box and whisker
details are explained in the Figure 3 caption. Mean total AODs are shown as solid black
circles and maximum AODs as stars. Maximum AOD values appear as appropriately
colored numerical values if they extend beyond the 0.2 plot maximum. The "spread"
alluded to in the text refers to the spread of the boxes and whiskers where "whiskers"
includes the vertical spread of the boxes as well as the maximum value.
3.5 Trends of extreme AOD events
There is, as shown in Figure 12 of Part 1, a multi-year decreasing MAM trend and an
increasing JJA trend for total AOD in the Arctic over the 2003-2019 sampling period.
This was attributed to an overall decrease in MMA and JJA sulfate/ABF AOD coupled
with a negative trend in MAM, and a strong positive trend in JJA for biomass-burning
smoke AOD. In terms of extreme event trends, $AOD_{95}$ (Fig. 6) and the average AOD
above $AOD_{95}$ (Fig. 7) generally increased over the boreal continents from the 2003-
2009 to 2019-2019 period (with the notable exception of Alaska and northeastern
Siberia in 2010-2019). This is consistent with the (Part 1) positive BB emission trends in
JJA north of 50°N and 60°N (for which the JJA trend dominated the MAM trend
inasmuch as JJA was associated with much higher BB emissions).
The negligible or slight decrease in high Arctic (>70°N) $AOD_{50}$, $AOD_{75}$ and $AOD_{95}$
values from the 2003-2009 to the 2010-2019 period (Figure 6), is likely associated with
the generally weak ABF decrease seen in Figure 8. However, the increase in the
maximum AOD value (Fig. 6) and the contribution of BB smoke to AOD extreme events
(Fig. 8) in the latter period is an indication of stronger extreme BB smoke influence in
more recent years. It is also noted that the maximum high-Arctic AOD occurred later in
the season (mostly August) in 2010-2019 compared with the more balanced variation
occurring in March through August in 2003-2009. This is likely attributable to overall
lower ABF levels in the 2010-2019 period (especially in MAM), and a shift in extreme
smoke events to later in the season (Table 3, Fig. 10).

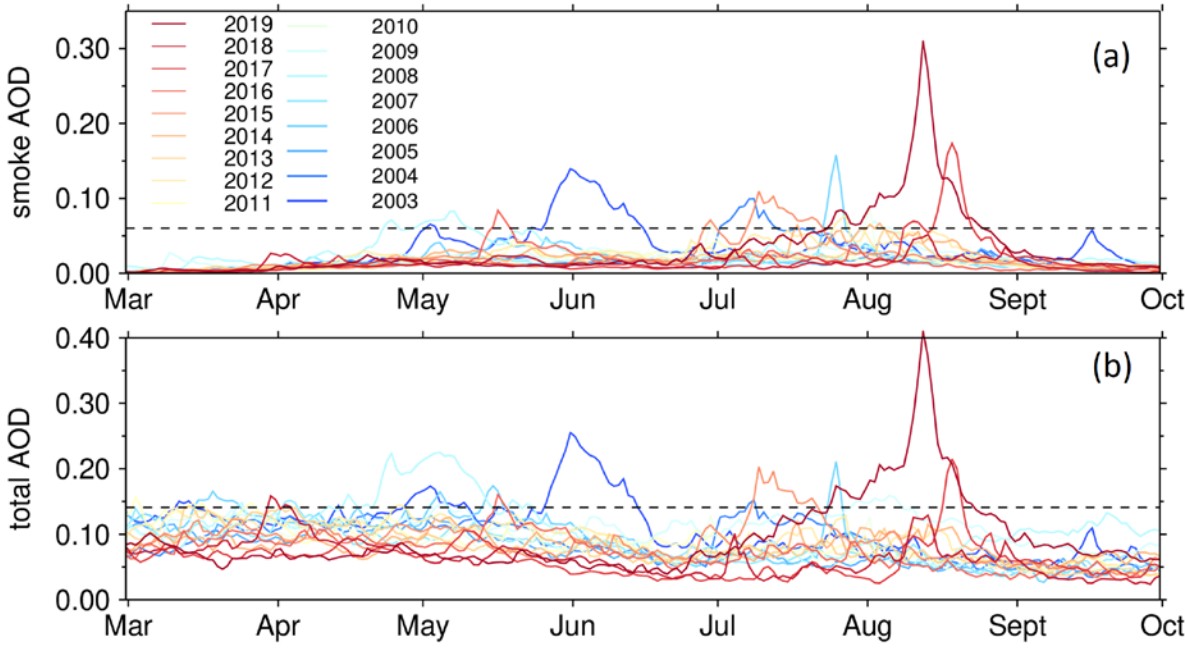


**Figure 10.** Seasonal (March to September) time series of daily-mean AODs averaged
over the (70°N-90°N) high-Arctic area for each individual year of the 2003-201 period:
(a) BB smoke AOD, and (b) total AOD. The years before 2010 are shown as cold
colors, and years after 2010 are shown as warm colors. The dashed horizontal lines
show the smoke $AOD_{95}$ value of 0.06 and the total $AOD_{95}$ value of 0.14 respectively
during the study period.

The time series of high-Arctic-averaged daily-mean BB smoke and total AOD from
March to September for all years between 2003-2019 is shown in Fig. 10. The extreme
total AOD variation is largely dictated by extreme BB smoke AOD. There is also a
discernible 2003-2009 to 2010-2019 springtime reduction in extreme total AOD: this, as
discussed in the previous paragraph, is likely due to an overall reduction in ABF AOD..
The occurrence of extreme smoke events tended to be more equally distributed over all
months (April-August) during the 2003-2009 period while being more concentrated in
the late season (July-August) during the 2010-2019 period. The extreme smoke and
total AOD trends resembled the extreme-smoke occurrence trends: more seasonally
balanced during the 2003-2009 period and summertime dominance during the 2010-
2019 period.

The occurrence of extreme high-Arctic smoke events thus demonstrates a clear smoke
and total AOD shift from a more balanced spring and summer to the late season

(notably the months of July and August; see also Table 3). This is consistent with the
temporal shift of fire activity to a later time in Siberia over 2003-2018 (Liu et al., 2020),
and the projection of emerging pan-Arctic fire regimes marked by increases in the
likelihood of extreme fires later in the growing season (McCarty et al., 2021). An earlier
fire season in the boreal region normally suggests a better-managed forest/land with
fewer large and destructive fires, while a later fire season indicates the opposite.
The shift of boreal fire activity, and the resulting BB smoke AOD extremes in the Arctic
from early season to late season, is probably related to early-season strengthening of
agriculture burning regulations and increased summertime lightning frequencies with
climate change in the latter decade. For example, the springtime BB smoke AOD peak
values in 2003, 2006 and 2008 are all associated with agricultural activity (resulting in
fires burning out of control) and widespread high-latitude burning (Korontzi et al, 2006;
Stohl et al., 2007; Saha et al., 2010). At the same time, with climate change, lightning
activity and lightning-caused wildfires in summertime high latitude regions were
observed to increase in the past two decades (Zhang et al., 2021; Bieniek et al, 2020;
Coogan et al., 2020). Also noted is a lengthening of growing season in boreal regions,
which infers lengthening fire season as well (Park et al., 2016). These factors aside,
climate oscillations, including the Arctic Oscillation, ENSO and Pacific Decadal
Oscillation, also affect boreal fire activities (Balzter et al., 2007; Macias Fauria and
Johnson, 2007; Kim et al., 2020). These climate factors also modulate interannual
variations and possibly the transport dynamics of pollutants from the mid-latitudes to the
Arctic region (e.g. Eckhardt et al., 2003; Fisher et al., 2010).
The dominant contributor (ABF) to regional extreme AOD occurrence and magnitude in
the lower European Arctic decreased slightly from 2003-2009 to 2010-2019 (Fig. 7 and
8): This observation is generally coherent with the Part 1 results showing a pan-Arctic
ABF AOD decrease in the 2003-2019 period and Fig. 10. Extreme total-AOD events
dominated by sea-salt contributions in the North Atlantic and Norwegian Sea increased
slightly in 2010-2019. This was possibly due to the observed increase in cyclonic
activities (Rinke et al., 2017; Waseda et al., 2021; Valkonen et al., 2021). Although the
model simulation of CM AOD is not as skillful as that of FM, trend analysis of CM AOD
which is based on relative change is arguably significant.
**Table 3.** Occurrence statistics of high-Arctic daily area-mean (>70˚N) BB smoke AOD extreme
event. These are defined as days with smoke AOD > smoke $AOD_{95}$ (~0.06) based on 2003-
2019 NAAPS-RA data. Years without an extreme smoke event are omitted but are counted in
the period average calculation. Cumulative extreme AOD is calculated as the sum of extreme
BB smoke AOD.

| year | Extreme BB smoke days | | | | | | max smoke AOD | cumulative extreme AOD |
|---|---|---|---|---|---|---|---|---|
| | APR | MAY | JUN | JUL | AUG | Annual total | | |
| 2003 | 0 | 9 | 16 | 0 | 0 | 25 | 0.14 | 2.4 |
| 2004 | 0 | 0 | 0 | 12 | 0 | 12 | 0.10 | 0.95 |
| 2006 | 0 | 0 | 0 | 4 | 0 | 4 | 0.16 | 0.49 |
| 2008 | 4 | 11 | 0 | 0 | 0 | 15 | 0.08 | 1.04 |
| 2009 | 0 | 0 | 0 | 0 | 5 | 5 | 0.07 | 0.32 |
| 2003-2009 ave | 0.6 | 2.9 | 2.3 | 2.3 | 0.7 | 8.7 | 0.08 | 0.74 |
| 2010 | 0 | 0 | 1 | 0 | 2 | 3 | 0.09 | 0.22 |
| 2012 | 0 | 0 | 0 | 3 | 0 | 3 | 0.08 | 0.22 |
| 2014 | 0 | 0 | 0 | 1 | 2 | 3 | 0.07 | 0.2 |
| 2015 | 0 | 0 | 2 | 17 | 0 | 19 | 0.11 | 1.51 |
| 2016 | 0 | 4 | 0 | 0 | 0 | 4 | 0.08 | 0.29 |
| 2017 | 0 | 0 | 0 | 0 | 13 | 13 | 0.17 | 1.27 |
| 2019 | 0 | 0 | 0 | 7 | 25 | 32 | 0.31 | 3.75 |
| 2010-2019 ave | 0 | 0.4 | 0.3 | 2.8 | 4.2 | 7.7 | 0.09 | 0.75 |


## 4. Conclusions

Aerosol optical depth (AOD) data from the U.S. Naval Aerosol Analysis and Prediction System-ReAnalysis (NAAPS-RA), the ground-based Aerosol Robotic Network (AERONET), and Marine Aerosol Network (MAN) were employed in analyzing the 2003-2019 statistics and trends of extreme Arctic-AOD events for spring and summer seasons (March-August). Extreme AODs are defined as any AOD greater than the 95th percentile ($AOD_{95}$) for any given distribution of AODs (whether that distribution is generated by the ensemble of AODs representing the time series of a specific location or of a regional average). Total, fine mode (FM) and coarse mode (CM) AODs at 550 nm from 6-hr resolution NAAPS-RA were first validated against AERONET and MAN AOD data. NAAPS-RA was shown to be capable of largely capturing FM and total AOD ranges and variability. The NAAPS-RA performance in simulating CM AOD was significantly better if the temporal resolution of the all-season statistics was less sensitive to high frequency dust and sea-salt events (i.e. the use of temporal resolution bins of a month rather than 6 hr). Statistics of the 6-hr Arctic AOD and extreme AOD events were analyzed. Finally, trends of extreme AOD in the Arctic were presented and analyzed.

***Baseline statistics for 6hrly AOD***: The median of 6-hr total AODs at 550 nm for all Arctic AERONET sites and MAN retrievals over the 2003-2019 period is ~ 0.06-0.07 while the 95th percentile value ($AOD_{95}$) is ~0.23. Both the median and $AOD_{95}$ values show a dominant FM AOD contribution. The CM AOD median is ~0.01 while $AOD_{95}$ is ~0.07. The maximum AOD over the 2003-2019 period varies between 0.5-3.0 for measurements made away from BB source regions, and 1.5 to greater than 3.0 for measurements made closer to BB source regions. The seasonal, NAAPS-RA spread of

smoke AOD is much higher than other speciated AODs (including anthropogenic and
biogenic fine (ABF), dust, and sea salt AODs) for all months between May and August:
the spread is especially large in July and August. These late-season smoke features
significantly contribute to the seasonality and interannual variabilities of extremes in
total AOD.
**Extreme AOD events:** Extreme AOD events using the Arctic spring and summer data
are largely attributable to FM AOD events (notably BB smoke transport events in
general). Extreme Arctic AOD events show large seasonal and interannual variability,
with the interannual AOD variability largely modulated by BB smoke. Extreme AOD
occurrences in the North American Arctic, the Asian Arctic, and the high Arctic (>70°N)
are dominated by BB smoke events (the lower European Arctic being the exception to
this affirmation). The occurrence of regionally extreme AOD events is attributed more to
ABF in the lower European Arctic. The extreme-event occurrence dominance of sea salt
aerosols is largely limited to the North Atlantic and Norwegian Seas. The extreme AOD
amplitudes of ABF and sea-salt AOD are, however, significantly lower than those
regions where extreme-AOD smoke AOD is dominant. Even for sites distant from BB
source regions, BB smoke is the principal driver of AOD variation above the $AOD_{95}$
threshold.
**Shift of extreme AOD events from spring-summer to summer season:** There is an
overall increase in the maximum AOD values in the high Arctic in 2010-2019 compared
to 2003-2009, suggesting stronger extreme BB smoke influence for more recent years.
Extreme AOD events are observed to occur in a more balanced fashion over the entire
April-August season during 2003-2009 while being more concentrated in the latter part
of the season (i.e., July and August) during 2010-2019. The seasonal shift in extreme
smoke AOD events is consistent with the multi-year negative MAM trend and positive
JJA trend in BB emissions (north of 50°N, Part 1). These trends are likely attributable to
early season agricultural burning controls, and increased lightning activity and lightning-
caused wildfires in summertime in the boreal high-latitude regions on top of the overall
lower level, especially in spring, of 2010-2019 vs 2003-2009 anthropogenic aerosols.
The shift in extreme smoke events is consistent with a general multi-year decreasing
springtime trend and an increasing summertime trend of BB emissions north of 50° N
(Part 1).
Global warming is expected to continue generating drier conditions and increased
wildfire activities in the high latitudes (McCarty et al., 2021) and thus render the Arctic
more susceptible to extreme smoke events. These events can significantly change the
regional aerosol budget by bringing large amounts of smoke aerosols into the Arctic.
These extreme smoke events will likely play an increasingly important Arctic aerosol
budget role given the decreasing (Part 1) baseline in anthropogenic pollution aerosols

over the 2003-2019 period. Smoke aerosols are, notably, much more light-absorbing than anthropogenic sulfate. As well, their different physical and chemical properties relative to anthropogenic aerosols will translate into different efficiencies in their role as CCN and IN. When deposited on surface snow and ice, they impact the surface radiative forcing budget by reducing surface albedo. The climate impacts of BB smoke would, accordingly, differ and possibly counteract the dynamics of anthropogenic aerosols. Therefore, the baseline AOD trends reported in Part 1 and the trends in extreme AOD events reported here are important in terms of implications for the changing Arctic climate. The greater sensitivity of Arctic climate to aerosol forcings relative to other regions of the globe (e.g. Wang et al., 2018), the impact of the extreme BB smoke events and their interannual variability and trends on Arctic climate warrants further exploration. The statistics of extreme AODs reported here are expected to help in the formulation of climate sensitivity experiments and improve our knowledge of the relative importance of aerosol processes compared to other factors of the changing Arctic climate.

**Code and Data Availability:** All data supporting the conclusions of this manuscript are available through the links provided below.

AERONET Version 3 Level 2 data:  http://aeronet.gsfc.nasa.gov

MAN data: https://aeronet.gsfc.nasa.gov/new_web/maritime_aerosol_network.html

NAAPS RA AOD: https://usgodae.org//cgi-bin/datalist.pl?dset=nrl_naaps_reanalysis&summary=Go

**Author contributions:** P.X. designed this study, performed most of the data analysis and wrote the initial manuscript. All authors contributed to scientific discussion, revision and editing of the manuscript.

**Competing interests:** The authors declare that they have no conflict of interest.

**Acknowledgments**

We thank the NASA AERONET, and MAN, and Environment and Climate change Canada (ECCC) AEROCAN group for the sun-photometer data. We acknowledge the use of imagery from the NASA Worldview application (https://worldview.earthdata.nasa.gov, last access: Mar 11, 2022), and NASA CALIPSO website (https://www-calipso.larc.nasa.gov/).

**Financial support**

The authors acknowledge support from NASA's Interdisciplinary Science (IDS) program (grant no. 80NSSC20K1260), NASA's Modeling, Analysis and Prediction (MAP) program (NNX17AG52G) and the Office of Naval Research Code 322. N.O. and K.R's work was supported by the Canadian Space Agency, SACIA-2 project, Ref. No. 21SUASACOA, ESS-DA program.

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
