# Peer review of "Arctic spring and summertime aerosol optical depth baseline from"

_Atmospheric Chemistry and Physics, 2021_

## Referee Comment (RC1)

"Artic spring and summertime aerosol optical depth baseline from long-term observations and model reanalyses, with implications for the impact of regional biomass burning processes" by Xian et al. takes a multi-sensor/dataset approach to characterizing Arctic aerosols climatologically and their trends over the past almost two decades. These results are interpreted geographically, seasonally, as well as by aerosol species and instrument sensors.

In general, my recommendation to the editor is minor revisions for this publication. While I am not familiar with all the literature out there on Arctic aerosols, this study seems to be quite comprehensive, which provides value in characterizing Arctic aerosols from many different angles across the entire region. However, there were a few scientific and presentation matters that should be addressed prior to publication.

For the AERONET data, I wonder about how the availability of AERONET data plays out when using a 6-hour averaging interval. AERONET data is primarily a daytime measurement, therefore it is affected by the changing of daylight. For example, if there are more measurements closer to summer solstice because of daylight hours, does that impact the results? I am not sure how that would play out on the results here but see Appendix B of "The Diurnal Variation of the Aerosol Optical Depth at the ARM SGP Site" by Balmes et al. (2021, Earth and Space Science; doi.org/10.1029/2021EA001852) which showed that the changing of the season affected the diurnal cycle of AOD when considering AERONET measurements. Since AERONET is the basis for much of the comparison in this study, the averaging interval should be carefully considered to ensure the conclusions are not artifacts of data availability.

An additional scientific issue I wonder about is that the CALIOP data only considers AODs greater than zero. Do other studies do this with CALIOP data? While there are instrument sensitivity limitations that preclude detecting all aerosols, leaving out when AOD=0 will artificially increase the mean AOD to a value not actually observed by the instrument. It is well documented that CALIOP cannot detect all aerosols and clouds and several of studies are cited in this reference, however, perhaps it would be more representative of the data to also include figures and data if AOD=0 is considered for CALIOP. Another option is the Level 3 AOD product which attempts to overcome the sensitivity issue. This data product is mentioned in the discussion but perhaps more discussion or a supplementary figure showing the various CALIOP AOD results from different data products and thresholds would be more representative of the instrument and data products.

Below are minor comments I had and typos I found while reviewing:

Minor Comments:

The title is really long. Perhaps it should be shortened for brevity.

Lines 138-139: "We define the Arctic/high-Arctic as regions north of 60°N/70°N, and sub-Arctic as regions between 60°N-70°N." It took me a second read through to understand this correctly. Since it is a definition sentence, it seems worth it to make it two sentences or edit it for clarity.

Data section: there is quite a lot of data used in this section so it leaves the reader a little overwhelmed to read through as well as to reference later on in the paper. Perhaps a table listing all the data described would be a useful summary to reference?

Figure 1: "Warm colors represent fine mode and cool colors represent coarse mode." I think this should be more explicit to avoid confusion, e.g., "warm colors (red, orange, and pink) … cool colors (green and blue)"

Line 765: "(i.e. the square …" should have a comma after the i.e

Line 989-997 and throughout: I think "95% percentile mark" should be "95th percentile mark"? 95% percentile sounds redundant

Figure 16: I think there may be a typo in the caption as it says 12 September 2012 after August 5, 2021?

Line 1078: "black colors, respectively"

Line 1124: figures should be capitalized

Line 1134: Does the parenthesis starting "(e.g. …" go all the way to line 1139? I think this should be rewritten, very challenging to make sense of a 5 line parentheses

Line 1134 and 1143: should have a comma after e.g. I think this might be an issue throughout for i.e. and e.g. so check throughout the text

---

## Author Comment (AC1)

Reply to review comments from reviewer #2

*"Artic spring and summertime aerosol optical depth baseline from long-term observations and model reanalyses, with implications for the impact of regional biomass burning processes" by Xian et al. takes a multi-sensor/dataset approach to characterizing Arctic aerosols climatologically and their trends over the past almost two decades. These results are interpreted geographically, seasonally, as well as by aerosol species and instrument sensors.*

*In general, my recommendation to the editor is minor revisions for this publication. While I am not familiar with all the literature out there on Arctic aerosols, this study seems to be quite comprehensive, which provides value in characterizing Arctic aerosols from many different angles across the entire region. However, there were a few scientific and presentation matters that should be addressed prior to publication.*

Reply: We thank for the reviewer's comments, which we think helps to improve the clarity and presentation of the manuscript.

*For the AERONET data, I wonder about how the availability of AERONET data plays out when using a 6-hour averaging interval. AERONET data is primarily a daytime measurement, therefore it is affected by the changing of daylight. For example, if there are more measurements closer to summer solstice because of daylight hours, does that impact the results? I am not sure how that would play out on the results here but see Appendix B of "The Diurnal Variation of the Aerosol Optical Depth at the ARM SGP Site" by Balmes et al. (2021, Earth and Space Science; doi.org/10.1029/2021EA001852) which showed that the changing of the season affected the diurnal cycle of AOD when considering AERONET measurements. Since AERONET is the basis for much of the comparison in this study, the averaging interval should be carefully considered to ensure the conclusions are not artifacts of data availability.*

Reply: AERONET data is more available during summer than in springtime due to longer daylight. This is reflected in the difference of the total number of observations for JJA and MAM in Table 1. We expect some impact of summer vs spring sampling on the annually-averaged diurnal variation of the AOD if there is baseline AOD change between the two seasons (e.g. much higher summer AOD than spring AOD as shown in Appendix B of Balmes et al. (2021)). However diurnal variation of the AOD is not the focus of this manuscript, nor is the annual-mean diurnal cycle (actually, spring- and summer-averaged diurnal cycles would help avoid those spring vs summer sampling influences on the annually-averaged diurnal cycle). Nevertheless, we performed resampling of AERONET AOD data to demonstrate that our result is changed little by 6hrly vs daily sampling. Most of the Arctic AERONET sites have stable numbers of observations between 6-18hr local time (i.e., magnitude of the number of observations are stable between 6-18hr, while it drops at some earlier or later hours): so we generated AERONET daily AOD statistics with data restricted to 6-18hr local time. A supplemental table was produced to enable a direct comparison of Table 1 with the

6hrly statistics. The following text is added in section 2.4 AERONET AOD data introduction:

"To explore the potential impact of different temporal sampling on the result (e.g., Balmes et al., 2021), we generated AERONET daily AOD statistics (Table S1) to enable a direct comparison of Table 1 with the 6hrly statistics. In general, the mean and median of MAM or JJA AODs (including total, FM and CM AODs) at the ten AERONET sites change very slightly (mostly 0.00, or <=0.01). As expected, the standard deviation is smaller for the daily AOD case than for 6hrly AOD (due to temporal averaging)."

**Table S1.** Analogous table to Table 1 but using daily AOD statistics. The AERONET daily AOD statistics was generated with data restricted to 6-18hr local time when there are stable numbers of observations (i.e., magnitude of the number of observations are stable between 6-18hr, while it drops at some earlier or later hours).

| sites | latitude | longitude | elevation (m) | region | MAM (mean\|median\|std) total AOD | FM AOD | CM AOD | n | MAM FMF mean | median | JJA (mean\|median\|std) total AOD | FM AOD | CM AOD | n | JJA FMF mean | median |
|---|---|---|---|---|---|---|---|---|---|---|---|---|---|---|---|---|
| Hornsund | 77.0N | 15.6E | 12 | Svalbard | 0.10\|0.09\|0.05 | 0.07\|0.06\|0.04 | 0.03\|0.02\|0.03 | 215 | 0.72 | 0.76 | 0.08\|0.06\|0.07 | 0.06\|0.04\|0.07 | 0.02\|0.01\|0.02 | 302 | 0.76 | 0.81 |
| Thule | 76.5N | 68.8W | 225 | Greenland | 0.09\|0.07\|0.05 | 0.06\|0.06\|0.03 | 0.03\|0.01\|0.04 | 324 | 0.76 | 0.81 | 0.07\|0.05\|0.08 | 0.06\|0.04\|0.08 | 0.01\|0.01\|0.02 | 464 | 0.85 | 0.87 |
| Kangerlussuaq | 67.0N | 50.6W | 320 | Greenland | 0.07\|0.06\|0.03 | 0.05\|0.04\|0.02 | 0.02\|0.02\|0.02 | 295 | 0.69 | 0.72 | 0.07\|0.05\|0.05 | 0.05\|0.04\|0.04 | 0.01\|0.01\|0.03 | 476 | 0.77 | 0.82 |
| Ittoqqortoormiit | 70.5N | 21.0W | 68 | Greenland | 0.06\|0.06\|0.03 | 0.04\|0.04\|0.02 | 0.02\|0.01\|0.03 | 193 | 0.72 | 0.78 | 0.06\|0.04\|0.04 | 0.05\|0.03\|0.04 | 0.01\|0.01\|0.02 | 369 | 0.80 | 0.84 |
| Andenes | 69.3N | 16.0E | 379 | Norway | 0.09\|0.07\|0.06 | 0.05\|0.04\|0.04 | 0.03\|0.02\|0.04 | 226 | 0.67 | 0.72 | 0.08\|0.06\|0.05 | 0.06\|0.05\|0.05 | 0.02\|0.01\|0.02 | 331 | 0.75 | 0.79 |
| Resolute_Bay | 74.7N | 94.9W | 35 | Canada | 0.10\|0.09\|0.05 | 0.07\|0.06\|0.03 | 0.03\|0.02\|0.03 | 173 | 0.72 | 0.74 | 0.07\|0.05\|0.09 | 0.06\|0.04\|0.09 | 0.02\|0.01\|0.02 | 371 | 0.78 | 0.83 |
| Barrow | 71.3N | 156.7W | 8 | Alaska | 0.12\|0.09\|0.10 | 0.08\|0.06\|0.07 | 0.04\|0.02\|0.06 | 158 | 0.69 | 0.74 | 0.09\|0.06\|0.09 | 0.07\|0.05\|0.09 | 0.02\|0.01\|0.02 | 335 | 0.79 | 0.82 |
| Bonanza_Creek | 64.7N | 148.3W | 353 | Alaska | 0.11\|0.07\|0.09 | 0.06\|0.04\|0.07 | 0.04\|0.02\|0.04 | 297 | 0.64 | 0.65 | 0.18\|0.09\|0.27 | 0.16\|0.06\|0.26 | 0.02\|0.02\|0.02 | 445 | 0.78 | 0.82 |
| Tiksi | 71.6N | 129.0E | 17 | Siberia | 0.09\|0.10\|0.03 | 0.07\|0.07\|0.02 | 0.03\|0.02\|0.02 | 13 | 0.73 | 0.78 | 0.13\|0.08\|0.19 | 0.11\|0.07\|0.18 | 0.02\|0.01\|0.02 | 139 | 0.81 | 0.85 |
| Yakutsk | 61.7N | 129.4E | 119 | Siberia | 0.15\|0.11\|0.15 | 0.11\|0.08\|0.13 | 0.04\|0.02\|0.06 | 517 | 0.73 | 0.77 | 0.17\|0.09\|0.23 | 0.14\|0.07\|0.23 | 0.02\|0.01\|0.03 | 748 | 0.81 | 0.84 |

*An additional scientific issue I wonder about is that the CALIOP data only considers AODs greater than zero. Do other studies do this with CALIOP data? While there are instrument sensitivity limitations that preclude detecting all aerosols, leaving out when AOD=0 will artificially increase the mean AOD to a value not actually observed by the instrument. It is well documented that CALIOP cannot detect all aerosols and clouds and several of studies are cited in this reference, however, perhaps it would be more representative of the data to also include figures and data if AOD=0 is considered for CALIOP. Another option is the Level 3 AOD product which attempts to overcome the sensitivity issue. This data product is mentioned in the discussion but perhaps more discussion or a supplementary figure showing the various CALIOP AOD results from different data products and thresholds would be more representative of the instrument and data products.*

Reply: We have now included more discussions in section 6 about the CALIOP AOD data used in the study and provided a supplemental figure (Fig. S4; also attached at the end of this reply document for your convenience) to help the comparison between the analysis with AOD=0 retained and removed. The artificial AOD value of zero is known by the CALIOP developing team and we are also well aware of the issue. We are hesitated to include AOD=0 into our study because those air columns with AOD=0 from

CALIOP actually represent air columns where CALIOP are blind to aerosol particles. Based on the paper first authored by Travis Toth (Toth et al., 2018), who is a coauthor of this paper, those air columns could have corresponding AOD (532 nm) of 0 to 0.1 or higher based on AERONET data. Giving the relative low mean AODs over or near the Arctic region, adding those air columns may likely introduce a low bias in climatological mean of AOD over the study region.

Toth T.D., J.R. Campbell, J.S. Reid, J.L. Tackett, M.A. Vaughan, and J. Zhang, Minimum Aerosol Layer Detection Sensitivities and their Subsequent Impacts on Aerosol Optical Thickness Retrievals in CALIPSO Level 2 Data Products, Atmos. Meas. Tech., 11, 499-514, https://doi.org/10.5194/amt-11-499-2018, 2018.

Our expanded discussion regarding this aspect in section 6 reads
"Often artificial AOD value of zero are observed over the Arctic in CALIOP V4.2 L2 and L3 data, resulted partially from algorithmically setting altitude bins with retrieval filled values in the aerosol profile to zero, as these represent undetectable levels of faint aerosol (i.e.,Toth et al., 2016; 2018). With AOD=0 values retained in the CALIOP V4.2 L2 data analysis (same processing in CALIOP V4.2 L3), the climatological seasonal mean AOD magnitude is much smaller (about half) than that shown in Fig. 3 and the AOD trends are slightly smaller than those in Fig. 9, although the spatial patterns of the seasonal AOD and trends are similar to those obtained with AOD data after removing the AOD=0 values (Fig. S4). After removing the pixels with filled and zero values, CALIOP AOD seasonal spatial AOD distributions are similar to those from MODIS and MISR. "

*Below are minor comments I had and typos I found while reviewing:*
*Minor Comments:*

*The title is really long. Perhaps it should be shortened for brevity.*
Reply: We think the current title represent the essence of the study and express the important implications for biomass burning processes, which we are really reluctant to remove.

*Lines 138-139: "We define the Arctic/high-Arctic as regions north of 60°N/70°N, and sub-Arctic as regions between 60°N-70°N." It took me a second read through to understand this correctly. Since it is a definition sentence, it seems worth it to make it two sentences or edit it for clarity.*
Reply: We have revised the statement to "We define the Arctic and the high-Arctic as regions north of 60°N and 70°N respectively. The lower-Arctic is defined as regions between 60°N-70°N.". We also changed all "sub-Arctic" to "lower-Arctic".

*Data section: there is quite a lot of data used in this section so it leaves the reader a little overwhelmed to read through as well as to reference later on in the paper. Perhaps a table listing all the data described would be a useful summary to reference?*
Reply: We have added the following table in the Appendix A and appended in the first paragraph of the data section "A summary of the datasets is provided in Appendix A."

| Products | Data | resolution | time |
|---|---|---|---|
| MODIS (Moderate Resolution Imaging Spectroradiometer) C6.1L3 | 550nm AOD | 1°x1° monthly | 2003-2019 |
| MISR (Multi-angle Imaging SpectroRadiometer) V23 | 558nm AOD | 1°x1°, monthly | 2003-2019 |
| CALIOP (Cloud-Aerosol Lidar with Orthogonal Polarisation) V4.2L2 | 532nm AOD | 2°x5°, monthly | 2006-2019 |
| AERONET (AErosol RObotic NETwork) V2L3 | SDA total, FM, CM AOD at 550nm | 6hrly, monthly | 2003-2019 |
| MAN (Marine Aerosol Network) Level2 | SDA total, FM, CM AOD at 550nm | 6hrly, monthly | 2003-2019 |
| MERRA-2 (Modern-Era Retrospective Analysis for Research and Applications, v2) | Total and speciated AOD at 550nm | 0.5°lat x0.63°lon, monthly | 2003-2019 |
| CAMSRA (Copernicus Atmosphere Monitoring Service Reanalysis) | Total and speciated AOD at 550nm | 0.7°x0.7°, monthly | 2003-2019 |
| NAAPS-RA v1 (Navy Aerosol Analysis and Prediction System reanalysis v1) | Total and speciated AOD at 550nm | 1°x1°, 6hrly, monthly | 2003-2019 |
| MRC (Multi-Reanalysis-Consensus) | Total and speciated AOD at 550nm | 1°x1°, monthly | 2003-2019 |
| FLAMBE (Fire Locating and Modeling of Burning Emissions) v1.0 | BB smoke emission flux | 1°x1°, monthly | 2003-2019 |

Note: These are final form of data used in the result section. Some pre-processing and quality-control were applied to remote sensing data as described in the data section.

*Figure 1: "Warm colors represent fine mode and cool colors represent coarse mode." I think this should be more explicit to avoid confusion, e.g., "warm colors (red, orange, and pink) … cool colors (green and blue)"*
Reply: Revised.

*Line 765: "(i.e. the square …" should have a comma after the i.e*
Reply: Corrected.

*Line 989-997 and throughout: I think "95% percentile mark" should be "95th percentile mark"? 95% percentile sounds redundant*
Reply: Revised accordingly.

*Figure 16: I think there may be a typo in the caption as it says 12 September 2012 after August 5, 2021?*
Reply: Thanks for capturing this typo. It is corrected.

*Line 1078: "black colors, respectively"*
Reply: Revised.

*Line 1124: figures should be capitalized*
Reply: Revised.

*Line 1134: Does the parenthesis starting "(e.g. …" go all the way to line 1139? I think this should be rewritten, very challenging to make sense of a 5 line parentheses*
Reply: We have broken this into two sentences to make it easier to read.

*Line 1134 and 1143: should have a comma after e.g. I think this might be an issue throughout for i.e. and e.g. so check throughout the text*
Reply: Thanks! This is corrected throughout the text.

[Figure]

**Figure S4.** CALIOP mean climatological MAM (upper-left) and JJA (upper-right) AOD at 532 nm (2006-2019) and AOD trends (lower) derived with AOD=0 values retained in the CALIOP V4.2 L2 data analysis, to compare with CALIOP AOD seasonal climatology and trends derived with AOD=0 values removed in Fig. 3 and Fig. 9. White area means lack of data.

---

## Author Comment (AC2)

*The paper entitled "Arctic spring and summertime aerosol optical depth baseline from long-term observations and model reanalyses, with implications for the impact of regional biomass burning processes" by Peng Xian and coauthors presents a comprehensive view on long-term measurements and modelling of aerosol optical depth (AOD) in the Arctic. They consider ground-based AERONET sun photometer measurements, observations by three spaceborne instruments, and results from three aerosol reanalyses as well as their composite to investigate (i) the consitency of the different data sets, (ii) the annual and seasonal variation as well as the long-term trend in AOD together with the importance of biomass-burning smoke, and (iii) statistics on the occurrence of extreme AOD events.*

*While the work is of interest to the readers of ACP, it is far to much material for one publication. This review is late, also because it is impossible to read the manuscript in one sitting. In fact, the content could be split in as much as three papers according to the list of topics provided above. Such an approach would lead to very good papers that could be much more reader-friendly than the current submission. This reviewer therefore recommends to reject the paper in its current form and to re-submit after a thorough revision of content and readability. Alternatively, the work requires major revisions, shortening, and a decision on which of the three topics to focus in this particular submission.*

Reply: We thank the reviewer for the comments on this manuscript. We intend to link remote sensing, ground observations, and reanalyses to provide comprehensive information on the Arctic AOD so that readers can have a consistent picture of the Arctic AOD status. It is known that the Arctic has far fewer observations compared to other regions of the world, so a single dataset or aspect (remote sensing vs modeling) is challenged to provide a true and comprehensive view of the Arctic AOD. We think it is essential to present the reanalyses results along with remote sensing and ground observations so that these complimentary data can be inter-supportive and comprise a rich and reliable picture of Arctic AOD. Surface-based AOD measurements stand for "accuracy", and satellite products provide "coverage", while model reanalyses provide information of "aerosol processes" that can help explain what's observed. We think missing either one of the three components would greatly degrade the strength of the study.

The consistency of the different data sets is discussed in the manuscript with a highlight on the effect of quality control of the remote sensing datasets. We highlight the importance of quality control because our climatological AOD (less than 0.1 near 70N) is quite different from some other studies using off-the-shelf MODIS and MISR products (specifically Tomasi et al., 2015 of AOD value of 0.2 near 70N; our result is more in line with AERONET measurements; see section 6 discussion). The requirement of quality control on the remote sensing data for data assimilation, climate/trend analysis (our case here) purposes is demonstrated in numerous studies (e.g. citations within the manuscript for MODIS, MISR, CALIOP products, Zhang et al., 2006; 2010; 2017; Shi et al., 2011, 2013; Toth et al., 2018; Hyer et al., 2015). So we think quality control is necessary but it does not need to be further addressed as the methods used here are common.

In response to the reviewer's concern on the length of the manuscript, we have split the manuscript into two separate manuscripts. One focuses on AOD climatology and trend (now Part 1) and the other focuses on statistics of extreme AODs (now Part 2). Part 1 has 55 pages, including 13 figures and 4 Tables (and references). Part 2 has 34 pages including 10 figures and 3 tables (and references). Part 1 is similar to the original manuscript, except for

the removal of the extreme AOD events and with marginal changes incorporating reviewers' and coauthors' comments. Part 2 focuses on extreme AOD events and with four additional figures and one additional table providing in-depth analyses of extreme AOD events (e.g., the geographical distribution of AODs at the 95$^{th}$ percentiles, a figure illustrating the trend of extreme AOD events). The added table (Table 1) is to provide AERONET site information and basic AOD statistics (similar to Part 1). We thank the reviewer's suggestion for splitting the paper into more than 1 papers.

*Please find some more specific comments below:*

*- The paper is rather lengthy and would benefit from trimming the text and content to what's really needed. A start would be a shorter title such as, e.g. to Arctic spring and summertime aerosol optical depth baseline from long-term observations and model reanalyses.*

Reply: The paper is split into two manuscripts as mentioned above. The titles are shorter than before.

*- Entire paragraphs could be omitted as they are repeating points made earlier or are just redundant, e.g. lines 24-31 (not needed in the Abstract), 141-148, 176-182, 299-303 (why mention if the statement end with "is not used here"),...*

Reply: Line 24-31: We think the inclusion of the discussions in the abstract is a personal preference. We prefer abstracts to be stand-alone with sufficient info for readers. Therefore, we kept the discussions in the abstract.

Line 141-148: The advantage of high temporal resolution biomass burning emissions used in the aerosol reanalyses was mentioned in two places in the manuscript: One in the introduction (listed line numbers), and the other in the discussion. We'd really like to highlight this point so we've kept it in the introduction as well as in the discussion.

Line 176-182, 299-303: These lines are removed following the reviewer's suggestion.

*- The entire part about FLAMBE (description and results) could be omitted. In fact, the point made here could be condensed down to something along the lines of "findings are also supported by burning emissions from FLAMBE" later in the discussion.*

Reply: The FLAMBE biomass burning (BB) emission climatology and the trend, which are derived based on satellite observations, are a critical data source for revealing changes in BB aerosol patterns in the Arctic region. Thus, we would like to include an introduction of FLAMBE for the benefit/convenience of readers. The FLAMBE-related information is only one figure and it is kept in the manuscript.

*- There are way too many figures for one publication. In addition, some information in these figures could be moved to the supplement to improve the discussion of the findings. For instance, the main figures could stick to the Multi-Reanalysis Concensus and their discussion could link to more detailed figures including the specific findings of the three models in the supplement. The authors should re-evaluate if a figure that isn't thoroughly discussed in the text is needed in the manuscript.*

Reply: See our reply to the first comment on the length of the manuscript. Regarding the usage of reanalyses, many of the figures contain AOD information from all three reanalyses and their consensus, because we intend to show the diversity and similarity of these reanalyses and avoid the consensus being dominated by any specific reanalysis product. We think this makes our result more convincing while providing information for readers interested in the difference of the reanalyses. We think every figure is discussed thoroughly except for some aspects that are supportive of other results in the manuscript (where we stated something like, for example, for the trend in reanalyses "consistent with the trend in remote sensing AOD").

*- The study makes use of height-resolved measurements from CALIOP and considers detailed aerosol re-analyses fields. The work would be even stronger of this information was to be used to also investigate the vertical distribution of Arctic aerosols. Such an attempt would partly compensate for the disadvantage of AOD to refer to column aerosol load.*

Reply: As the reviewer commented earlier, the current manuscript is already long, so we prefer to focus on AOD only for this submission. We mentioned that the vertical distribution of aerosols for extreme events "is the topic for another manuscript" in the original submitted manuscript. We are working on that aspect and will report our findings in another manuscript. Thank you.

---

## Author Response (AR2)

Reply to review comments #2

This is my second review of Peng Xian et al.: Arctic spring and summertime aerosol optical depth baseline from long-term observations and model reanalyses.
The authors have followed my suggestion to split their work in two papers, which somewhat helps with taking in all the provided information. However, they have dismissed most of my other suggestions without providing convincing rebuttals. Therefore, I am sorry for having to repeat much of my earlier criticism. I have worked through both versions and it took me considerable time to do so. My suggestions are therefore focussed on distilling the provided information down to what is essential and to make it easier for potential readers to take it in. I start with general comments that apply to both papers before providing specific comments for the individual parts.
Reply:  We really appreciate the reviewer's detailed and constructive comments.

General comments
The authors should make a serious effort in improving the quality of their writing. The current version of the papers does not make for enjoyable reading.

• There a plenty of statements that span over 4, 5 or even 6 lines and are very hard to follow. Please carefully go through the manuscripts and shorten or split those sentences.
Reply: We've made significant efforts splitting long sentences or shortening them.

• The authors' use of parentheses to specify a statement made just before is particularly annoying. Why not go with the text in the parentheses in the first place? Or make up your mind which one is the preferred phrasing. There are so many of these statements that I won't list them. Please revise.
Reply: We've also removed dozens of parentheses by incorporating the information explicitly into sentences or removing the unnecessary/repetitive information.

• The authors should properly define all the used terms in the beginning or when first used and stick to those terms rather than providing their definition whenever they are used. This occurs particularly often for the term MAM, JJA, and high or low Arctic. Please make sure that all acronyms are introduced only once in the text. No need to re-do so in the summary. The readers should also be able to remember the extend of the covered time period without repeating it at almost every instance.
Reply:  Thanks for pointing this out. The first author defined the acronyms again in the summary section, besides where they were first used, as a convention (due to requirements of some other journals and that often times readers will just read the conclusions). We read the ACP submission guide and don't see such a requirement, so we have followed the reviewer's suggestion and dropped the definitions of acronyms in the summary section and a few places in the middle of the text. Following the reviewer's suggestion, we've also removed over a dozen of "2003-2019", which is the extent of the covered time period.

• The authors give values in the text that are often put behind a ~, i.e. maximum AOD of ~1.2. Do you know the specific value or do you provide estimates? Please be specific.
Reply:  We have now provided specific numbers whenever applicable.

• The text is very repetitive. For instance, the point that biomass-burning smoke is the dominating contributor to Arctic AOD in summer is repeated at multiple occasions. Please identify the instances where repetitive statements fit best and move them there. You can expect the reader to remember. Please thoroughly go through both manuscripts to get rid of redundant text, figures and tables. Some examples are given in the specific comments below.

Reply: We have removed repetitive sentences as suggested (see our replies to specific comments) and in other identified places in the text.

• The abstracts of both papers – but particularly of the second one - are much too long. Please try to reduce to the essential findings.

Reply: We've shortened the discussions as suggested.

• Please make sure that there is a proper connection between the two parts of this study. While part 1 is referred to in part 2 there is just one mention of part 2 in the introduction to part 1. The authors are missing the opportunity to properly connect their work.

Reply: Thanks. We have now added a sentence in Sect. 5.2.1 "Interannual variability of AOD" to make closer connection of Part 1 to Part 2. The sentence reads "The statistics of extreme AOD events, and implications for the impact of regional biomass burning processes are provided in Part 2".

Paper 1 on climatology and trend

Major comments:

• Redundant text: lines 160-164 (this is clear), 195/196 (the general time period has already been defined), 215-217 (repeats what has been just stated), 220-224 (reference to Toth et al. 2018 is sufficient, also why consider them in the analysis when you just made the point that artificial AODs of zero are unphysical?), 246-253 (no need to discuss a parameter that has not been used in your work), 280/281 (you can assume that readers are can draw this conclusion themselves), 431-433, 917-936 (redundant or should be part of the data or methodology sections)

Reply: We have removed the lines listed, except 220-224, 246-253, line 280/281 and line 917-936. Line 220-224 is a discussion added after the first round of reviews in response to the other reviewer's comment on the impact on result from the QA process of the CALIOP data. We prefer keeping line 246-253, as our FMF (fine mode fraction) result in Table 1 is different from SMF (sub-micron fraction) results shown in other Arctic AOD studies as cross references, and here we discuss the fundamental cause of the difference in methodology. The Table 1 result in terms of difference between FMF and SMF is discussed in the Sect. 4 line 515-526. For line 280/281, we just want to be explicit without relying on reader's own assumption that MAN data was obtained over water on ships. Line 917-936 is the first paragraph of the Discussion Sect. Here we stress the importance of quality control process of remote sensing data, and provide a parallel comparison with other Arctic AOD studies using off-the-shelf satellite retrieval data. We think the discussion is valuable for future Arctic studies tending to use satellite retrieval AOD, and thus keeping it. We keep this paragraph in the Discussion Section as it is a discussion of the result, while QA processes are already provided in the data and method section.

• I still don't see the need to include Section 2.10 and Figure 12 in the paper. None of the other biomass-burning emission inventories is referred to in that much detail. Also, FLAMBE is used as input to NAAPS-RA and it is not clear why showing FLAMBE maps provides added information to showing findings for BB aerosols from NAAPS-RA.

Reply: We still think FLAMBE biomass burning emission climatology and the trend is an important support for the BB AOD climatology and trend in the Arctic, and it provides BB source information for Part 2, as opposed to the AOD coverage-which is typically not a linear relationship (figure 12 is explicitly referred in Part 2). The added information of providing FLAMBE maps in addition to BB smoke AOD maps is that smoke AOD trend is due to the first order of significance to emission trend in the lower latitudes. And other factors, e.g. mid-latitude to Arctic transport, if they plays a role in smoke AOD trend in the Arctic, would be a second order significance. We have added the following discussion in the new subsection 5.3.4 "Possible causes of BB smoke AOD trends". All other text of this subsection is moved from the previous section 5.3.2 "AOD summertime trends".

"Compared with the BB emission trend, trend in the atmospheric processes, e.g., transport and removals, probably plays a secondary role in the Arctic smoke AOD trend. This is illustrated by the similarity in spatial patterns of smoke AOD and BB emission trends, and the coincidence of peak years for emissions and the high Arctic area-mean smoke AODs. For example, 2012 and 2019 are associated with JJA peaks in emission and high Arctic smoke AOD, while 2003 and 2008 correspond to MAM peaks in both (Figs. 12 and 13)."

• I suggest to stick with fine mode and coarse mode rather then introducing FM and CM. This would increase readability a lot. Right now, the authors switch between using fine mode, FM, an even FM mode…

Reply: Thanks for the suggestion. However, "FM" appeared 60/50 times and "CM" 54/46 times in the text of Part 1/Part 2, which, we think, makes a good reason to abbreviate them. Also these abbreviations in tables help the tables to fit in space. And actually we tried to replace "FM" with "fine mode" and "CM" with "coarse mode" and that increased the length of Part 1 by 1 page (including shifts of figures and tables). So we keep using the FM and CM abbreviations, but we've made sure that these abbreviations are defined when they first appear in the text, and there is no switch back and forth between the abbreviations and the full expressions after that.

• Please make sure that the description of a figure or table is confined to the figure or table caption. The main text should not be used, e.g., to describe what a line of a certain colour represents.

Reply: Thank you. We have now removed such descriptions from the main text, including these for Fig. 2, Fig. 6.

• I expect that most readers are interested in the general findings of the authors' work rather than the peculiarity of individual reanalysis models. I therefore still think that the paper would be much improved if the authors were to focus on the multi reanalysis consensus (MRC) in the figures of the main text. Presenting just the plots for the MRC in Figures 2-7, 10, 11, and 13 doesn't prevent to authors from pointing towards differences in the considered models. If the plots (2-7, 10, 11, 13) and tables (2, 3, 4) for the three individual reanalysis are moved to the supplement, they would still be accessible to readers that are particularly interested in these differences.

Reply: We prefer keeping the results from individual reanalysis along with the MRC, because the diversity and similarity of these reanalyses is indeed part of the main result of the paper. By showing the individual reanalyses side by side with the MRC, we also intend to avoid the consensus being dominated by any specific reanalysis product. We think this makes our result more convincing while providing information for readers interested in the difference of the reanalyses. In addition, although indirectly, this study also serves as an inter-comparison of model performance for the listed models over the Arctic

region. We believe those results should also be of interest to readers who are users of any of the models.

• Please make sure that you properly and specifically refer to figures you are discussing rather than just providing a figure number or a list of figures at the beginning of a paragraph.
Reply: We tried to avoid leading a paragraph with a figure number following the suggestion.  Thanks for your suggestion.

• Section 5.2.1 doesn't really provide an objective assessment of interannual variability and is largely based on referring to individual events that should be discussed in the introduction to Part 2. I suggest to omit this section.

Reply: Section 5.2.1 provides general features of AOD interannual variability and explains some of the large interannual variability signals in monthly AOD time series from AERONET and MRC shown in Figure 2 by providing corresponding known biomass burning cases. Some of the cases were recorded during field campaigns and were well studied. We think these are useful information for understanding the cause of interannual variabilities. To make the purpose clear and connect to Part 2, we have now included the following two sentences below. Also some individual events are already mentioned as examples of extreme AOD events in the introduction of Part 2.
"Some of the BB smoke events cause short-term record-high AOD, and some lasted weeks to months, resulting in high monthly mean AOD. The statistics of extreme AOD events, and implications for the impact of regional biomass burning processes are provided in Part 2."
Beside the above changes, we've also removed a redundant paragraph in this subsection.

• It is not clear to me what is shown in Figure 8 or how the plot has been compiled. Please provide a better description.

Reply: Figure 8 shows the percentage of interannual total AOD variability explained by speciated AODs. We have now added a paragraph to explain how this (and other statistical variables) is calculated in the "Method" section. It reads "For verification purpose, bias, root-mean-square deviation (RMSE) and coefficient of determination (denoted $r^2$) of reanalysis AODs compared to AERONET/MAN AODs are calculated. $r^2$ equals the square of the Pearson correlation coefficient between the observed and the modeled AODs. When estimating contributions of individual species to total AOD interannual variability, $r^2$ is calculated as the square of the Pearson correlation coefficient between the seasonally-binned modeled speciated AOD and total AOD. In that form, $r^2$ provides the percentage of "explained variance" of total AOD by a speciated AOD. The statistical definition and interpretation of $r^2$ can be found https://en.wikipedia.org/wiki/Coefficient_of_determination. "

• There is no discussion of Figure 9. What about moving this figure to an earlier position after Figure 3 so that the presentation of the satellite data is all finished before moving on to the models?
Reply: We discussed Fig. 9 in Sect. 5.3 and mentioned "Fig. 9" twice there and once in Sect. 6. We have now referred Fig. 9 seven times (not adding text, but explicitly referring it in a few more places where applicable). We keep the trend analysis with both the satellite data and the reanalyses under the same section (i.e. Sect. 5.3), so that the trends derived from the two different types of datasets can be compared conveniently.

Minor comments.
• Please update the reference IPCC (2013) to IPCC (2021)
Reply: updated.

• Ice nuclei are now generally referred to as ice nucleating particles (INP)
Reply: updated.

• Line 335: on the other hand requires an earlier on the one hand
Reply: "On the other hand" is now changed to "Furthermore".

• Lines 392-400 should be moved to the introduction
Reply: This part states the reason why biomass burning smoke is treated as a singularly important
species in this study, so we think it belongs to the "Methods" section.

• Line 414: What is it, hourly or daily data?
Reply: Changed to "either hourly or daily"

• Lines 476-481: this should be moved to the methods section
Reply: This part is to explain the difference between the mean and the median as shown in Table 1. Thus
we kept the discussion as a part of the result section.

• Caption Figure 1: Add that the size of the circles refers to the magnitude in AOD.
Reply: Thanks for the suggestion. Added.

• Caption Figure 3: omit second sentence. This has already been stated in the Data section.
Reply: Thanks. The 2nd sentence is now removed in the figure caption.

• Lines 588-605: Remove reference to Figure 3 and move a generalised version of this text to Section 2
Reply: This paragraph describes partial results of Figure 3 and explains the coverage patterns of the
sensors for different seasons. The coverage pattern is better explained in reference to Fig. 3. Therefore
we keep this paragraph here.

• Line 626: CALIOP has a footprint of 70 m.
Reply: Thanks for spotting this. We have updated the text to "The swath for MODIS and MISR is on the
order of a few hundred to a few thousand kilometers, while the "beam diameter" for CALIPSO is on the
order of 70m (Winker et al., 2009; Colarco et al., 2014)." The reference paper, Winker et al., 2009 is
added.

• Line 630: no need to use an acronym for data assimilation. It's used only once and I forgot what DA
was supposed to stand for by the time is was used…
Reply: All "DA" in the text are now expressed explicitly as "data assimilation".

• Figure 4: lines 645-650 should be moved to suitable places in the text. They don't belong into a figure
caption.

Reply: We have moved the text to "Methods" section.

• Figure 5: move the second sentence of the caption into the main text.
Reply: We have removed the 2nd sentence, as the information is already included in the "Methods" section.

• It seems that Figure 7 is discussed before Figure 6. Also the discussion of Figure 7 doesn't seem to be quite objective: no change can be extracted if the error bars were to be considered!
Reply: Figure 7 is discussed after Figure 6. Figure 6 first appears in "Speciated AODs have more variability than total AOD among the three reanalyses, and a little more so for MAM than for JJA (Fig. 4, 5, 6).", and two paragraphs earlier than Fig. 7 in sect. 5.1.2. For Fig. 7, error bars represent monthly AOD variability. It is clear that interannual AOD variability for July and August is much larger than other months.

• Lines 868/869: not clear which figure the authors are referring to
Reply: The text was "For the high Arctic, AOD trends are hardly seen with the same color scale as those for the lower latitudes because of lower AOD. Thus, they are shown separately in Fig. 13….". We have changed "are" to "will" after "AOD trends".

• Line 939: what climate models? Please specify.
Rely: We specify the climate models as AEROCOM and CMIP5 models and give details right after the sentence.

Part 2 on extreme events
General comments
• Please shorten the Abstract to present just the essential findings.
Reply: The abstract is shortened.

• Please treat part 2 as a stand-alone paper. As such, the introduction should give a short review of the findings of part 1.
Reply: Thanks for the suggestion! A brief summary of Part 1 findings is now added at the end of Part 2 introduction section.
• I suggest to restructure the paper to first discuss all findings from AERONET (particularly Table 2) as the observational foundation for your methodology to identify extreme AODs. This would then allow to clearly define which sites are affected by biomass-burning aerosol to which extend. Afterwards, you can move on with the comparison to the reanalysis data (Figure 1 and Table 1) and the contribution of different components (Figure 4).
Reply: Thanks for the suggestion. We seriously thought about restructuring the paper as suggested. However we ended up using the current structure as we would like to provide readers with an impression of NAAPS-RA's performance and then involve NAAPS-RA in later discussions, including general statistics of extreme events as the reviewer proposed. This is also because there are only limited AERONET sites over/near the Arctic region (10 from our study), which may not provide a comprehensive picture of extreme AODs over the region, and thus AERONET data are used as an evaluation tool and supporting dataset in this study.

• Redundant text: 95-102 (not needed and repeated later anyway), 119-121 (method section), 197-200, 425-429, 457-459 (should be clearly described in the methods section)

Reply: Thanks! Line 95-102 is actually not repeated anywhere in Part 2, despite that aerosol cloud impact and albedo impact was introduced in Part 1. But different from what was introduced in Part 1, the two sentences here list the "observable" impacts from extreme aerosol events (Part 1 lists some modeling studies). So we are keeping the lines. We have moved Line 119-121 to "Data and Methods" section, removed line 197-200, line 425-429. Line 457-459 is rewritten and provided in "Data and Methods" section.

• Section 3.3 should be omitted. Parts of its content – when authors list earlier observations of extreme events – should be moved to the introduction. Figure 5 and the brief discussion don't add much insight and should be removed.

Reply: It is our strong preference to include Section 3.3 where Figure 5 (now Figure 4 after removal of Figure 2 following the reviewer's suggestion) resides. Figure 5 gives readers a visual look of an example extreme smoke event from satellite imageries and lidar. We are reluctant to remove this vivid example (and solely extreme smoke example that is discussed in relative detail) in the paper. We've now mentioned the big field campaign examples in the introduction, and left the long list of extreme events in this section. We also added "More extreme BB smoke cases in the Arctic can be found in Sec. 3.3. " after the field campaign examples in the introduction section.

Specific comments

• Line 59: Omit Arctic.

Reply: Done

• Line 82: TOA introduced twice but there's no need to use the acronym at all

Reply: TOA acronym is removed.

• Lines 125-135: move text to the respective subsections in the data section

Reply: Subsection "2.4 Methods" with the text is added.

• Line 153: it's MODIS imagery

Reply: Corrected.

• Line 171: what's the difference between quality controlled and quality assured?

Reply: To avoid confusion, "quality assured" is removed.

• Figure 2: Why does the figure include data from 14 stations when only 10 are listed in Table 1? Is there any discussion of Figure 2? Also, why not just add a line with the regression parameters for all data points to Table 1 and omit this figure?

Reply: We've accepted your suggestion, and removed Fig. 2. The regression parameters from Fig. 2 is now incorporated into Table 1. "14" was a typo.

• Figure 3: The definition of pairwise (or better temporally and spatially matched) should be provided in the method section and not in a figure caption.

Reply: Now the definition of "pairwise" is in "Methods" subsection.

• Figure 6: It is not clear what is shown in the maps. Please properly describe the data treatment in the methods section.
Reply: With the removal of Fig.2, Fig. 6 becomes Fig. 5. In 2.4 "Methods" subsection, there are descriptions as "We define extreme events as those corresponding with AOD exceeding the 95th percentile mark in 6 hr or daily AOD data relative to climatological means at a specific location or across a given region (the region north of 70°N for example)…….. To simplify some of the discussion below, we frequently employed the symbol "$AOD_n$" to represent the AOD associated with the n% percentile of its cumulative (histogram) distribution." We have now added "$AOD_{75}$, $AOD_{90}$, $AOD_{99}$, $AOD_{99.5}$ and maximum AOD are also calculated to show AOD gradient for high AODs."

• Figure 9: please define spread in the methods section. No stars are visible in the plot.
Reply: With the removal of Fig. 2, Fig. 9 is now Fig. 8. We've added, in the figure caption, "The box and whiskers represent AOD at 95, 90, 75, 50, 25, 10, and 5% percentiles." We've also replotted the figure to increase readability by increasing the size of circles, adding stars and moving legend from the right to the bottom. We've removed the "spread" sentence in the figure caption to avoid confusion.

• Table 3: There is no need for Table 3 as it doesn't add to what is shown in Figure 10. Also, there's no discussion of this table except for a brief reference such as see also Table 3
Reply: Thanks. Table 3 is now moved to the supplement as Table S1.

• Conclusions should be renamed to Summary. Also, please don't re-introduce all acronyms.
Reply: "Conclusions" is now renamed to "Summary". And all the definitions of acronyms are removed in the summary section.

Reply to review comments #2

Thank you for addressing and responding to my comments as well as the other reviewer's comments. Overall, the manuscript is improved, especially with the separation of the material into two manuscripts. I recommend publishing as is. My only suggestions for revisions are a handful of minor typos I found in the revised manuscript listed below:
- Figure S2: "seaonal" to "seasonal"
- Line 293: space in-between 6 and hrly ("6hrly" to "6 hrly")
- Line 1191: I think "value" should be "values"?

Reply: We thank the reviewer for the positive review comments. All the listed typos are corrected.